

# The quasilocal degrees of freedom of Yang-Mills theory

Henrique Gomes[1★] and Aldo Riello[2†]

**1** Trinity College, Cambridge University, Cambridge CB2 1TQ, England
**2** Physique Théorique et Mathématique, Université libre de Bruxelles,
Campus Plaine C.P. 231, B-1050 Bruxelles, Belgium

★ gomes.ha@gmail.com, † aldo.riello@ulb.be

## Abstract

Gauge theories possess nonlocal features that, in the presence of boundaries, inevitably lead to subtleties. We employ geometric methods rooted in the functional geometry of the phase space of Yang-Mills theories to: (*1*) characterize a basis for quasilocal degrees of freedom (dof) that is manifestly gauge-covariant also at the boundary; (*2*) tame the non-additivity of the regional symplectic forms upon the gluing of regions; and to (*3*) discuss gauge and global charges in both Abelian and non-Abelian theories from a geometric perspective. Naturally, our analysis leads to splitting the Yang-Mills dof into Coulombic and radiative. Coulombic dof enter the Gauss constraint and are dependent on extra boundary data (the electric flux); radiative dof are unconstrained and independent. The inevitable non-locality of this split is identified as the source of the symplectic non-additivity, i.e. of the appearance of new dof upon the gluing of regions. Remarkably, these new dof are fully determined by the regional radiative dof only. Finally, a direct link is drawn between this split and Dirac's dressed electron.

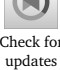

# 1 Introduction and summary of the results

Physical degrees of freedom in gauge theories cannot be completely localized, since gauge-invariant quantities have a certain degree of nonlocality; the prototypical example being a Wilson line.

Here, we will address the problem of defining *quasilocal* degrees of freedom (quasilocal dof) in electromagnetism and Yang-Mills (YM) theories. By "quasilocal", we specifically mean "confined to a finite and bounded region", with a certain degree of nonlocality allowed *within* the region. When the role of the specific region needs to be emphasized, we will call such properties *regional*.

In electromagnetism, or any Abelian YM theory, although the field strength $F_{\mu\nu} = \partial_\mu A_\nu - \partial_\nu A_\mu$ provides a complete set of local gauge-invariant observables, a canonical formulation unveils the underlying nonlocality. The components of $F_{\mu\nu}$ (i.e. the electric and magnetic fields $E$ and $B$) fail to provide gauge-invariant *canonical* coordinates on field space: in 3 space dimensions, $\{E^i(x), B^j(y)\} = \epsilon^{ijk}\partial_k \delta(x,y)$ is not a canonical Poisson bracket and the presence of the derivative on the right-hand-side is the first sign of a nonlocal behaviour. (For a striking

proof of the tension between locality and even gauge-*co*variance in the quantum formalism, see [1, Thm. 8.1].)

From a canonical perspective, the constraint whose Poisson bracket generates gauge transformations, namely the Gauss constraint, is responsible for the non-local attributes of gauge theories—and indeed of most of their peculiar properties (both classical, and quantum [1,2]). The Gauss constraint gives an *elliptic* equation which must be satisfied by initial data on a Cauchy surface $\Sigma$. In other words, the initial values of the fields cannot be freely specified throughout $\Sigma$; for instance the allowed values of the electric field inside a region depend on the distribution of charges within the region and the flux of the electric field at its boundary. Ultimately, this is the source of both the nonlocality and the difficulty of identifying freely specifiable initial data—the "true" dof of the theory. The viewpoint often adopted in the literature is that such nonlocality also prevents the factorizability of gauge-invariant observables and of physical degrees of freedom across regions (e.g. [3–5]). In this paper, we will clarify these statements, characterizing the quasilocal dof of Yang-Mills theory as well as their non-local properties.

That is, we will address the definition of YM quasilocal dof in a linearized setting around a background configuration. We refer to these first-order perturbations as "fluctuations" or, often, as "modes." Geometrically, these modes are identified with tangent vectors to the YM configuration space over a Cauchy hypersurface $\Sigma$ at a certain base-point in configuration space—the background configuration. Such tangent vectors are the basic objects required by the study of symplectic geometry, as encoded in the (pre)symplectic form $\Omega$.

Our approach seamlessly adapts to the treatment of bounded regions $R \subset \Sigma$, $\partial R \neq \emptyset$, without ever requiring any restriction on the dof: *not even in the form of boundary conditions at $\partial R$*. This feature makes our approach uniquely adaptable to the study of arbitrary *fiducial boundaries*—that is, of interfaces that do not presume any boundary condition on the fields—with foreseen applications in e.g. entanglement entropy computations discussed in the outlook section. Although restrictive boundary conditions (see e.g. [6]) on the physical content can in principle be incorporated in the formalism by restricting the definition of the configuration space, we will not analyze this possibility here (we refer to [7] by one of the authors for considerations regarding asymptotic null infinity).

To be more explicit: more than leave boundary conditions open, we *never* fix the gauge freedom, *not even at the boundary*. Manifest covariance, including at the boundary, is the central feature of our approach, lying at the core of all our results. Moreover, this freedom fundamentally distinguishes our approach to gauge theories in regions with either finite or asymptotic boundaries from other standard approaches (e.g. [8–11]—see also [7] for a discussion of this point). Since we also restrain from introducing any additional dof at the boundary, our approach is more economical than the edge-mode approach [12–16] (to be discussed in the concluding section).

This paper is centered on three physical questions: (*1*) How do we characterize the quasilocal dof of YM theory? (*2*) What are their covariantly conserved regional charges and how are these related to the underlying gauge symmetry? And finally, (*3*) how do the quasilocal dof behave upon composition, or gluing, of the underlying regions?

These three questions will be addressed through the development of appropriate mathematical tools, respectively: (*1*) A decomposition of the linearized dof over a region, into a basis that is covariant with respect to gauge transformations of the background configuration. The main tool here is the introduction of a functional connection form over the phase-space of Yang-Mills theory [17–19]. Here we show how the introduction of this connection naturally leads to a split of the dof into Coulombic and radiative. Coulombic dof are those that enter the initial-value Gauss constraint and, in the presence of boundaries, rely on extra independent boundary data—the electric flux. In [20] by one of the authors, this dependence

on boundary data is shown to be at the source of superselection sectors. Within each of these sectors, a quasi-local gauge-reduction procedure can be meaningfully performed. Radiative dof, on the other hand, are unconstrained and independent of any other data: they are the "true" quasi-local degrees *of freedom* of the theory. Although the split itself depends on the choice of functional connection, our results hold for an arbitrary such choice. Nonetheless, a geometrically privileged functional connection exists which satisfies some extra, convenient, properties. We called this connection the Singer–DeWitt (SdW) connection [19]. The gauge-geometry of phase space is described in section 2, while the consequences at the symplectic level are discussed in section 3.

(*2*) Together with [21–23], we will argue that non-trivial global charges can only be associated to reducible configurations of the gauge potential. In Abelian theories, every configuration is reducible (with reducibility parameter the constant "gauge transformations") and global charges admit a Hamiltonian symplectic flow in the reduced quasilocal phase space—notice that the global charges over $\Sigma$, for $\partial\Sigma = \emptyset$, must vanish. In contrast to Abelian theories, in the *non*-Abelian case, reducible configurations are extremely rare (i.e. *ir*reducible configurations are dense in configuration space) and possess an intricate geometric structure [24–29]. This means not only that the physical relevance of global charges in the non-Abelian theories is less clear (fluctuations that are not fine-tuned generically break the global symmetry under study), but also that an extension of our geometric formalism that encompasses non-Abelian reducible configurations would require substantially more work. For these reasons, in this article we limit ourselves to laying down some general considerations on the non-Abelian case and leave the detailed analysis of the symplectic geometry associated to these charges to future work. Charges are discussed in section 4. The relationship of this formalism with Dirac's dressed electron is explained in section 5.

(*3*) Our analysis of the gluing of the YM dof across adjacent regions leverages a novel gluing-theorem that we prove in the case of (topologically trivial) bipartite systems. This theorem shows that: (i) the regional *radiative* dof are sufficient to reconstruct the global symmetry-reduced symplectic form; and yet (ii) the composition of the radiative symplectic forms is non-additive, i.e. that the global symmetry-reduced symplectic form contains (in a precise sense) more dof than the combination of the regional radiative ones. This is the classical analogue of the non-factorizability of the Hilbert spaces of (lattice) gauge theory. Remarkably, in the SdW case, the gluing theorem leads to an explicit gluing formula for the radiative dof which shows that the "missing" dof that emerge upon gluing are indeed encoded in the *mismatch* between the two regional radiatives across the interface. As the gluing theorem shows, at a generic configuration of the non-Abelian theory, if gluing is possible—i.e. if the two radiatives can be composed at all—then it is unique. However, at reducible configurations, and in the presence of matter, gluing is ambiguous due to the presence of the non-trivial global symmetries analyzed in (2). This is particularly relevant in the Abelian case, where the ambiguity is related to the total regional electric charge. Finally, we explore in a simple 1-dimensional case the consequences of non-trivial space topology and the emergence of Aharonov-Bohm phases within out formalism. Gluing is discussed in section 6.

Crucially, the key feature in all these results is the nonlocal nature of the "physical dof" of Yang–Mills theory, a property which is manifest in our answer to (*3*).

Of course, this nonlocality is a property that we expect Yang–Mills theory to share with (all) other gauge theories—such as Chern-Simons theory. For example the decomposition of linear fluctuations along gauge and transverse directions in field space, as well as the results on their gluing, apply to any gauge theory described by a Lie-algebra valued gauge potential *A*. Having said that, precise statements on the nature of the dof of a gauge theory can rely only upon a detailed analysis of the *symplectic structure* of the theory, especially in relation to gauge transformations. And since this analysis can only be performed on a theory-by-theory basis,

the conclusions we draw in this paper only apply—strictly speaking—to Yang–Mills theory.

We conclude our discussion in section 7 with a brief outlook. A list of symbols can be found in appendix C.

## 2 Field-space geometry: setup and definitions

This section will set the stage for our future considerations. It mostly reviews constructions and results that have already appeared in our previous work [17, 19]. Nonetheless, the inclusion of this material aims for more than just reviewing: our current presentation will be more rigorous, complete, and systematic than those previously available. Throughout this article we will not strive for functional analytic rigour: our constructions will rather focus on the algebraic aspects of the geometry of field space.

Most of the field-space objects introduced in this paper are understood within the setting of "local" calculus in the sense of the pullback from the (infinite) jet bundle, and not in the setting of general differential geometry on Frechet manifolds. For example, the "cotangent bundle" of the space of connection $\mathcal{A}$ introduced later is the fiberwise dualisation of the vector bundle whose sections are the fields. However, as it will become clear later on, these local spaces have to be slightly generalized to introduce certain nonlocal objects such as Green's functions. We will not attempt a rigorous characterization of this extension.

Before starting we notice one important remark: all the constructions will be performed at the quasilocal level, by formally replacing a Cauchy surface $\Sigma$ with any compact subregion $R$ thereof, with $\partial R \neq \emptyset$. Since our interest lies mostly in bounded regions, we take this replacement for granted. Motivated by the study of subregions of $\Sigma$ defined by fiducial boundaries, in the following we will assume *no* boundary condition at $\partial R$, not even in the allowed gauge freedom. Unless otherwise specified, all integrals are understood to be over $R$, i.e. $\int := \int_R d^D x$, and all boundary integrals over $\partial R$, i.e. $\oint := \int_{\partial R} d^{D-1} x$.

### 2.1 Horizontal splittings in configuration space

To start, we introduce notation and recall some basic facts.

Consider a Lagrangian $D + 1$ formulation of YM theory on a globally hyperbolic spacetime $M \cong \Sigma \times \mathbb{R}$ foliated by equal-time Cauchy surfaces[1] $\Sigma_t \cong \Sigma$.

To distinguish issues of global (topological) nature—which will only be considered in section 6.8—from those associated with finite boundaries—which constitute our main focus,—we assume $\Sigma \cong \mathbb{R}^D$. This choice is made for mere convenience and will play no role in the following where our focus will be on compact subregions $R \subset \Sigma$, diffeomorphic to a $D$-disk.

Denote the corresponding *quasilocal YM configuration space* $\mathcal{A}$ (see figure figure 1). This is the space of Lie-algebra valued one-forms on $R \subset \Sigma$,[2]

$$A \in \mathcal{A} := \Omega^1(R, \mathrm{Lie}(G)). \tag{1}$$

---

[1]Concerning the extrinsic geometry of our foliation, i.e. how $\Sigma_t$ is embedded in spacetime: Unless stated otherwiese, all our formulae we will hold when $\Sigma$ belongs to an Eulerian foliation of spacetime, i.e. to a foliation whose lapse is equal to one and whose shift vanishes. In other words, $\Sigma$ is an equal-time hypersurface in a spacetime with metric $ds^2 = -dt^2 + g_{ij}(t, x)dx^i dx^j$. The inclusion of nontrivial lapse and shift is in principle straightforward, but makes some formulae more cluttered, and most likely wouldn't add much to our considerations here. However, we point the reader to [7] for a situation where the introduction of a nontrivial shift plays a crucial role in dealing with asymptotic gauge transformations and charges.

[2]Rigorously speaking, dealing with a non-compact Cauchy surface would require us to consider only fields that vanish fast enough at infinity. However, our focus on compact region will make this restriction virtually irrelevant in the following. Therefore, we do not concern ourselves with a precise determination of the fall off rates and hereafter neglect them completely. For an application of our formalism where asymptotic conditions at null infinity are carefully treated, see [7].

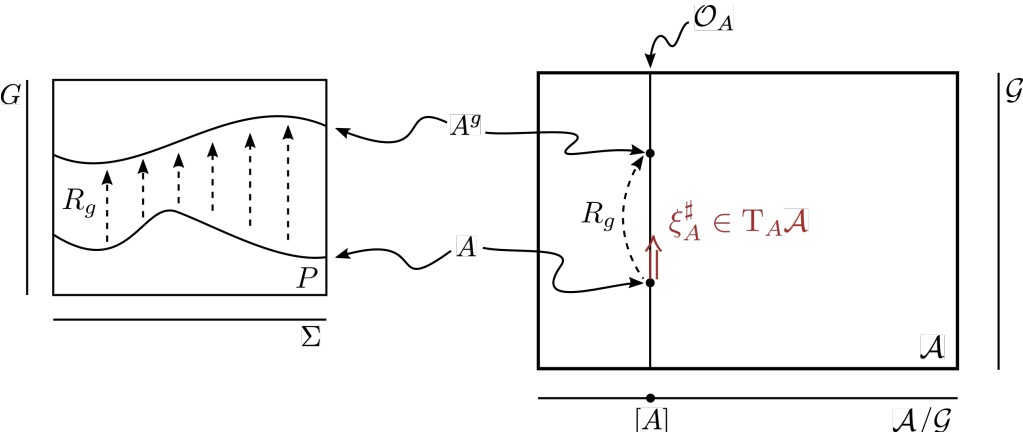

Figure 1: A pictorial representation of the configuration space $\mathcal{A}$ seen as a principal fibre bundle, on the right. We have highlighted a generic configuration $A$, its (gauge-transformed) image under the action of $R_g : A \mapsto A^g$, and its orbit $\mathcal{O}_A \cong \mathcal{G}$. We have also represented the quotient space of 'gauge-invariant configurations' $\mathcal{A}/\mathcal{G}$. On the left hand side of the picture, we have "zoomed into" a representation of $A$ and $A^g$ as two gauge-related local sections of a connection $\omega$ on $P$, the finite dimensional principal fibre bundle with structural group $G$ over $R$. The principal fibre bundle picture of $\mathcal{A}$ will be partially revisited in section 4—see figure 8.

Since we will be using a Hamiltonian (phase-space) framework, the component of $A$ in the transverse direction to $\Sigma$, $A_0$, is left out of the description.

The group $G$ is assumed to be compact and semisimple and will be referred to as the *charge group* of the theory. In specific applications, we will have $G = \mathrm{SU}(N)$ in mind. We write, $A = A_i \mathrm{d}x^i = A_i^\alpha \mathrm{d}x^i \tau_\alpha$, where $\{\tau_a\}$ is a basis of generators of $\mathrm{Lie}(G)$ which is orthonormal with respect to a rescaled Killing form on $\mathrm{Lie}(G)$, i.e. $\frac{1}{2N}\mathrm{k}(\tau_\alpha, \tau_\beta) = \mathrm{Tr}(\tau_\alpha \tau_\beta) = \delta_{\alpha,\beta}$.

The space of gauge transformations i.e. the space of smooth (compactly supported) $G$-valued functions on $\Sigma$, $\mathcal{C}_o^\infty(\Sigma, G)$ inherits a group structure from $G$ via pointwise multiplication. This group is in general not connected. Although this fact has crucial physical consequences, in this article we shall be concerned exclusively with the properties of infinitesimal gauge transformations, thus turning a blind eye to these issues.[3] Most often, we shall focus on the space of *quasilocal* gauge transformations within $R \subset \Sigma$, which we call the *gauge group* and indicate by

$$\mathcal{G} := \mathcal{C}^\infty(R, G) \ni g. \tag{2}$$

The gauge transformation $g : R \to G$ acts on the gauge potential's configuration $A$ as

$$R_g : \mathcal{A} \to \mathcal{A}, \quad A \mapsto A^g = g^{-1}Ag + g^{-1}\mathrm{d}g. \tag{3}$$

This defines an action of $\mathcal{G}$ on $\mathcal{A}$. The orbits of this action, $\mathcal{O}_A$, are called gauge orbits and they define a foliation[4] of $\mathcal{A}$, denoted $\mathcal{F} = \{\mathcal{O}_A\}$ and called the *vertical foliation* of $\mathcal{A}$. The space of orbits, $\mathcal{P} \cong \mathcal{A}/\mathcal{G}$, is the "gauge-invariant" space of configurations which is only defined abstractly through an equivalence relation, and is most often inaccessible for practical

---

[3]The non-connectedness of $\mathcal{G}$ has physical consequences e.g. for chiral symmetry breaking in the full quantum theory; for a thorough discussion see [30].

[4]$\mathcal{G}$ does not act freely on every orbit. Indeed, certain configurations $A \in \mathcal{A}$, said reducible, admit a *finite-dimensional* stabilizer. For more on this, see section 4 and in particular appendix B, where the consequences of this fact will be explored. Until then, we will ignore this complication.

purposes. Rigorous mathematical work has shown that $\mathcal{A}$ and the vertical foliation $\mathcal{F}$ provide indeed (locally[5]) a principal fibre bundle structure with $\mathcal{G}$ the structure group [24–29].

We will denote the tangent bundle to the vertical foliation by $V := \mathrm{T}\mathcal{F} \subset \mathrm{T}\mathcal{A}$.

An infinitesimal gauge transformation $\xi \in \mathrm{Lie}(\mathcal{G}) \cong \mathcal{C}^\infty(R, \mathrm{Lie}(G))$ defines a vector field tangent to $\mathcal{F}$. This is denoted by $\xi^\sharp \in V$, and its value at $A$ is

$$\xi_A^\sharp = \int (\mathrm{D}_i \xi)^\alpha(x) \frac{\delta}{\delta A_i^\alpha(x)} \in \mathrm{T}_A \mathcal{O}_A \subset \mathrm{T}_A \mathcal{A}, \tag{4}$$

where $\mathrm{D}_i \xi := \partial_i \xi + [A_i, \xi]$ is the gauge-covariant derivative in the adjoint representation. Clearly, at $A \in \mathcal{A}$, $V_A = \mathrm{Span}(\{\xi_A^\sharp\}_{\xi \in \mathrm{Lie}(\mathcal{G})})$. Thus, we say that $V$ comprises the "pure gauge directions" in $\mathcal{A}$.

Later applications, such as the study of charges and especially gluing, require us to consider so-called "*field-dependent* gauge transformations". Let us first provide a heuristic intuition of this concept: field-dependent gauge transformations correspond to choices of different $\xi \in \mathrm{Lie}(\mathcal{G})$'s at different configurations $A \in \mathcal{A}$ (hence their "field dependence"). Note that the definition of $\xi^\sharp$ (4) holds point-wise on $\mathcal{A}$ and can thus be canonically extended to the field-dependent case. This leads to field-dependent gauge transformations being associated to *generic* vertical vector fields in $V \subset \mathrm{T}\mathcal{A}$.

These heuristic ideas can be formalized by introducing the *action* (or *transformation*) *Lie algebroid* $(\mathfrak{A}, \cdot^\sharp, \mathcal{A})$ associated to the action of $\mathcal{G}$ on $\mathcal{A}$ (see e.g. [31]). Here, $\mathfrak{A} = \mathcal{A} \times \mathrm{Lie}(\mathcal{G})$ is a trivial bundle on $\mathcal{A}$; $\xi$ is promoted to a (non-necessarily constant) section of $\mathfrak{A}$, i.e.

$$\xi \in \Gamma(\mathcal{A}, \mathfrak{A}) \cong \Omega^0(\mathcal{A}, \mathrm{Lie}(\mathcal{G})); \tag{5}$$

and the anchor $\cdot^\sharp : \mathfrak{A} \to \mathrm{T}\mathcal{A}$ is still defined through (4). The Lie algebroid $(\mathfrak{A}, \cdot^\sharp, \mathcal{A})$ is canonically isomorphic to the Lie algebroid of the foliation $\mathcal{F} \subset \mathrm{T}M$, understood as the canonical Lie algebroid of vertical vector fields endowed with their Lie bracket.

An important formula is the isomorphism between, on one side, the Lie bracket $[\![\cdot, \cdot]\!]_{\mathrm{T}\mathcal{A}}$ between vectors in $\mathrm{T}\mathcal{A}$ and, on the other, the action Lie algebroid bracket in $\mathfrak{A}$. This isomorphism can be expressed more elementarily in terms of the Lie bracket $[\cdot, \cdot]$ of $\mathrm{Lie}(\mathcal{G})$—which is a point-wise extension of the Lie bracket on $\mathrm{Lie}(G)$,—according to:

$$[\![\xi^\sharp, \eta^\sharp]\!]_{\mathrm{T}\mathcal{A}} = \left([\xi, \eta] + \xi^\sharp(\eta) - \eta^\sharp(\xi)\right)^\sharp. \tag{6}$$

On the right-hand side $\xi$ and $\eta$ are treated as zero-forms on $\mathcal{A}$ with values in $\mathrm{Lie}(\mathcal{G})$, thus $\xi^\sharp(\eta) \equiv \xi^\sharp(\eta^\alpha) \tau_\alpha \in \mathrm{Lie}(\mathcal{G})$.

Moreover, the formulation in terms of Lie algebroids not only allows us to formalize the notion of "field-dependent" gauge transformations, but also opens the door to future generalizations of our framework, e.g. general relativity in the formalism of [32, 33].

In terms of the action Lie algebroid, field-*in*dependent gauge transformations are constant sections in $\Gamma(\mathcal{A}, \mathfrak{A}) \cong \Omega^0(\mathcal{A}, \mathrm{Lie}(\mathcal{G}))$. Introducing a formal de-Rham differential $\mathbb{d}$ on $\mathcal{A}$, this condition reads $\mathbb{d}\xi = 0$. Since field-independent gauge transformations play a distinguished role in our framework, we expect that generalizations beyond the action Lie-algebroid will involve Lie algebroids equipped with a connection, i.e. $(\mathfrak{A}, \cdot^\sharp, \mathcal{A}, \mathbb{D})$ with $\mathbb{D} : \Gamma(\mathcal{A}, \mathfrak{A}) \to \Omega^1(\mathcal{A}) \otimes \Gamma(\mathcal{A}, \mathfrak{A})$: indeed this allows to generalize the field-independence condition to $\mathbb{D}\xi = 0$ (see also [34]). An action Lie algebroid like the one appearing in YM theory comes equipped with the canonical flat connection $\mathbb{D} = \mathbb{d}$, $\mathbb{d}^2 \equiv 0$.

Since vertical directions in $\mathrm{T}\mathcal{A}$ are identified with pure-gauge directions, the 'physical' directions can be defined as those transverse to $V$. Thus, physical directions are encoded in a

---

[5]Cf. previous footnote.

complementary distribution $H \subset \mathrm{T}\mathcal{A}$, $H \oplus V = \mathrm{T}\mathcal{A}$, that we call the "horizontal" distribution. The decomposition $H \oplus V = \mathrm{T}\mathcal{A}$ is however not canonically defined.

The *choice* of any such decomposition that is compatible with the gauge structure of $\mathcal{A}$ is encoded in the choice of an Ehresmann connection on $\mathcal{A}$ valued in $\mathrm{Lie}(\mathcal{G})$, that we call $\varpi$, which satisfies two compatibility conditions.

**Definition 2.1** (Functional connection[6] [19])**.** *Let*

$$\varpi \in \Omega^1(\mathcal{A}, \mathrm{Lie}(\mathcal{G})), \tag{7}$$

*then $\varpi$ is said a $\mathcal{G}$-compatible functional connection form on $\mathcal{A}$, or simply a* functional connection*, if it satisfies the following properties for all field-dependent gauge transformations $\xi$:*

$$\begin{cases} \mathbb{i}_{\xi^\sharp}\varpi = \xi\,, \\ \mathbb{L}_{\xi^\sharp}\varpi = [\varpi, \xi] + \mathbb{d}\xi\,. \end{cases} \tag{8}$$

*We will call these properties the* projection *and* covariance *properties, respectively.*[7]

Notice that this definition demands $\varpi$ to be a local 1-form over field-space, $\mathcal{A}$, but says nothing on its locality properties over space, $\Sigma$. Indeed, as we will see in section 2.2, $\varpi(A)$ will be a nonlocal functional of $A(x)$. We will come back on this point shortly.

Hereafter, double-struck symbols refer to geometrical objects and operations in configuration space: $\mathbb{d}$ is the (formal) field-space de Rham differential,[8,9] $\mathbb{i}$ is the inclusion operator of field-space vectors into field-space forms, and $\mathbb{L}_{\mathbb{X}}$ is the field-space Lie derivative along the vector field $\mathbb{X} \in \mathfrak{X}^1(\mathcal{A})$. Its action on field-space forms is given by Cartan's formula, $\mathbb{L}_{\mathbb{X}} = \mathbb{i}_{\mathbb{X}}\mathbb{d} + \mathbb{d}\mathbb{i}_{\mathbb{X}}$. Finally, the curly wedge $\curlywedge$ will denote the wedge product in $\Omega^\bullet(\mathcal{A})$, where $\bullet$ stands in for arbitrary degrees.

The projection property means that $\varpi$ defines a horizontal complement $H$ to the fixed vertical space $V$, via

$$H := \ker(\varpi)\,. \tag{9}$$

The horizontal projector $\widehat{H} : \mathrm{T}\mathcal{A} \to H$ is thus given by $\mathbb{X} \mapsto \widehat{H}(\mathbb{X}) := \mathbb{X} - \varpi(\mathbb{X})^\sharp$. See figure 2.

The covariance property intertwines the action of vertical vector fields on 1-forms over $\mathcal{A}$ (the lhs) to the adjoint action of $\mathrm{Lie}(\mathcal{G})$ on itself (the rhs). This condition ensures the compatibility of the above definition with the group action of $\mathcal{G}$ on $\mathcal{A}$, i.e. it embodies the covariance of $\varpi$ under gauge transformations. The term $\mathbb{d}\xi$ on the right hand side of the covariance property is only present if $\xi$ is an infinitesimal *field-dependent* gauge transformation. Using Cartan's formula, its presence can be deduced from the covariance of $\varpi$ under field-independent gauge transformation and the projection property of $\varpi$ which holds pointwise in field-space (see [19]).

**Remark 2.2** (On nonlocality)**.** *Since a gauge transformation transforms $A$ by a derivative of the gauge parameter, in order to satisfy the projection property, $\varpi$ must be nonlocal over $\Sigma$. Indeed, recalling that on $\mathrm{T}\mathcal{A}$, $\xi^\sharp = \int \mathrm{D}\xi \frac{\delta}{\delta A}$ (4), the projection property $\mathbb{i}_{\xi^\sharp}\varpi = \xi$ can be formally rewritten as $\varpi(\mathrm{D}\xi) = \xi$. From this perspective, $\varpi$ is morally the inverse operator to the covariant*

---

[6]Cf. [35] for the finite dimensional case.

[7]In the non-Abelian theory, this definition is viable only within the dense subset of irreducible configurations. In the Abelian theory, this definition requires an adjustment to the definition of $\mathcal{G}$ with important physical consequences. Discussion of these issues is postponed until section 4.

[8]We prefer this notation to the more common $\delta$, because the latter is often used to indicate vectors as well as forms, hence creating possible confusions.

[9]More concretely, given a zero-form $\mathcal{S} \in \Omega^0(\mathcal{A})$, i.e. a functional $\mathcal{S} : \mathcal{A} \to \mathbb{R}$, and a vector field $\mathbb{X} = \int X_A \frac{\delta}{\delta A} \in \mathfrak{X}^1(\mathcal{A})$, one has that $\mathbb{X}(\mathcal{S}) \equiv \mathbb{i}_{\mathbb{X}}\mathbb{d}\mathcal{S} = \lim_{\epsilon \to 0} \frac{1}{\epsilon}\big(\mathcal{S}(A + \epsilon X_A) - \mathcal{S}(A)\big)$. Hence, $\mathbb{d}\mathcal{S}$ is the Fréchet differential of $\mathcal{S}$. In the following, we will simply assume that these differential exist for the class of vector fields we are interested in. We will not pursue functional analytic questions.

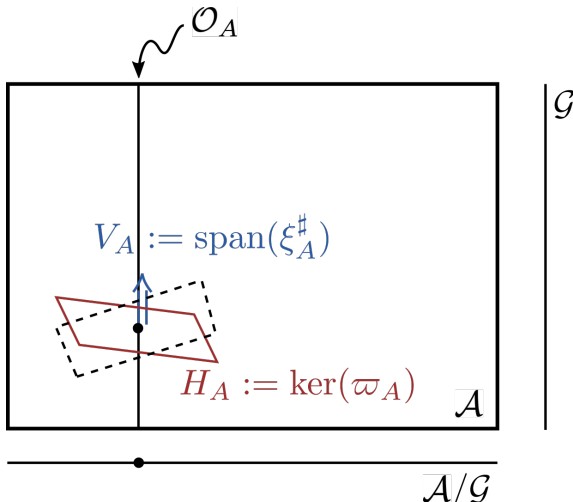

Figure 2: A pictorial representation of the split of $T_A\mathcal{A}$ into a vertical subspace $V_A$ spanned by $\{\xi^\sharp_A, \xi \in \text{Lie}(\mathcal{G})\}$ and its horizontal complement $H_A$ defined as the kernel at $A$ of a functional connection $\varpi$. With dotted lines, we represent a different choice of horizontal complement associated to a different choice of $\varpi$.

*derivative* $D = d + A$ *and as such it must be an integral operator. That is, making it explicit that* $\varpi$ *is valued in* $\text{Lie}(\mathcal{G})$, $\varpi$ *is expected to be of the form*

$$\varpi^\alpha(x) = \int dy \, \varpi^{\alpha,j}{}_\beta(x,y) \mathrm{d}A^\beta_j(y), \tag{10}$$

*for some integral kernel* $\varpi^{\alpha,j}{}_\beta(x,y)$. *Then the equation* $\xi = \varpi(\xi^\sharp)$ *reads:*

$$\xi^\alpha(x) = \int dy \, \varpi^{\alpha,j}{}_\beta(x,y) D_j \xi^\beta(y). \tag{11}$$

*In section 2.2, we will introduce an explicit example of functional connection that has this form (see also section 5 for a well-known realization in electromagnetism).*

*Conversely, by working over the space of matter fields that transform homogeneously under gauge transformations (no derivatives involved), spatially-local functional connections can be constructed. See e.g. [19, Sect. 7].*

Given a functional connection form satisfying (8), alongside $\mathrm{d}$ we can introduce the horizontal differential, $\mathrm{d}_H$ [17–19]. Horizontal differentials are by definition transverse to the vertical, pure gauge, directions:

**Definition 2.3** (Horizontal differential). *The horizontal differential* $\mathrm{d}_H\mu$ *of a form* $\mu \in \Omega^k(\mathcal{A})$ *is the* $(k+1)$-*form such that* $\mathbb{i}_{\mathbb{X}}\mathrm{d}_H\mu := \mathbb{i}_{\widehat{H}(\mathbb{X})}\mathrm{d}\mu$ *for all* $\mathbb{X} \in T\mathcal{A}$.

Of course, the definition implies $\mathbb{i}_{\xi^\sharp}\mathrm{d}_H\mu \equiv 0$.

The following proposition shows that a simpler, and more intuitive, characterization of $\mathrm{d}_H$ in terms of a "$\varpi$-covariant" differential on field space can be given for horizontal differentials of *horizontal and equivariant* field-space forms of general degree.

For example, one could consider a $\lambda \in \Omega^k(\mathcal{A}) \otimes \Gamma(\Sigma, W)$ such that for all field-independent $\xi$'s ($\mathrm{d}\xi = 0$) satisfies (i) $\mathbb{i}_{\xi^\sharp}\lambda = 0$ (horizontality) and (ii) $\mathbb{L}_{\xi^\sharp}\lambda^a = -(R(\xi))^a{}_b\lambda^b$ (equivariance), where $(W, R)$ is a representation of $G$, and $a, b$ are indices in the vector space $W$. Then:

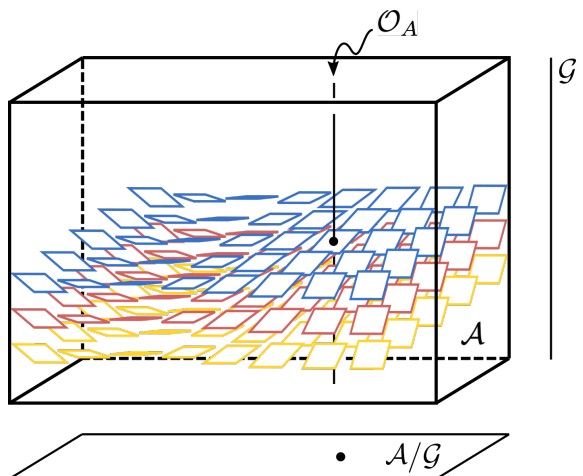

Figure 3: Pictorial representation of anholonomic horizontal plances in $\mathcal{A}$, corresponding to a non-vanishing curvature $\mathbb{F} \neq 0$.

**Proposition 2.4.** *The horizontal differential of a horizontal and equivariant form* $\lambda \in \Omega^k(\mathcal{A}) \otimes \Gamma(\Sigma, W)$ *is itself horizontal and equivariant, and it is given by*

$$\mathbb{d}_H \lambda^a = \mathbb{d}\lambda^a + (R(\varpi))^a{}_b \curlywedge \lambda^b \in \Omega^{k+1}(\mathcal{A}) \otimes \Gamma(\Sigma, W), \tag{12}$$

*where* $R(\varpi) \in \Omega^1(\mathcal{A}) \otimes \Gamma(\Sigma, \mathrm{End}(W))$ *is constructed from the representation* $R : \mathrm{Lie}(G) \to \mathrm{End}(W)$ *and the connection form* $\varpi \in \Omega^1(\mathcal{A}, \mathrm{Lie}(\mathcal{G})) \cong \Omega^1(\mathcal{A}) \otimes \Gamma(\Sigma, \mathrm{Lie}(G))$ *in the obvious way.*

*Proof.* The proposition is a straightforward application of (8), the properties of $\lambda$, and the anticommutativity of $\mathbb{d}$ and $\curlywedge$; see [17]. $\qquad\square$

The all-important horizontal differential of $A_i$, seen as a "coordinate" map from $\mathcal{A}$ to $\Omega^1(\Sigma, \mathrm{Lie}(G))$ is characterized by the following:

**Proposition 2.5.** *The horizontal differential of* $A_i$ *is given by*

$$\mathbb{d}_H A_i = \mathbb{d}A_i - \mathrm{D}_i \varpi, \tag{13}$$

*and it is equivariant under any (possibly field-dependent) gauge transformation, that is*

$$\mathbb{L}_{\xi^\sharp} \mathbb{d}_H A_i = [\xi, \mathbb{d}_H A_i]. \tag{14}$$

*Proof.* These two statements can be easily checked using (8). $\qquad\square$

A central property of the horizontal distribution $H \subset \mathrm{T}\mathcal{A}$ is its anholonomicity, i.e. its non-integrability in the sense of Frobenius theorem—figure 3. As standard, this is characterized by failure of the Lie bracket between two horizontal vector fields to be itself horizontal. Thanks to the projection property of $\varpi$, this quantity can be encoded in the following definition:

**Definition 2.6** (Functional curvature [19, 36])**.** *Given a functional connection* $\varpi$, *the anholonomicity of the associated horizontal distribution* $H_\varpi = \ker(\varpi) \subset \mathrm{T}\mathcal{A}$ *as quantified by the functional two-form*

$$\mathbb{F}_\varpi := \varpi(\llbracket \widehat{H}(\cdot), \widehat{H}(\cdot) \rrbracket) \in \Omega^2(\mathcal{A}, \mathrm{Lie}(\mathcal{G})), \tag{15}$$

*is called the* functional curvature *of the functional connection* $\varpi$. *The subscript* $\bullet_\varpi$ *will generally be omitted.*

As standard in the theory of principal fibre bundles, the curvature of $\varpi$ satisfies the following properties

**Proposition 2.7.** *The curvature $\mathbb{F}$ of $\varpi$ is horizontal $\mathring{\imath}_{\xi^\sharp}\mathbb{F} \equiv 0$, equivariant $\mathbb{L}_{\xi^\sharp}\mathbb{F} = [\mathbb{F}, \xi]$, and its horizontal differential satisfies the algebraic Bianchi identity $\mathbb{d}_H\mathbb{F} \equiv 0$. Moreover, $\mathbb{F}$ can be expressed as*

$$\mathbb{F} = \mathbb{d}_H\varpi \equiv \mathbb{d}\varpi + \tfrac{1}{2}[\varpi \overset{\wedge}{,} \varpi]. \tag{16}$$

*Proof.* Horizontality is manifest from the definition of $\mathbb{F}$. The equivalence between the definitions (15) and the expressions of (16) is standard and can be checked using (8), (9) and Cartan's calculus.[10] Once the right-most formula of (16) has been established, the other properties can be checked by direct computation. □

We conclude this section with a (new) simple proposition which will help us clarify the relationship between $\varpi$ and gauge fixings in section 3.

**Lemma 2.8** (On exact connection forms). *The functional connection $\varpi$ is exact, i.e. $\varpi = \mathbb{d}\varsigma$ for some $\varsigma \in \Omega^0(\mathcal{A}, \mathrm{Lie}(\mathcal{G}))$, if and only if $G$ is Abelian and $\varpi$ is flat.*

*Proof.* If $G$ is Abelian, it follows from (16) and the affine nature of $\mathcal{A}$ that $\varpi$ is exact if and only if $\varpi$ is flat. Conversely, assume that $\varpi = \mathbb{d}\varsigma$. Then, through Cartan's formula, the projection property (8) implies

$$\mathbb{L}_{\xi^\sharp}\varsigma = \mathring{\imath}_{\xi^\sharp}\varpi = \xi, \tag{17}$$

for all $\xi \in \Omega^0(\mathcal{A}, \mathrm{Lie}(\mathcal{G}))$. From this, $\mathbb{L}_{\xi^\sharp}\varpi = \mathbb{L}_{\xi^\sharp}\mathbb{d}\varsigma = \mathbb{d}\mathbb{L}_{\xi^\sharp}\varsigma = \mathbb{d}\xi$. Comparing this formula with the second of (8), it follows that for all $\xi$, $[\xi, \varpi] = 0$. By contracting with an arbitrary $\eta^\sharp$ and using again the projection property, one concludes that $G$ is Abelian. □

## 2.2 Metric structure on $\mathcal{A}$ and the Singer-DeWitt connection

Consider a positive-definite (super)metric on $\mathcal{A}$, i.e. $\mathbb{G} \in \Gamma(\mathrm{T}^*\mathcal{A} \otimes_S \mathrm{T}^*\mathcal{A})$, with $\otimes_S$ standing for the symmetric part of the tensor product. Through such a metric one can fix a notion of horizontality via the condition of orthogonality to the vertical foliation $\mathcal{F} \subset \mathcal{A}$:

$$H_{\mathbb{G}} := (\mathrm{T}\mathcal{F})^\perp \equiv V^\perp. \tag{18}$$

The question is whether such a notion of horizontality can be encoded in a connection form, i.e. if it is gauge-covariant along the orbits. In [19], we showed that this is the case if and only if $\mathbb{G}$ is gauge compatible in the following sense:[11]

**Definition 2.9** (Gauge compatible supermetric). *A supermetric $\mathbb{G} \in \Gamma(\mathrm{T}^*\mathcal{A} \otimes_S \mathrm{T}^*\mathcal{A})$ is said gauge compatible if*

$$(\mathbb{L}_{\xi^\sharp}\mathbb{G})(\eta^\sharp, \mathbb{h}) = 0, \qquad (\mathbb{d}\xi = 0), \tag{19}$$

*holds for all gauge-independent[12] gauge transformation $\xi \in \mathrm{Lie}(\mathcal{G})$, all arbitrary vertical vectors $\eta^\sharp \in V$, and arbitrary horizontal vectors $\mathbb{h} \in H_{\mathbb{G}}$.*

---

[10]See e.g. [19, Sect. 4.2].

[11]Notice that this notion of gauge-compatibility for the supermetric is different from that for a "bundle-like" metric common in the mathematical literature (e.g. as a sufficient condition for the existence of Ehresmann connections [37,38]). The bundle-like condition can be written without reference to field-independent vertical vectors and involves the inner product of two horizontal vectors, rather than of one vertical and one horizontal vector. Although we won't make use of it, we write here, as a reference, the the bundle-like condition in our (infinite dimensional) notation: $(\mathbb{L}_{\eta^\sharp}\mathbb{G})(\mathbb{h}, \mathbb{h}') = 0$ for all $\eta^\sharp \in V$ and $\mathbb{h}, \mathbb{h}' \in H_{\mathbb{G}}$.

[12]Notice that this condition requires a notion of field-independence for the $\xi$'s which is automatic in the YM context (which is described by an action Lie algebroid), but might not be obvious in the context of a more general Lie-algebroid over some configuration space). Cf. footnote 15.

**Proposition 2.10.** *Let $\mathbb{G}$ be a gauge compatible supermetric. Then the following equation implicitly defines a $\varpi_{\mathbb{G}}$ satisfying the defining properties* (8),

$$\mathbb{G}(\xi^{\sharp}, \mathbb{X} - \varpi_{\mathbb{G}}^{\sharp}(\mathbb{X})) \stackrel{!}{=} 0 \quad \forall \xi, \mathbb{X}. \tag{20}$$

*Proof.* See [19, Section 4.1]. $\qquad\qquad\square$

In YM theory, a most natural choice of supermetric is given by inspecting its second-order Lagrangian, and in particular its kinetic term. In temporal gauge, on the $(D+1)$-dimensional spacetime $M \cong \Sigma \times \mathbb{R}$, this is $L = K - U$ with potential

$$U = \tfrac{1}{4} \int_{\Sigma} \mathrm{d}^D x \sqrt{g}\, g^{ii'} g^{jj'} \mathrm{Tr}(F_{ij} F_{i'j'}), \tag{21}$$

where $F_{ij} = 2\partial_{[i} A_{j]} + [A_i, A_j]$, and with kinetic term

$$K = \tfrac{1}{2} \int_{\Sigma} \mathrm{d}^D x \sqrt{g}\, g^{ij} \mathrm{Tr}(\dot{A}_i \dot{A}_j) = \tfrac{1}{2} \mathbb{G}(\dot{\mathbb{A}}, \dot{\mathbb{A}}). \tag{22}$$

In the last term we have introduced the velocity vector[13] $\dot{\mathbb{A}} = \int \dot{A} \frac{\delta}{\delta A} \in \mathrm{T}\mathcal{A}$, as well as the kinetic supermetric $\mathbb{G}$:

**Definition 2.11** (Kinetic supermetric). *On the quasilocal configuration space of YM theory $\mathcal{A}$, the* kinetic supermetric *is defined as*

$$\mathbb{G}(\mathbb{X}, \mathbb{Y}) := \int_R \mathrm{d}^D x \sqrt{g}\, g^{ij} \mathrm{Tr}(\mathbb{X}_i \mathbb{Y}_j), \qquad \forall \mathbb{X}, \mathbb{Y} \in \mathrm{T}\mathcal{A}. \tag{23}$$

*From now on the symbol $\mathbb{G}$ will refer exclusively to the kinetic supermetric* (23).

It is then straightforward to prove that

**Proposition 2.12.** *The kinetic supermetric $\mathbb{G}$ is gauge invariant, i.e.*[14,15]

$$\mathbb{L}_{\xi^{\sharp}} \mathbb{G} = 0, \qquad (\mathrm{d}\xi = 0), \tag{24}$$

*and therefore gauge compatible.*

One can then introduce the connection associated to $\mathbb{G}$ (see [19] for an account of the historical origin of this connection in gauge theories):

**Definition 2.13** (Singer-DeWitt connection). *The connection associated to the kinetic supermetric* (23) *via* (20) *is called the Singer-DeWitt (SdW) connection, $\varpi_{SdW}$. SdW horizontal differentials will be denoted by $\mathrm{d}_{\perp} = \mathrm{d} + \varpi_{SdW}$.*

An independent argument for the derivation of $\varpi_{SdW}$ that is based on generalizing Dirac's dressing of the electron to non-Abelian theories in the presence of boundaries, is discussed in section 5.

---

[13]Notice that the dot is just a notational device and does not stand here for any time derivative: on par to the momentum $E$ here the velocity $\dot{A}$ is an independent quantity relatively to $A$.

[14]This condition implies (19) as well as the bundle-like condition mentioned in the previous footnote.

[15]A finite dimensional analogue of this condition was recently studied and generalized to more general Lie algebroids $\mathfrak{A}_{KS}$ than the action Lie algebroid featuring studied here, by Kotov and Strobl [34,39]. They named Lie algebroids satisfying such a generalized condition Killing Lie algebroids, and related their properties to the ability of "gauging" a Poisson-sigma-model with a $\mathfrak{A}_{KS}$-symmetry.

Although we have motivated the choice of the kinetic supermetric (23) by reference to the Lagrangian formulation of YM, this reference is not necessary for the analysis that will follow—and therefore we won't pursue it any further. However, we find relevant that the whole YM Lagrangian is nothing but a gauge- and Lorentz-covariant extension of the kinetic term $K = \frac{1}{2}\mathbb{G}(\dot{\mathbb{A}},\dot{\mathbb{A}})$: this simple observation explains the wealth of properties satisfied by the connection form associated through (20) to the kinetic supermetric.

An alternative, and fully explicit, characterization of the SdW connection can be given in terms of an elliptic boundary value problem:[16]

**Proposition 2.14** (SdW boundary value problem). *Over a bounded region $R$, $\partial R \neq \emptyset$, $\varpi_{SdW}$ can be equivalently defined through the following* elliptic *boundary value problem*[17]

$$\begin{cases} \mathrm{D}^2\varpi_{SdW} = \mathrm{D}^i \mathbb{d}A_i\,, & in\ R\,, \\ \mathrm{D}_s\varpi_{SdW} = \mathbb{d}A_s\,, & at\ \partial R\,, \end{cases} \tag{25}$$

*where $\mathrm{D}^2 = \mathrm{D}^i\mathrm{D}_i$ is the covariant Laplace operator, and the subscript $\bullet_s$ denotes the contraction with the outgoing unit normal $s^i$ at $\partial R$. We will call this type of elliptic boundary value problem (with this covariant-Neumann boundary condition) a* SdW boundary value problem.

*Proof.* For $0 = \mathbb{G}\left(\mathbb{X} - \varpi^\sharp_{\mathrm{SdW}}(\mathbb{X}),\xi^\sharp\right) = \int \sqrt{g}\,g^{ij}\mathrm{Tr}\left(\left(X_i - \mathrm{D}_i\varpi(\mathbb{X})\right)\mathrm{D}_j\xi\right)$ to hold for all $\xi$'s and $\mathbb{X}$ gives condition (25) after an integration by parts. See [18, 19] for a detailed derivation. $\square$

The following proposition then characterizes the curvature of the SdW connection in terms of another SdW boundary value problem:

**Proposition 2.15** (SdW curvature). *The curvature of the SdW-connection $\varpi_{SdW}$, denoted $\mathbb{F}_{SdW}$ (15), satisfies the following boundary value problem:*

$$\begin{cases} \mathrm{D}^2\mathbb{F}_{SdW} = g^{ij}[\mathbb{d}_\perp A_i \stackrel{\wedge}{,} \mathbb{d}_\perp A_j]\,, & in\ R\,, \\ \mathrm{D}_s\mathbb{F}_{SdW} = 0\,, & at\ \partial R\,. \end{cases} \tag{26}$$

*Notice that $\mathbb{F}_{SdW} \equiv 0$ in the Abelian case.*

*Proof.* In the absence of boundaries, this formula was given by Singer in [36]. In [19, eq. 5.6], the differential equation for $\mathbb{F}_{\mathrm{SdW}}$ is explicitly derived in the context without boundary. To find the boundary condition used in (26), we note that, in [19] to obtain equation 5.6, one uses equations 5.4 and 5.5. The first requires no integration by parts, contrary to the second, which yields an extra boundary term: $\oint \sqrt{h}\,\mathrm{Tr}(\xi s^i\mathrm{D}_i\mathbb{F})$. Hence, from the arbitrariness of $\xi$ at the boundary, we deduce the boundary condition of (26). $\square$

In [19], the significance of $\mathbb{F}_{\mathrm{SdW}}$ for the non-Abelian theory is extensively discussed in relation to: (*i*) the obstruction to the extension of the dressing of matter fields à la Dirac (see e.g. [40–43]) to the non-Abelian setting [44]; (*ii*) the Gribov problem [45, 46]; and (*iii*) the Vilkovisky-DeWitt geometric effective action [23, 47–51]. See also section 5 in the present article.

---

[16]This proposition is subjected to the same limitations of the definition (8). See footnote 7.

[17]This equation between 1-forms should be understood as follows. Given any $\mathbb{X} = \int X_i^\alpha \frac{\delta}{\delta A_i^\alpha}$, its contraction into (25)—recall, $\mathbb{d}A_i(\mathbb{X}) \equiv X_i$—defines the contraction $\varpi_{\mathrm{SdW}}(\mathbb{X})$ as the unique solution to:

$$\begin{cases} \mathrm{D}^2\varpi_{\mathrm{SdW}}(\mathbb{X}) = \mathrm{D}^iX_i\,, & in\ R\,, \\ \mathrm{D}_s\varpi_{\mathrm{SdW}}(\mathbb{X}) = X_s\,, & at\ \partial R\,. \end{cases}$$

Knowledge of $\varpi_{\mathrm{SdW}}(\mathbb{X})$ for an arbitrary $\mathbb{X}$ is what defines the one-form $\varpi_{\mathrm{SdW}}$.

As a consequence of the bulk and boundary properties of $\varpi_{\text{SdW}}$, SdW-horizontal modes $\mathbb{h}$, i.e. those in the kernel of $\varpi_{\text{SdW}}$ (that is $\mathring{\imath}_{\mathbb{h}}\varpi_{\text{SdW}} = 0$), do satisfy specific bulk and boundary properties:[18]

**Proposition 2.16** (SdW horizontal modes)**.** *The quasilocal* SdW-horizontal modes *of the gauge potential, $\delta A = h$, are covariantly divergenceless in the bulk and vanish when contracted with $s^i$ at the boundary $\partial R$, i.e.*

$$\mathbb{h} = \int h_i \frac{\delta}{\delta A_i} \quad \text{is SdW-horizontal iff} \quad \begin{cases} \mathrm{D}^i h_i = 0, & \text{in } R, \\ h_s = 0, & \text{at } \partial R. \end{cases} \tag{27}$$

*Physically, SdW-horizontal modes generalize to the non-Abelian setting and to the presence of boundaries the notion of transverse photon. We will therefore sometimes call them* radiative modes.

*Proof.* Contracting $\mathbb{h}$ into the boundary value problem for the SdW connection (25), and using $\mathring{\imath}_{\mathbb{h}}\varpi_{\text{SdW}} = 0$ (by definition of SdW-horizontality) and the identity $\mathring{\imath}_{\mathbb{h}}\mathbb{d}A_i = h_i$, readily gives the sought result. Alternatively, observe that the SdW horizontal modes $\mathbb{h}$ are by definition $\mathbb{G}$-orthogonal to all $\xi^{\sharp} \in V \subset \mathrm{T}\mathcal{A}$, thus $0 \overset{!}{=} \mathbb{G}(\mathbb{h}, \xi^{\sharp}) = \int \sqrt{g}\, g^{ij} \mathrm{Tr}(h_i \mathrm{D}_j \xi) = -\int \sqrt{g}\, \mathrm{Tr}(\mathrm{D}^i h_i \xi) + \oint \sqrt{h}\, \mathrm{Tr}(h_s \xi)$. The conclusion then follows from the arbitrariness of $\xi$. $\square$

Mathematically, the SdW decomposition of $\mathbb{d}A := \mathbb{d}_{\perp}A + \mathrm{D}\varpi_{\text{SdW}}$ is a non-Abelian generalization of the orthogonal Helmholtz decomposition (in the presence of boundaries) of 1-tensors in a pure-gradient part and a divergence-free part.

Notice that, consistently with our goals, the SdW boundary value problem (25) defines a connection form on $\mathcal{A}$ that is quasi-local to $R$: i.e. non-local within $R$ (it requires the inversion of a covariant Laplacian) but completely determined by the value of the fields within $R$.

For this to work, it is important that the boundary value boundary for $\varpi_{\text{SdW}}$ involves boundary conditions for $\varpi_{\text{SdW}}$, but *not* for the background gauge potential $A_i$ nor for its fluctuations $\mathbb{d}A_i$. The boundary conditions on $\varpi_{\text{SdW}}$ ensure that the connection in a region $R$, and the corresponding horizontal projections, are uniquely defined. In this way, no restriction is imposed on the gauge-variant fields $A_i$ nor on the gauge parameters $\xi$, neither in $R$ nor at $\partial R$—but restrictions naturally arise for the horizontal linearized fluctuations $h_i$.

In this regard the horizontality conditions (27) can be interpreted as a (gauge-covariant) gauge-fixing for the linearized fluctuations in a bounded region. This gauge fixing encompasses the *entire* physical content of possible linearized fluctuations over a given region; that is, although the boundary conditions might seem restrictive, a completely general linear fluctuation $\mathbb{X} \in \mathrm{T}\mathcal{A}$, with *any* other boundary condition can be generated from a $\mathbb{h}$ of the form (27) with the aid of a unique infinitesimal gauge transformation. Importantly, this is only possible because gauge freedom at the boundary is unrestricted, and therefore the $\mathbb{G}$-orthogonal projection is a complete and viable gauge fixing for the linearized fluctuations around $A \in \mathcal{A}$. In particular, the technical demand that the gauge parameters $\xi$ are fully *un*constrained at the boundary will have far-reaching repercussions, and it distinguishes our approach from others in the literature, e.g. [12–16, 52, 53] (cf. also the discussion of the "edge mode" framework in the conclusions).

Now that we have established these fundamental properties of the SdW connection and the SdW horizontal properties, we shall comment on some general properties of the SdW boundary value problem.

In footnote 7, we have anticipated (without explanation) that a connection form satisfying the projection and covariance properties can be successfully defined in the non-Abelian theory

---

[18]Of course, the properties of a SdW-horizontal mode can be deduced with no reference to $\varpi_{\text{SdW}}$, but only to $\mathbb{G}$. See the proof of the following proposition.

only on a dense subset of configurations $A \in \mathcal{A}$, and in the Abelian theory only for a slightly modified definition of the gauge group $\mathcal{G}$. Interestingly, the kernel of the SdW boundary value problem reflects these issues.

In electromagnetism (EM), the boundary value problem (25) is of Neumann type. This means that in EM the solution to this boundary value problem is not unique for constant gauge transformations are in its kernel. These are precisely the gauge transformations that have to be "removed" from $\mathcal{G}$ for the definition of a connection form to apply. It is intriguing that these gauge transformations are related to the definition of the electric charge, an observation that we will develop on in section 4.

In general, the kernel of the SdW boundary value problem is characterized by the following:

**Definition 2.17** (Reducible configurations)**.** *Configurations $A \in \mathcal{A}$ such that the equation $\mathrm{D}\chi \equiv \mathrm{d}\chi + [A, \chi] = 0$ admits nontrivial solution $\chi \in \mathrm{Lie}(\mathcal{G}) \setminus \{0\}$ are called* reducible*; all other configurations are called* ir*reducible. At a reducible configuration $A \in \mathcal{A}$, the nonvanishing solutions $\chi$ are called* reducibility parameters *or* stabilizers *of A.*

**Proposition 2.18** (Kernel of the SdW boundary value problem)**.** *At the background configuration $A \in \mathcal{A}$, the kernel of the SdW boundary value problem is given by the reducibility parameters of A.*

*Proof.* We want to show that $\mathrm{D}^2\chi = 0 = \mathrm{D}_s\chi_{|\partial R}$ if and only if (iff) $\mathrm{D}\chi = 0$ throughout $R$. One implicaftion is obvious. For the other, we observe that $0 = -\int \mathrm{Tr}(\chi \mathrm{D}^2\chi) + \oint \mathrm{Tr}(\chi \mathrm{D}_s\chi) = \mathbb{G}(\chi^\sharp, \chi^\sharp)$. From the non-degeneracy of $\mathbb{G}$, this vanishes iff $\chi^\sharp = 0$ i.e. iff $\mathrm{D}\chi \equiv 0$. $\qquad\square$

Note the prominent role the SdW boundary condition plays in this proposition: e.g. replacing it with a Dirichlet condition would leave us with a kernel which is always trivial.

Reducibility is to YM configurations as the existence of Killing vector fields is to spacetime metrics in general relativity. We will argue in section 4 that, just as for Killing vector fields, the existence of reducibility parameters is related to the existence of "global" charges in YM—the electric charge being the most basic such example.

In EM (or any Abelian theory), all configurations are reducible for $\chi = const$ (hence the universal nature of the electric charge). In non-Abelian YM, on the other hand, reducible configurations are "rare," just like spacetime metrics with Killing vector fields are rare. More precisely, reducible configurations constitute a meagre subset of $\mathcal{A}$—i.e. *ir*reducible configurations are everywhere dense in $\mathcal{A}$.

In section 4 (and appendix B), we will review the topological and geometrical properties of the set of reducible configurations within the configuration space $\mathcal{A}$. From our field-space perspective it is indeed important that these configurations are imprinted in the geometry of $\mathcal{A}$ as well as on that of the reduced field space $\mathcal{A}/\mathcal{G}$. From that discussion it will be clear why at reducible configurations *no* connection form can be defined. Until then, however, we will work in the generic subspace of $\mathcal{A}$ and neglect the existence of reducible configurations or—in the Abelian case—we will assume that $\mathcal{G}$ is appropriately replaced. In sum, *unless stated otherwise, we will henceforth consider the SdW boundary value problem as invertible.*[19]

We conclude this section with a remark that will play an important role in the following: the SdW connection for EM (for the appropriately modified $\mathcal{G}$) provides a concrete example of a connection form which is exact.

---

[19]The fact that the SdW boundary value problem is not invertible at reducible configurations means that the definition (25) of $\varpi_{\mathrm{SdW}}$ is not viable there. In fact, it turns out, the very notion of connection form fails at reducible configurations. Again, this is discussed in section 4.

**Theorem 2.19** (SdW connection in EM). *In noncompact electromagnetism (EM),*[20] *i.e. if $G = \mathbb{R}$, the SdW horizontal distribution $H_{\mathbb{G}} \equiv V^\perp$ is integrable and related to the Coulomb gauge fixing.*

*Proof.* Define the real-valued field-space function $\varsigma \in \Omega^0(\mathcal{A})$ to be the solution of the following SdW boundary value problem:

$$\begin{cases} \nabla^2 \varsigma = \nabla^i A_i\,, & \text{in } R\,, \\ \partial_s \varsigma = A_s\,, & \text{at } \partial R\,. \end{cases} \quad \text{(EM)} \tag{28}$$

Then, the SdW connection satisfying (25) can be obtained by simple field-space differentiation, that is $\varpi_{\text{SdW}} = \mathbb{d}\varsigma \in \Omega^1(\mathcal{A}, \text{Lie}(\mathcal{G}))$—notice that for this step it is crucial that the spatial differential operator $\nabla_i$ is field-independent, i.e. independent of the configuration $A \in \mathcal{A}$). By lemma 2.8 it follows that the SdW horizontal distribution is flat. More explicitly, $\mathbb{F}_{\text{SdW}} = \mathbb{d}\varpi_{\text{SdW}} = \mathbb{d}^2\varsigma = 0$. By Frobenius theorem, a flat distribution is also integrable.

For each field-independent function $\sigma : R \to \mathbb{R}$, $\mathbb{d}\sigma = 0$, define the "constant-value" hypersurface $\mathcal{H}_\sigma := \{A : \varsigma(A) = \sigma\} \subset \mathcal{A}$. Notice that the invertibility[21] of the SdW value problem means that every $A$ belongs to one and only one hypersurface $\mathcal{H}_c$, which therefore foliate $\mathcal{A}$. The SdW horizontality condition $0 = \mathring{\mathbb{i}}_{\mathbb{h}}\varpi_{\text{SdW}} = \mathbb{h}(\varsigma)$ says that $\varsigma$ is constant in the SdW horizontal directions within $\mathcal{A}$. As a consequence, the SdW horizontal directions at $A \in \mathcal{A}$ coincide with the directions tangent to the appropriate $\mathcal{H}_\sigma$ through $A$. In other words, if $A \in \mathcal{H}_\sigma$, then $(H_{\mathbb{G}})_A = T_A \mathcal{H}_\sigma$. From this, $H_{\mathbb{G}} = T(\bigcup_\sigma \mathcal{H}_\sigma)$. This also shows that the foliation $\mathcal{H} = \bigcup_\sigma \mathcal{H}_\sigma$ is transverse to the vertical foliation $\mathcal{F}$.

Finally, consider the vanishing parameter $\sigma \equiv 0$. Then, setting $\varsigma = 0$ in (28) shows that the configurations lying on $\mathcal{H}_0$ satisfy the Coulomb gauge condition $\nabla^i A_i = 0$, completed—if $\partial R \neq \emptyset$—by the boundary condition $A_s = 0$. This means that $\mathcal{H}_0$ is the section of $\mathcal{A}$ corresponding to the Coulomb fixing.[22] More generally, the "constant-value" hypersurfaces $\mathcal{H}_\sigma$ generalize Coulomb gauge according to the spatial properties of $\sigma : R \to \mathbb{R}$. $\qquad\square$

Notice that in the Dirac-Bergmann formalism for constrained systems, $\varsigma = 0$ is the second class constraint associated to the Coulomb gauge fixing. However, not all second class constraints (gauge-fixings) define a connection form, since they must satisfy the restrictive covariance condition (17). E.g. even a change in the boundary condition of (28) would jeopardize that covariance property.

## 2.3 Horizontal splitting in phase space

In the last section we have introduced configuration space. In this section we will introduce *phase space* and *matter fields*. Most constructions are immediate extensions of those performed in the previous section and will therefore be only sketched.

The YM phase space is defined as the cotangent bundle of the configuration space $\mathcal{A}$, and its elements are

$$(A, E) \in T^*\mathcal{A}\,. \tag{29}$$

---

[20]As mentioned above, the SdW connection for $G = U(1)$ or $\mathbb{R}$ is not invertible. For the time being we will work formally: in section 4 we will show how to get around this issue by modding-out constant gauge transformations. The present conclusions will not be altered by the more rigorous treatment.

[21]See the next footnote.

[22] Seemingly, any spatially constant parameter $\xi = \xi_o \in \text{Lie}(G)$ would do. This is because constant gauge transformations constitute precisely the stabilizer gauge transformations in the kernel of the Abelian SdW boundary problem (proposition 2.18). However, following the discussion above, in order to have a well-defined $\varpi_{\text{SdW}}$ we have (implicitly) modified $\mathcal{G}$ by modding-out the stabilizer gauge transformations—i.e. the constant gauge transformations. Hence, from this perspective we have identified all $\xi = \xi_o$ with $\xi = 0$ and therefore the Coulomb gauge fixing as defined in the text indeed corresponds to one section of $\mathcal{A}$ which crosses all fibres once and only once. See section 4.4 for a detailed construction of the appropriately modified gauge group $\mathcal{G}$ for electromagnetism.

The coordinates $(A, E)$ have been chosen so that the tautological 1-form on $T^*\mathcal{A}$ reads

$$\theta_{\text{YM}} = \int \sqrt{g}\, \text{Tr}\big(E^i \mathbb{d}A_i\big) \in \Omega^1(T^*\mathcal{A}), \tag{30}$$

that is so that—interpreting $\theta_{\text{YM}}$ as the *off-shell*[23] *symplectic potential* of Yang-Mills theory—$E^i(x) = E^{i\alpha}(x)\tau_\alpha$ is the $\text{Lie}(\mathcal{G})$-valued *electric field*.

As customary in second-order Lagrangian theories, the canonical momentum (a one-form) is related to the configuration velocity (a one-vector) via the kinetic supermetric. This is most succinctly expressed in terms $\theta_{\text{YM}}$:

$$\theta_{\text{YM}} = \int \sqrt{g}\, \text{Tr}(E^i \mathbb{d}A_i) = \mathbb{G}(\dot{\mathbb{A}}). \tag{31}$$

This is nothing else than the YM analogue of the usual Legendre transform relating momenta and velocities in particle mechanics, $p_I \mathrm{d}q^I = g_{IJ}\dot{q}^I \mathrm{d}q^J$.

Since under a gauge transformation $E$ transforms in the adjoint representation, the configuration-space gauge symmetry is lifted to phase space as follows:

$$\xi^\sharp_{(A,E)} = \int (D_i\xi)^\alpha(x)\frac{\delta}{\delta A_i^\alpha(x)} + [E^i, \xi]^\alpha(x)\frac{\delta}{\delta E^{i\alpha}(x)} \in T_{(A,E)}T^*\mathcal{A}. \tag{32}$$

As in the previous section, vectors fields of this form are called *vertical*. Through their span they locally define an integral distribution $\tilde{V} \subset TT^*\mathcal{A}$, and thus a foliation $\tilde{\mathcal{F}}$ of $T^*\mathcal{A}$, which identifies the pure-gauge directions in phase space (we temporarily introduce tildes to distinguish these spaces from their configuration space analogues).

The inclusion of matter can be done in similar fashion. For definiteness, we consider complex Dirac fermions, $\psi$, valued in the fundamental representation $W$ of the gauge group $G = \text{SU}(N)$,

$$\psi^{B,b} \in \Psi = \mathcal{C}^\infty(\Sigma, \mathbb{C}^4 \otimes W). \tag{33}$$

The conjugate momenta, $\overline{\psi}$, thus live in[24] $\overline{\Psi} = \mathcal{C}^\infty(\Sigma, \mathbb{C}^4_* \otimes W^\dagger)$.

Under the action of a gauge transformation $g \in \mathcal{G}$, $\psi$ and $\overline{\psi}$ transform as

$$\psi \mapsto g^{-1}\psi, \quad \text{and} \quad \overline{\psi} \mapsto \overline{\psi}g. \tag{34}$$

Thus, the $(\psi, \overline{\psi})$-components of $\xi^\sharp$ read

$$\xi^\sharp_{|\psi,\overline{\psi}} = \int (-\xi\psi)^{B,b}(x)\frac{\delta}{\delta\psi^{B,b}(x)} + (\overline{\psi}\xi)^{B,b}(x)\frac{\delta}{\delta\overline{\psi}^{B,b}(x)}, \tag{35}$$

where $(\xi\psi)^{B,b}(x) = \xi^\alpha(x)(\tau_\alpha)^b{}_{b'}\psi^{B,b'}(x)$, with $(\tau_\alpha)^{b'}{}_b$ an anti-Hermitian generator of $G$ in the fundamental representation $W$.

The charged fermions carry a $\text{Lie}(G)$-current density

$$J^\mu = (\rho, J^i), \quad \text{with} \quad J^\mu_\alpha = \overline{\psi}\gamma^\mu\tau_\alpha\psi, \tag{36}$$

where $(\gamma^\mu)^{B'}{}_B$ are the Dirac matrices.[25]

---

[23]"Off shell" refers to the Gauss constraint, see below.

[24]In a Lagrangian setting, $\overline{\psi} = i\psi^\dagger\gamma^0$. See the next footnote for details on $\gamma^0$. Here $\mathbb{C}^4_*$ indicates that the action of the Lorentz group on $\overline{\Psi}$ differs from that over $\Psi$ [54]. The details won't be needed.

[25]For a metric $g_{ij}$ on $\Sigma$, the commutator is $\{\gamma^\mu, \gamma^\nu\} = 2g^{\mu\nu} = 2\text{diag}(-1, g^{ij})$, i.e. $\gamma^\mu := e^\mu_I\gamma^I$ for $\gamma^I$ the flat-space Dirac matrices and $e^\mu_I$ a local inertial frame, $g_{\mu\nu}e^\mu_I e^\nu_J = \eta_{IJ}$ (see also footnote 1). We adopt the following conventions for the $\gamma^I$ [54]: $\gamma^0 = -i\begin{pmatrix} 0 & 1 \\ 1 & 0 \end{pmatrix}$, $\gamma^j = -i\begin{pmatrix} 0 & \sigma^j \\ -\sigma^j & 0 \end{pmatrix}$ with $\sigma^j$ the Hermitian Pauli matrices.

It is convenient to introduce the following notation for the *total phase space*,

$$\Phi = \mathrm{T}^*\mathcal{A} \times (\overline{\Psi} \times \Psi). \tag{37}$$

Then the (complex) contribution of the Dirac fermions to the *total off-shell symplectic potential*

$$\theta = \theta_{\mathrm{YM}} + \theta_{\mathrm{Dirac}} \in \Omega^1(\Phi) \tag{38}$$

is:

$$\theta_{\mathrm{Dirac}} = -\int \sqrt{g}\,\overline{\psi}\gamma^0 \mathbb{d}\psi \in \Omega^1(\overline{\Psi} \times \Psi). \tag{39}$$

As on $\mathcal{A}$, the total action of gauge transformations on $\Phi$, $\xi^\sharp = \xi^\sharp_{A,E} + \xi^\sharp_{\psi,\overline{\psi}}$, can be promoted to a field-dependent one. That is, from now on

$$\xi \in \Omega^0(\Phi, \mathrm{Lie}(\mathcal{G})), \tag{40}$$

with the isomorphism (6) extended to

$$[\![\xi^\sharp, \eta^\sharp]\!]_{\mathrm{T}\Phi} = \left([\xi,\eta] + \xi^\sharp(\eta) - \eta^\sharp(\xi)\right)^\sharp. \tag{41}$$

Given a connection form on $\mathcal{A}$, a connection form can be introduced on $\Phi$ by pullback:

**Proposition 2.20.** *Denoting by $\pi : \Phi \to \mathcal{A}$ the canonical projection from the full phase space to the gauge-potential configuration space $\mathcal{A}$, the pullback $\pi^*\varpi$ of the $\mathcal{G}$-compatible connection form $\varpi$ onto $\Phi$ defines a connection form on $\Phi$—i.e. it defines a $\mathrm{Lie}(\mathcal{G})$-valued 1-form on $\Phi$ that satisfies the corresponding projection and covariance properties. In particular, $\pi^*\varpi$ defines a horizontal distribution $\tilde{H} := \ker(\pi^*\varpi) \subset \mathrm{T}\Phi$ transverse to the vertical distribution spanned by the $\xi^\sharp$, also denoted $\tilde{V} \subset \mathrm{T}\Phi$; i.e. $\tilde{H} \oplus \tilde{V} = \mathrm{T}\Phi$.*

*Proof.* This follows directly from the fact that $A$, $E$, $\psi$, and $\overline{\psi}$ transform in concert under gauge transformations, together with the fact that $A$ necessarily changes under a gauge transformation (recall that we are here considering irreducible configurations only). $\qquad\square$

There is therefore little use in having different notations for $\varpi$ and its pullback on phase space; we will henceforth denote $\pi^*\varpi$ simply by $\varpi$, and $(\tilde{H}, \tilde{V})$ by $(H, V)$. (For an alternative, of more limited use, to this pullback construction from $\mathcal{A}$ to $\Phi$, see the so-called Higgs connection introduced in [19].)

We can now turn to the computation of horizontal differentials in $\Phi$. Following the definitions given in the previous sections, as well as equation (12) for the horizontal differential of horizontal and equivariant forms, it is straightforward to prove that

**Proposition 2.21.** *The single and double horizontal differentials of $A$, $E$, $\psi$ and $\overline{\psi}$ are respectively given by*

$$\begin{cases} \mathbb{d}_H A = \mathbb{d}A - \mathrm{D}\varpi \\ \mathbb{d}_H E = \mathbb{d}E - [E,\varpi] \end{cases} \quad and \quad \begin{cases} \mathbb{d}_H \psi = \mathbb{d}\psi + \varpi\psi \\ \mathbb{d}_H \overline{\psi} = \mathbb{d}\overline{\psi} - \overline{\psi}\varpi \end{cases}, \tag{42}$$

and[26]

$$\begin{cases} \mathbb{d}_H^2 A = -\mathrm{D}\mathbb{F} \\ \mathbb{d}_H^2 E = -[E,\mathbb{F}] \end{cases} \quad and \quad \begin{cases} \mathbb{d}_H^2 \psi = \mathbb{F}\psi \\ \mathbb{d}_H^2 \overline{\psi} = -\overline{\psi}\mathbb{F} \end{cases}. \tag{43}$$

If $\varpi$ is flat, then the horizontal differentials assume a particular meaning in terms of dressed field [40, 42, 55]. This is spelled out in the following definition and proposition:

---

[26]In general, for an horizontal and equivariant form $\lambda$, $\mathbb{d}_H^2 \lambda^a = -R(\mathbb{F})^a{}_b \wedge \lambda^b$. See (12).

**Definition 2.22** (Dressed fields)**.** *Assume the existence of a covariant field-space function* $h : \Phi \to \mathcal{G}$ *such that* $R_g^* h = hg$ *for all* $g \in \mathcal{G}$, *then the following composite fields, called the* dressed fields, *can be defined:*

$$
\begin{cases} \widehat{A} = hAh^{-1} + h\mathrm{d}h^{-1} \\ \widehat{E} = hEh^{-1} \end{cases} \quad and \quad \begin{cases} \widehat{\psi} = h\,\psi \\ \widehat{\overline{\psi}} = h^{-1}\overline{\psi} \end{cases} \quad (\varpi = h^{-1}\mathrm{d}h). \tag{44}
$$

*In these formulas h is called the* dressing factor.

Then it is straightforward to check the following:

**Proposition 2.23.** *Dressed fields can be defined if and only if a flat connection* $\varpi = h^{-1}\mathrm{d}h$ *exists. Moreover, the dressed fields are* gauge invariant *and their differential is related to the horizontal differential through the following:*

$$
\begin{cases} \mathrm{d}\widehat{A} = h(\mathrm{d}_H A)h^{-1} \\ \mathrm{d}\widehat{E} = h(\mathrm{d}_H E)h^{-1} \end{cases} \quad and \quad \begin{cases} \mathrm{d}\widehat{\psi} = h\,\mathrm{d}_H \psi \\ \mathrm{d}\widehat{\overline{\psi}} = h^{-1}\mathrm{d}_H \overline{\psi} \end{cases} \quad (\varpi = h^{-1}\mathrm{d}h). \tag{45}
$$

Therefore whenever the connection is not flat and the dressing construction is not available, one can see the horizontal differential as the only viable generalization of the dressing construction. This provides a physical intuition on the meaning of $\varpi$ and will be further discussed in section 5.

As shown by the following theorem, the dressed-field construction *is* available, and indeed quite familiar, in electromagnetism:

**Theorem 2.24** (Coulomb potential and Dirac's dressed electron)**.** *In EM, where* $\varpi_{SdW} = \mathrm{d}\varsigma$ *is exact, the SdW dressing of the field can be defined. Moreover, if* $R = \mathbb{R}^3$ *and we assume standard rapid-fall-off boundary conditions for the fields at infinity, the SdW dressed gauge potential* $\widehat{A}$ *coincides with the gauge potential in Coulomb gauge, whereas the dressed electron* $\widehat{\psi}$ *coincides with Dirac's dressed electron (cf. section 5 for details on Dirac's dressed electron).*

*Proof.* In EM the SdW connection is flat $\varpi_{\mathrm{SdW}} = \mathrm{d}\varsigma$ and $h = e^{\varsigma}$ defines the SdW dressing factor. Then the expression for the dressed fields simplifies to

$$
\begin{cases} \widehat{A} = A - \mathrm{d}\varsigma \\ \widehat{E} = E \end{cases} \quad and \quad \begin{cases} \widehat{\psi} = e^{\varsigma}\psi \\ \widehat{\overline{\psi}} = e^{-\varsigma}\overline{\psi} \end{cases} \quad (\varpi = \mathrm{d}\varsigma). \tag{46}
$$

The fact that in $R = \mathbb{R}^3$, $\widehat{A}$ is the gauge potential in Coulomb gauge and $\widehat{\psi}$ is Dirac's dressed electron are both direct consequences of (28)—notice that in EM, the electric field $E$ is already gauge invariant and therefore its dressing is trivial. $\qquad\square$

A thorough discussion (with references) of Dirac's dressing is postponed to section 5, where we will also discuss—from a field-space perspective—a possible generalization of dressed fields to the non-flat setting and in particular to the non-Abelian SdW case.

In the above example the dressing factor is spatially nonlocal. Conversely, if the dressing factor can be chosen to be space(time) local, then the passage to dressed fields is just a local field redefinition that completely "reabsorbs" the gauge symmetry. In [55] this circumstance is interpreted—and we agree—as meaning that a gauge symmetry that can be "neutralized" in this way is non-substantial. This is the case e.g. when the gauge symmetry is introduced through a so-called Stückelberg trick, but it is also the case for the Lorentz gauge symmetry in tetrad gravity (here the dressing factor is given by the inverse tetrad) and, with certain subtleties [19, Sect. 9], in the presence of spontaneous symmetry breaking (here the dressed fields are the fields expressed in unitary gauge).

# 3 Horizontal splittings and symplectic geometry

This section is dedicated to the study of the symplectic structure of YM theory in the presence of boundaries. In particular, we will study the horizontal/vertical split of the symplectic structure induced by the horizontal/vertical decomposition of the (co)tangent bundle of the total phase space introduced in the previous section. This study was initiated in [17, 19] and is here pushed (much) further.

Of course, many of the propositions presented in this section are (a rephrasing of) well known facts.

We should point out that ultimately the choice of a $\varpi$—including the SdW one, which in certain respects is a more convenient choice—is entirely fiducial. As described in [20], on-shell of the Gauss constraint, one *can* write the physical, i.e. reduced, symplectic form *independently* of a choice of $\varpi$. However, the explicit description of the physical degrees of freedom *will* involve a choice of connection $\varpi$. This was to be expected from the standard symplectic duality between the gauge constraints and gauge-fixings. It should also be noticed that the ability of writing down the $\varpi$-independent reduced symplectic structure relies on the introduction of superselection sectors and a *canonical* completion of the symplectic structure; this completion does not add any new dof. We will briefly review these results in Section 3.4, and refer to [20] for details.

## 3.1 Horizontal/vertical split of the symplectic structure

Given the total symplectic potential, $\theta$ and the horizontal/vertical split of $T\Phi$, we introduce a horizontal/vertical split of $\theta$ itself:

**Definition 3.1** (Horizontal/vertical split of $\theta$). *The* horizontal/vertical split *of the off-shell symplectic potential* $\theta = \theta_{YM} + \theta_{Dirac}$ *with respect to a connection form* $\varpi$ *is defined as:*

$$\theta = \theta^H + \theta^V, \quad \text{where} \quad \begin{cases} \theta^H := \int_R \sqrt{g}\,\mathrm{Tr}(E^i \mathbb{d}_H A_i) - \int_R \sqrt{g}\,\overline{\psi}\gamma^0 \mathbb{d}_H \psi \\ \theta^V := \int_R \sqrt{g}\,\mathrm{Tr}(E^i D_i \varpi) + \int_R \sqrt{g}\,\overline{\psi}\gamma^0 \varpi \psi \end{cases}, \tag{47}$$

$\theta^H$ *(resp. $\theta^V$) is said the* horizontal *(resp. vertical) off-shell symplectic potential.*

By construction $\theta^H(\xi^\sharp) := \mathbb{i}_{\xi^\sharp}\theta^H \equiv 0$ for all $\xi$, and $\theta^V(\mathbb{h}) \equiv 0$ for all $\mathbb{h} \in H \subset T\Phi$, hence the horizontal/vertical nomenclature.

Although in the above formulas we have explicitly decomposed $\mathbb{d}A$ into pure-gauge and horizontal modes, we haven't yet decomposed the different modes of the electric field.

**Definition 3.2** (Radiative/Coulombic decomposition). *Given a connection form $\varpi$, define the following functional decomposition of the electric field into* radiative *and* Coulombic *components*

$$E = E_{rad} + E_{Coul}, \tag{48}$$

*through the cotangent dual of the decomposition of $\mathbb{X} \in T\mathcal{A}$ into its horizontal and vertical parts, $E_{rad}$ being dual to horizontal vectors and $E_{Coul}$ to vertical ones.*

In other words, the radiative/Coulombic decomposition is defined by demanding that $\int \sqrt{g}\,\mathrm{Tr}(E_{rad}^i D_i \xi) \equiv 0 \equiv \int \sqrt{g}\,\mathrm{Tr}(E_{Coul}^i h_i)$, for all $\xi \in \mathrm{Lie}(\mathcal{G})$ and all horizontal vectors $\mathbb{h} = \int h\frac{\delta}{\delta A}$. Therefore, by definition, $E_{rad}$ drops from $\theta^V$ and is therefore the component of the electric field conjugate to $\mathbb{d}_H A$; and conversely, $E_{Coul}$ drops from $\theta^H$ and is (loosely speaking) the component of $E$ conjugate to $\varpi$.

In more detail: we use the cotangent dual to the horizontal/vertical decomposition of vectors $\mathbb{X} = \mathbb{h} + \xi^{\sharp} \in \mathrm{T}\mathcal{A}$ to decompose the *covector* $\theta = \int \sqrt{g}\,\mathrm{Tr}(E^i \mathbb{d}A_i) \in \mathrm{T}^*\mathcal{A}$ into $\theta^H$ and $\theta^V$—so that by definition $\theta^H(\xi^{\sharp}) \equiv 0 \equiv \theta^V(\mathbb{h})$. The decomposition of $E$ into $E_{\mathrm{rad}}$ and $E_{\mathrm{Coul}}$ is then a rewriting of the decomposition of $\theta$ in terms of its coordinate components.[27] Moreover, from the gauge-covariance of the whole construction it follows that $E$, $E_{\mathrm{rad}}$ and $E_{\mathrm{Coul}}$ all transform in the adjoint representation and are therefore equally gauge variant. Indeed, since $\theta$ is a *co*vector rather than a vector, what we call its horizontal/vertical decomposition has *nothing* to do with a split of $E$ into its pure-gauge and "physical" components, as it was for $\mathbb{X} = \mathbb{h} + \xi^{\sharp}$. This point is most evident in electromagnetism, where $E$, $E_{\mathrm{rad}}$, and $E_{\mathrm{Coul}}$ are all gauge *in*variant and equally "physical." The only place in which the "pure gauge" part of $E$ (or of $E_{\mathrm{rad}}$, or of $E_{\mathrm{Coul}}$) is distinguished through a geometric construction, is when we build a horizontal variation of $E$ or, dually, the horizontal differential $\mathbb{d}_H E$ (or $\mathbb{d}_H E_{\mathrm{rad}}$, or $\mathbb{d}_H E_{\mathrm{Coul}}$).

Regarding notation, we will see that $E_{\mathrm{rad}}$ is a generalization of the transverse electric field of a photon to a finite region, thus the labeling "rad" which stands for *radiative*. Conversely, $E_{\mathrm{Coul}}$ will be tasked with solving the Gauss constraint within $R$, and for this reason it is labeled "Coul" which stands for *Coulombic*.

Another convenient way to understand the above definition uses the supermetric $\mathbb{G}$ to dualize the electric field by introducing an associated field space vector. Despite the fact that to perform the dualization we will use $\mathbb{G}$, the following construction holds for *any* choice of $\varpi$, not only the SdW one.

Following the hint of (31), we can use $\mathbb{G}$ to convert the definition of $E$ as a cotangent vector to a tangent one: define $\mathbb{E} \in \mathrm{T}\mathcal{A}$ to be the field-space vector such that[28]

$$\theta = \mathbb{G}(\mathbb{E}) \equiv \mathbb{G}(\mathbb{E}, \mathbb{d}A). \tag{49}$$

More explicitly,

$$\mathbb{E} := \int \sqrt{g}\, g_{ij} E^i \frac{\delta}{\delta A_j} \in \mathrm{T}\mathcal{A}. \tag{50}$$

With this notation, the radiative/Coulombic split of $E$ can be seen to be defined by the following orthogonality relations:

$$\begin{cases} \mathbb{G}(\mathbb{E}_{\mathrm{rad}}\, \xi^{\sharp}) \equiv 0, & \text{for all } \xi, \\ \mathbb{G}(\mathbb{E}_{\mathrm{Coul}}\, \widehat{H}(\mathbb{X})) \equiv 0, & \text{for all } \mathbb{X}, \end{cases} \tag{51}$$

where we recall that $\widehat{H}(\mathbb{X}) := \mathbb{X} - \varpi(\mathbb{X})^{\sharp}$ is the $\varpi$-horizontal projection in $\mathrm{T}\mathcal{A}$. These equations are of course just a rewriting of the dual nature of the decomposition of $E$ relatively to that of $\mathbb{X} \in \mathrm{T}\mathcal{A}$. See figure 4.

From the first of these equations we readily see that:

**Proposition 3.3** (Radiative electric field)**.** *The radiative component of the electric field is (covariant-)divergence-free and fluxless, i.e.*

$$\begin{cases} \mathrm{D}_i E^i_{rad} = 0, & \text{in } R, \\ s_i E^i_{rad} = 0, & \text{at } \partial R. \end{cases} \tag{52}$$

*Proof.* The proof follows from (51) is formally identical to the proof of proposition 2.16 on the properties of the SdW-horizontal modes of the gauge potential. $\square$

---

[27] Note that $\theta^H = \int \sqrt{g}\,\mathrm{Tr}(E_{\mathrm{rad}} \mathbb{d}_H A) = \int \sqrt{g}\,\mathrm{Tr}(E \mathbb{d}_H A) = \int \sqrt{g}\,\mathrm{Tr}(E_{\mathrm{rad}} \mathbb{d}A)$, and similarly for $\theta^V$.

[28] The last expression of (49) has been introduced for notational convenience, even if geometrically imprecise. But the meaning is intuitively clear: $\theta(\mathbb{X}) \equiv \mathbb{i}_{\mathbb{X}}\mathbb{G}(\mathbb{E}) \equiv \mathbb{G}(\mathbb{E}, \mathbb{X})$ for any $\mathbb{X} \in \mathrm{T}\mathcal{A}$. We also notice that $\mathbb{i}_{\mathbb{X}}(\mathbb{d}_H A) = \widehat{H}(\mathbb{X})$ and $(\mathbb{i}_{\mathbb{X}}\varpi)^{\sharp} = \widehat{V}(\mathbb{X})$ where $\widehat{H}$ and $\widehat{V}$ are the horizontal and vertical projections respectively.

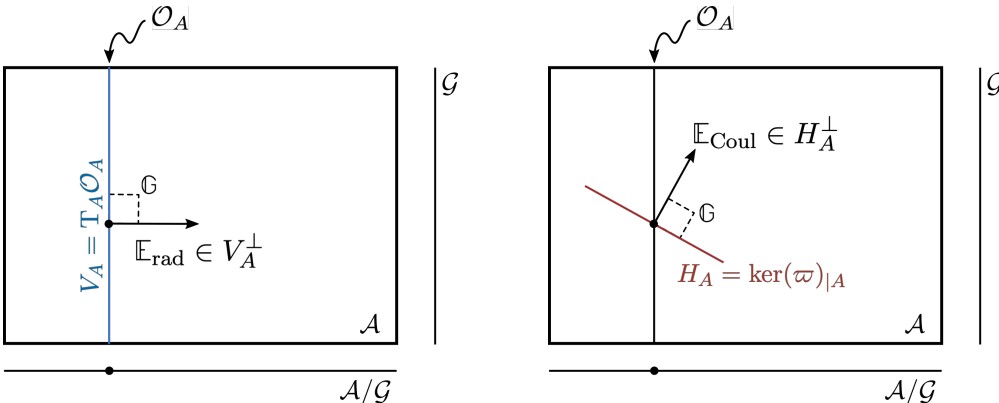

Figure 4: A graphical representation in $T\mathcal{A}$ of $\mathbb{E}_{\text{rad}}$ and $\mathbb{E}_{\text{Coul}}$ as vectors in the $\mathbb{G}$-orthogonal complements of $V_A$ and $H_A$, respectively. Notice that only with the SdW choice $\varpi = \varpi_{\text{SdW}}$, one has $H_A = V_A^\perp$ and therefore $\mathbb{E}_{\text{rad}} \in H_A^{\text{SdW}}$ and $\mathbb{E}_{\text{Coul}} \in V_A$; that is, only with this choice do the pictures on the right and left align—see section 3.2.

Equation (52) reduces the number of local dof of $E_{\text{rad}}$ with respect to $E$ by one (times $\dim(G)$), as required for $E_{\text{rad}}$ to be conjugate to $\mathbb{d}_H A$. The remaining degree of freedom is then encoded in $E_{\text{Coul}}$. As exemplified by the second equation of (51), the functional properties of $E_{\text{Coul}}$, contrary to those of $E_{\text{rad}}$, are not universal i.e. they depend on the choice of horizontal distribution, that is of $\varpi$.

We will see shortly that the vertical symplectic potential $\theta^V$ is tightly related to the Gauss constraint. It should therefore not come as a surprise that $E_{\text{rad}}$ is completely absent from the Gauss constraint, due to its being divergence-free in the bulk. Moreover, the boundary condition $s_i E_{\text{rad}}^i = 0$ in (52), which expresses the fluxless property of $E_{\text{rad}}$, already suggests that the Gauss constraint, in a bounded region, should be complemented by a boundary condition involving the electric flux $E_s$.

In section 4 (and especially 4.2) we will argue how, contrary to $\rho$, the electric flux is *not* determined by the field content of the region $R$. This means, in particular, that charged matter can be introduced into $R$ without[29] modifying $E_s$. Following this argument, as well as the analysis of the Gauss constraint performed in [20], we are led to consider the value $f$ of $E_s$ as an *external* datum which is not on par with $\rho$. Rather, this external datum defines *super-selection sectors* of the theory as restricted to $R$.

Hence, *given* a functional connection $\varpi$—which allows us to define the radiative/coulombic split of $E$,—and a flux $f$, we introduce the following version of the Gauss constraint (see [20] for details):

$$\mathsf{G}_f : \quad \begin{cases} \mathsf{G} := D_i E_{\text{Coul}}^i - \rho \approx 0, & \text{in } R, \\ \mathsf{G}_f^\partial := s_i E_{\text{Coul}}^i - f \approx 0, & \text{at } \partial R. \end{cases} \tag{53}$$

This equation has then a unique solution, as discussed in section 3.2.

The above-mentioned relation between $\theta^V$ and the Gauss constraint (paragraphs following (52)) becomes manifest through an integration by parts:

$$\theta^V = -\int \sqrt{g}\, \text{Tr}(\mathsf{G}\,\varpi) + \oint \sqrt{h}\, \text{Tr}(E_s\,\varpi) \approx \oint \sqrt{h}\, \text{Tr}(f\,\varpi), \tag{54}$$

---

[29]There are caveats to these statements in Abelian theories and more generally at reducible configurations, where a finite number of modes of $E_s$ over $\partial R$ is related to as many integrals of $\rho$ over $R$. E.g. in electromagnetism $\int \sqrt{g}\, \rho_{\text{EM}} = \oint \sqrt{h} f$. We refer to section 4 for a discussion.

where we have introduced $h_{ab} = (\iota^*_{\partial R} g)_{ab}$, the induced metric on $\partial R$, and the square-root of its determinant $\sqrt{h}$. This shows that the vertical symplectic potential is, on shell of the Gauss constraint, a pure boundary term.

We are finally ready to introduce the split of the symplectic *form* and thus state our theorem on the horizontal/vertical split of the symplectic structure in the presence of boundaries.

Recall that from the off-shell symplectic potential $\theta = \theta_{YM} + \theta_{Dirac}$, one builds the *off-shell symplectic 2-form* by differentiation:

$$\Omega = \Omega_{YM} + \Omega_{Dirac} = \mathbb{d}\theta_{YM} + \mathbb{d}\theta_{Dirac} = \mathbb{d}\theta , \qquad (55)$$

i.e.

$$\Omega = \int_R \sqrt{g}\,\mathrm{Tr}(\mathbb{d}E^i \curlywedge \mathbb{d}A_i) - \int_R \sqrt{g}\,\mathbb{d}\overline{\psi} \curlywedge \gamma^0 \mathbb{d}\psi . \qquad (56)$$

**Definition 3.4** (Horizontal/vertical split of $\Omega$). *The* horizontal/vertical split *of the off-shell symplectic 2-form $\Omega = \Omega_{YM} + \Omega_{Dirac}$ is defined as:*

$$\Omega^H = \Omega^H_{YM} + \Omega^H_{Dirac} := \mathbb{d}\theta^H_{YM} + \mathbb{d}\theta^H_{Dirac} = \mathbb{d}\theta^H \qquad and \qquad \Omega^\partial := \mathbb{d}\theta^V . \qquad (57)$$

$\Omega^H$ *(resp. $\Omega^\partial$) is said the* horizontal *(resp.* boundary*) symplectic form.*

Notice that, when referring to $\Omega^H$ and $\Omega^\partial$ the use of the adjective "symplectic" is technically incorrect, since they have degenerate directions in $T\Phi$. This fact can be emphasized in the case of $\Omega^H$ by rather using the term *pre*-symplectic.

We also warn the reader that when referred to $\Omega$, the nomenclature "horizontal/vertical split" should not be misinterpreted: the horizontal/vertical decomposition of $T\mathcal{A}$ is at the basis of the split—hence its name,—but: $\Omega^\partial$ fails to be purely vertical *and* $\Omega^H$ sometimes fails to be the *entirety* of the horizontal components present in $\Omega$. These points are clarified by the following theorem.

**Theorem 3.5** (Horizontal/vertical split of the symplectic structure). *The horizontal/vertical split of the off-shell symplectic potential and off-shell symplectic 2-form read, respectively:*

$$\theta = \theta^H + \theta^V , \quad where \quad \begin{cases} \theta^H = \displaystyle\int \sqrt{g}\,\mathrm{Tr}(E^i_{rad}\mathbb{d}_H A_i) - \int \sqrt{g}\,\overline{\psi}\gamma^0 \mathbb{d}_H \psi \\[2mm] \theta^V = \displaystyle\int \sqrt{g}\,\mathrm{Tr}(-\mathsf{G}\,\varpi) + \oint \sqrt{h}\,\mathrm{Tr}(E^s_{Coul}\varpi) , \\[2mm] \approx \displaystyle\oint \sqrt{h}\,\mathrm{Tr}(f\,\varpi) \end{cases} \qquad (58)$$

*and*

$$\Omega = \Omega^H + \Omega^\partial , \quad where \quad \begin{cases} \Omega^H = \displaystyle\int \sqrt{g}\,\mathrm{Tr}(\mathbb{d}_H E^i_{rad} \curlywedge \mathbb{d}_H A_j) - \int \sqrt{g}\left(\mathbb{d}_H \overline{\psi} \curlywedge \gamma^0 \mathbb{d}_H \psi - \mathrm{Tr}(\rho\,\mathbb{F})\right) \\[2mm] \Omega^\partial = -\displaystyle\int \sqrt{g}\,\mathrm{Tr}\left(\tfrac{1}{2}\mathsf{G}[\varpi \stackrel{\curlywedge}{,} \varpi] + \mathbb{d}_H \mathsf{G} \curlywedge \varpi + \mathsf{G}\mathbb{F}\right) \\[2mm] \qquad + \displaystyle\oint \sqrt{h}\,\mathrm{Tr}\left(\tfrac{1}{2}E_s[\varpi \stackrel{\curlywedge}{,} \varpi] + \mathbb{d}_H E_s \curlywedge \varpi + E_s\mathbb{F}\right) \\[2mm] \approx \displaystyle\oint \sqrt{h}\,\mathrm{Tr}\left(\tfrac{1}{2}f[\varpi \stackrel{\curlywedge}{,} \varpi] + \mathbb{d}_H f \curlywedge \varpi + f\,\mathbb{F}\right) \end{cases}$$

$$\qquad (59)$$

*(In Appendix A, we provide a quick bridge to a more common notation, analogous to that used e.g. by Ashtekar and Streubel [56] or Wald and Lee [57].)*

*Proof.* The proof (58) was given in equations (47) and (54). The proof of (59) follows from a straightforward albeit tedious calculation which employs the following relations: $\Omega^H := \mathbb{d}_H \theta^H$ (57), $\mathbb{d}_H^2 A = -D\mathbb{F}$ and $\mathbb{d}_H^2 \psi = \mathbb{F}\psi$ (43), as well as (52); (16) is also needed to compute $\Omega^\partial = \mathbb{d}\theta^V$. □

*In the absence of boundaries*, we come to the following conclusion:

**Corollary 3.6.** *In the* absence *of boundaries* $\partial R = \emptyset$ *and on-shell of the Gauss constraint* $\mathsf{G} \approx 0$, *the total symplectic form equals the horizontal one,* $\Omega \approx \Omega^H$. *Therefore, in the absence of boundary and on shell of the Gauss constraint,* $\Omega^H$ *is independent of the choice of functional connection* $\varpi$ *used to build it.*

*In the presence of boundaries*, on the other hand, even on-shell of the Gauss constraint, the pure-gauge and Coulombic dof *fail to fully drop* from the symplectic structure: both $f$ and the boundary value of $\varpi$ survive[30] in $\Omega^\partial$, i.e. $\Omega - \Omega^H = \Omega^\partial \not\approx 0$. We stress that the boundary value of $\varpi$ is in general a nonlocal function of the fields within $R$, as in the SdW case.

Notice that, contrary to $\theta^V$, $\Omega^\partial$ is *not* purely-vertical: it features one pure-vertical contribution (the first one in (59)), one mixed horizontal-vertical contribution (the second one), and—if $\mathbb{F} \neq 0$—even a *purely horizontal* contribution. This has the following consequences:

**Corollary 3.2** (Implications of $\partial R \neq \emptyset$ on $\Omega^H$)**.**
*If $\partial R \neq \emptyset$:*

(i) *The horizontal symplectic form* $\Omega^H$ *coincides with the horizontal projection of* $\Omega$, *i.e. with* $\Omega(\widehat{H}(\cdot), \widehat{H}(\cdot))$, *if and only if* $\varpi$ *is flat—that is if and only if* $\mathbb{F} = 0$;

(ii) *The horizontal projection* $\Omega(\widehat{H}(\cdot), \widehat{H}(\cdot))$ *is not closed unless* $\mathbb{F} = 0$;

(iii) *The horizontal symplectic form* $\Omega^H$ *depends on the choice of functional connection* $\varpi$ *used to build it.*

Ultimately, this hints at a deeper fact: in the presence of boundaries, $\Omega^H$ does not provide, by itself, a canonical symplectic structure on the reduced phase space. We will come back to this point in section 3.4.

We conclude this section with an analysis of the special case in which the functional connection is flat, $\varpi = h^{-1}\mathbb{d}h$, as in the case of the SdW connection for EM (see theorem 2.19). Using the dressed field formalism (definition 2.22), the horizontal/vertical split of the symplectic structure acquires a more transparent physical meaning in terms of a symplectic structure for the dressed fields ($\Omega^H$) and one for the dressing factor and the Gauss constraint ($\Omega^\partial$):

**Corollary 3.3.** *Suppose* $\varpi$ *is flat, then* $\varpi = h^{-1}\mathbb{d}h$ *and*[31]

$$
\begin{cases}
\theta^H = \int \sqrt{g}\, \mathrm{Tr}(\widehat{E}_{rad}^i\, \mathbb{d}\widehat{A}_i) - \int \sqrt{g}\, \overline{\widehat{\psi}}\gamma^0 \mathbb{d}\widehat{\psi} \\[2mm]
\theta^V = \int \sqrt{g}\, \mathrm{Tr}\big(-\widehat{\mathsf{G}}\, h^{-1}\mathbb{d}h\big) + \oint \sqrt{h}\, \mathrm{Tr}(\widehat{E}_{Coul}^s\, h^{-1}\mathbb{d}h) \qquad (\varpi = h^{-1}\mathbb{d}h) \\[2mm]
\phantom{\theta^V} \approx \oint \sqrt{h}\, \mathrm{Tr}(\widehat{f}\, h^{-1}\mathbb{d}h)
\end{cases}
\tag{60}
$$

---

[30] This is tightly related to the introduction of so-called edge-modes [12]; cf. the discussion in section 7.

[31] The dressed Gauss constraint $\widehat{\mathsf{G}}$ has the same functional expression of $\mathsf{G}$ with the fields $\phi = (A, E_{\mathrm{Coul}}, \psi, \overline{\psi})$ replaced by their dressed counterparts $\widehat{\phi} = (\widehat{A}, \widehat{E}_{\mathrm{Coul}}, \widehat{\psi}, \overline{\widehat{\psi}})$. As a result $\widehat{\mathsf{G}} = h\mathsf{G}h^{-1}$. Similarly for the definition of $\widehat{E}_{\mathrm{rad}} = hE_{\mathrm{rad}}h^{-1}$ and $\widehat{E}_{\mathrm{Coul}} = hE_{\mathrm{Coul}}h^{-1}$.

*and*

$$
\begin{cases}
\Omega^H = \displaystyle\int \sqrt{g}\,\mathrm{Tr}\big(\mathbb{d}\widehat{E}_{rad}^i \curlywedge \mathbb{d}\widehat{A}_j\big) - \int \sqrt{g}\big(\mathbb{d}\widehat{\overline{\psi}} \curlywedge \gamma^0 \mathbb{d}\widehat{\psi}\big) \\[2mm]
\Omega^\partial = -\displaystyle\int \sqrt{g}\,\mathrm{Tr}\big(\mathbb{d}\widehat{G} \curlywedge h^{-1}\mathbb{d}h\big) + \oint \sqrt{h}\,\mathrm{Tr}\big(\mathbb{d}\widehat{E}_s \curlywedge h^{-1}\mathbb{d}h\big) \qquad (\varpi = h^{-1}\mathbb{d}h). \\[2mm]
\qquad \approx \displaystyle\oint \sqrt{h}\,\mathrm{Tr}\big(\mathbb{d}\widehat{f} \curlywedge h^{-1}\mathbb{d}h\big)
\end{cases}
\tag{61}
$$

In EM, $h = e^\varsigma$ and $h^{-1}\mathbb{d}h = \mathbb{d}\varsigma$ and thus these formulas show that the dressing factor $\varsigma$ *is the dof conjugate to the Gauss constraint.*

This has a nice interpretation in terms of the Dirac formalism for constrained system: the choice of $\mathsf{G} = 0$ as the first class constraint and of $\varsigma = 0$ as the gauge-fixing second class constraint, puts the Dirac's matrix of (off-shell) Poisson brackets between the constraints in normal (Darboux) form. In this article, we will not elaborate on this observation any further.

## 3.2 The radiative/Coulombic split and the SdW connection

In this brief interlude, we turn to the SdW choice of connection in relation to the radiative/Coulombic split. Since the SdW connection is built out of similar orthogonality conditions as those involved in the split of $E$, the SdW choice leads to a more harmonious formalism.

First, the radiative part of the electric field (52) and the SdW-horizontal perturbations of $A$ (27) satisfy the same functional properties, that is they are both covariantly divergence-free and fluxless. This is of course a consequence of them both being *de facto* determined by orthogonality to $V = \mathrm{T}\mathcal{F}$, i.e. to the pure-gauge directions in $\mathcal{A}$ (figure 4). This agreement of their functional properties is particularly welcome in the Lagrangian context, for then one has, in temporal gauge, that the radiative electric field corresponds to the SdW-horizontal component of the velocity vector[32] $\mathbb{E}_{\mathrm{rad}} = \widehat{H}_{\mathbb{G}}(\dot{\mathbb{A}})$—a relationship that holds if and only if one makes use of the SdW notion of horizontality, with or without boundaries.

To the extent that there is a parallel between $E_{\mathrm{rad}}$ and $\mathbb{d}_\perp A$, a parallel also exists between $E_{\mathrm{Coul}}$ and $\varpi_{\mathrm{SdW}}$.

**Proposition 3.4** (SdW radiative/Coulombic decomposition). *Let $\varpi = \varpi_{SdW}$. Then, $E_{Coul}$ is the pure-gradient part in the (generalized) Helmholtz decomposition of $E$, denoted*

$$
E_{Coul}^i = g^{ij}\mathrm{D}_j\varphi \qquad (SdW),
\tag{62}
$$

*with $\varphi$ the* (SdW-)Coulombic potential. *Expressed in terms of $\varphi$, the Gauss constraint (53) then reads*

$$
\mathsf{G}_f: \quad
\begin{cases}
\mathsf{G} = \mathrm{D}^2\varphi - \rho \approx 0, & in\ R, \\[1mm]
\mathsf{G}_f^\partial = \mathrm{D}_s\varphi - f \approx 0, & at\ \partial R,
\end{cases}
\qquad (SdW),
\tag{63}
$$

*which is another SdW value problem, cf. (25).*

*Proof.* This directly follows from the formal analogy between (51) and (20). Cf. proposition 2.14 $\qquad\square$

We have now all the tools necessary to state the existence and uniqueness of the solution to the Gauss constraint, *once a a choice of functional connection $\varpi$ is given* (53):

**Proposition 3.5** (Uniqueness of $E_{\mathrm{Coul}}$ [20]). *For any choice of functional connection $\varpi$ and electric flux $f$, the Gauss constraint $\mathsf{G}_f = 0$ has one and only one solution $E_{Coul} = E_{Coul}(A,\rho,f)$.*

---

[32] $\mathbb{E}_{\mathrm{rad}} := \int \sqrt{g}\, g_{ij}E_{\mathrm{rad}}^i \frac{\delta}{\delta A_j} \in H = V^\perp \subset \mathrm{T}\mathcal{A}$.

*Proof.* The proof of this statement proceeds in two steps. In the first step we prove the existence and uniqueness of the solution to the Gauss constraint for the SdW choice of connection, i.e. $\varpi = \varpi_{\text{SdW}}$. This is a consequence of (63) and the general properties of the SdW boundary value problem, see proposition 2.18 (notice that in this case uniqueness holds even at reducible configurations, since we are ultimately interested in $E_{\text{Coul}}$, not $\varphi$). In the second step, one can show that this result for the SdW connection can be used to prove existence and uniqueness for any other choice of connection. For details on the second step, see [20]. □

### 3.3 Gauge properties of the horizontal/vertical split

In this section we will characterize the properties of $\theta^{H,V}$ and $\Omega^{H,V}$ in relation to gauge. First, however, we characterize the gauge properties of the off-shell symplectic potential $\theta$:

**Proposition 3.6.** *The following propositions on $\theta$ hold true:*

   (i) *$\theta$ is gauge invariant only under field-independent gauge transformations $\xi$, $\mathrm{d}\xi = 0$;*

   (ii) *on-shell of the Gauss constraint $\mathsf{G}_f \approx 0$ and in the absence of boundaries $\partial R = \emptyset$, $\theta \, \Theta$ is gauge invariant under all gauge transformations.*

*Proof.* Using the gauge transformation properties of $A$, $E$, $\psi$ and $\psi$ (e.g. $\mathbb{L}_{\xi^\sharp} A = \mathrm{D}\xi$), as well as $[\mathbb{L}, \mathrm{d}] = 0$, an explicit computation shows that

$$\mathbb{L}_{\xi^\sharp}\theta = \int \sqrt{g}\,\mathrm{Tr}(E^i \mathrm{D}\mathrm{d}\xi) + \int \sqrt{g}\,\overline{\psi}\gamma^0(\mathrm{d}\xi)\psi = -\int \sqrt{g}\,\mathrm{Tr}(\mathsf{G}\mathrm{d}\xi) + \oint \sqrt{h}\,\mathrm{Tr}(E_s\mathrm{d}\xi). \quad (64)$$

The two propositions follow from the formula above upon imposing respectively $\mathsf{G} \approx 0$ and $\partial R = \emptyset$ for (*ii*), and $\mathrm{d}\xi = 0$ for (*i*). The latter case gives:

$$\mathbb{L}_{\xi^\sharp}\theta = 0, \qquad (\mathrm{d}\xi = 0). \quad (65)$$

Notice that since $\mathrm{d}$ and $\mathbb{L}$ commute, this implies $\mathbb{L}_{\xi^\sharp}\Omega = 0$ if $\mathrm{d}\xi = 0$. □

Ultimately, the reason (65) holds is that, in YM theory, the conjugate momentum to $A$ transforms covariantly, rather than as a connection. Thus, the fact that a polarization of the symplectic potential exists such that (65) holds, is a property of Yang-Mills theory not shared by either $BF$ or Chern-Simons theories.[33] This property will be implicitly at the root of much of the following analysis.

From (65) one can readily deduce the following corollary characterizing the Hamiltonian flow of a gauge transformation with an eye for the field-dependence of the gauge transformation involved.

**Corollary 3.7.** *The following propositions hold true:*

   (i) *Off shell of the Gauss constraint (and irrespectively of boundaries), only field-independent gauge transformations $\xi$, such that $\mathrm{d}\xi = 0$, have a Hamiltonian generator $H_\xi$ with respect to $\Omega = \mathrm{d}\theta$. This generator, up to a field-space constant, is given by*

$$H_\xi := \theta(\xi^\sharp) = \theta^V(\xi^\sharp), \qquad (\mathrm{d}\xi = 0); \quad (66)$$

   (ii) *In the absence of boundaries $\partial R = \emptyset$, $H_\xi$ is a smearing of the Gauss constraint and therefore vanishes on shell of the Gauss constraint, $H_\xi \approx 0$;*

---

[33]See also [58], where the *failure* to satisfy an extended analogue of the above equation plays a role in the BV-BFV derivation of the Chern-Simons's edge theory.

*(iii)* In the presence *of boundaries* $\partial R \neq \emptyset$, $H_\xi$ is not *a smearing of the Gauss constraint, and generally fails to vanish on shell of the Gauss constraint; indeed, in this case,* $H_\xi \approx \oint \sqrt{h}\,\mathrm{Tr}(f\,\xi)$.

*Proof.* Application of Cartan's formula $\mathbb{L}_{\xi^\sharp} = \mathbb{i}_{\xi^\sharp}\mathbb{d} + \mathbb{d}\mathbb{i}_{\xi^\sharp}$ to the left-most expression in (64), together with the definition $\Omega = \mathbb{d}\theta$, gives

$$\mathbb{i}_{\xi^\sharp}\Omega + \mathbb{d}\theta(\xi^\sharp) = \mathbb{L}_{\xi^\sharp}\theta = -\int \sqrt{g}\,\mathrm{Tr}(\mathrm{G}\mathbb{d}\xi) + \oint \sqrt{h}\,\mathrm{Tr}(E_s\mathbb{d}\xi). \tag{67}$$

Off shell of the Gauss constraint, the right hand side is exact, and actually vanishes, only if $\mathbb{d}\xi = 0$ irrespectively of the presence of boundaries. Also, from the remark below (47), it is clear that $\theta(\xi^\sharp) \equiv \theta^V(\xi^\sharp)$. Hence,[34]

$$0 = \mathbb{L}_{\xi^\sharp}\theta = \mathbb{i}_{\xi^\sharp}\Omega + \mathbb{d}H_\xi, \qquad (\mathbb{d}\xi = 0). \tag{68}$$

This proves *(i)*. To prove *(ii-iii)*, it is enough to write $H_\xi$ explicitly, starting from the expression (54) for the vertical symplectic form:

$$H_\xi = \int \sqrt{g}\,\mathrm{Tr}(E^i\mathrm{D}_i\xi + \rho\,\xi)$$

$$= -\int \sqrt{g}\,\mathrm{Tr}(\mathrm{G}\xi) + \oint \sqrt{h}\,\mathrm{Tr}(E_s\xi) \approx \oint \sqrt{h}\,\mathrm{Tr}(f\,\xi). \tag{69}$$

This concludes the proof. $\qquad\square$

(This corollary provides an explicit answer to the question of why one should introduce field-dependent gauge transformations at all: field-dependent gauge transformation serve as a diagnostic tool to detect the presence of spacetime boundaries from a geometric analysis within field-space. Of course, a more abstract answer to this same question is that arbitrary vertical vector fields—aka field dependent gauge transformations—are natural geometric objects on field space.)

Heuristically, this corollary emphasizes once again that the dof contained in $H_\xi$ are precisely those dof whose Hamiltonian flow generates translations in the pure-gauge part of the fields, that is (loosely speaking) $\varpi$. Conversely, the following two results confirm that the dof contained in $\theta^H$ play no role in the flow equation along the pure gauge directions (68):

**Proposition 3.8** (Gauge properties of $\Omega^H$). *The horizontal symplectic form* $\Omega^H := \mathbb{d}\theta^H$ *is horizontal and gauge-invariant, i.e.* basic, *and can be expressed as* $\Omega^H = \mathbb{d}_H\theta^H$.

*Proof.* The first part of the proposition is another consequence of (65)—together with the equivariance of $\mathbb{d}_H A$ (14) [17,19]. Indeed, these two equations imply that $\theta^H$ itself *basic*:[35]

$$\mathbb{i}_{\xi^\sharp}\theta^H = 0 \quad \text{and} \quad \mathbb{L}_{\xi^\sharp}\theta^H = 0. \tag{70}$$

Notice that both of these equations—contrary to (65)—hold for field-*dependent* $\xi$'s as well. Because $\theta^H$ is basic, using the result (12) on the differential of horizontal and equivariant forms, it is immediate to see that $\mathbb{d}_H\theta^H = \mathbb{d}\theta$ and hence that $\Omega^H$ is also basic. Crucially, these results hold[36] for *any* $\varpi$, even when $\mathbb{F} \neq 0$. $\qquad\square$

---

[34]In the following, to remind the reader which equations are subject to the conditional $\mathbb{d}\xi = 0$, we will explicitly include it in parenthesis.

[35]The second equations follows from (14) and an analogous formula for $\mathbb{d}_H\psi$ which can be deduced from (12). For more explicit details see equation (6.29) in [19].

[36]Beside the previous abstract argument, an explicit, albeit non-illuminating, proof of the right-most equality can be found in appendix B.2 (equation 109) of [17].

**Corollary 3.9** (A trivial flow for gauge transformations)**.** *With respect to the horizontal symplectic structure $\Omega^H$, gauge transformations have a* trivial *Hamiltonian flow.*

*Proof.* Application of Cartan's formula to (70) gives

$$0 = \mathbb{L}_{\xi^\sharp} \theta^H = \mathbb{i}_{\xi^\sharp} \Omega^H + \mathbb{d}\theta^H(\xi^\sharp). \tag{71}$$

This flow equation can be called trivial, because each of the two terms on the right-most formula vanish *independently* (even if $\mathbb{d}\xi \neq 0$). □

### 3.4 Quasilocal symplectic reduction

The results presented in this section are discussed and proved in greater detail in [20]. We briefly review them here for completeness, but they will not be needed in the following.

As proved in the previous sections, the horizontal symplectic form $\Omega^H$ is basic, i.e. both horizontal and gauge-invariant. As a consequence it can be unambiguously projected down to a 2-form $\Omega^H_{\text{proj}}$ on the reduced, on-shell phase space $\Phi//\mathcal{G}$.[37] Moreover, since $\Omega^H$ is closed, $\Omega^H_{\text{proj}}$ is also closed. However, for it to define a *symplectic* structure on $\Phi//\mathcal{G}$, $\Omega^H_{\text{proj}}$ would need to be non-degenerate as well. It turns out that, *in the presence of boundaries*, this is not the case.

Physically, this is simple to understand: $\Omega^H_{\text{proj}}$ fails to provide a symplectic structure for the *Coulombic* dof. The reason why this does not happen in the absence of boundaries is because $E_{\text{Coul}}$ is fully determined by the matter degrees of freedom, and therefore does not need to independently appear in the symplectic structure. However, in the presence of boundaries, $E_{\text{Coul}}$ is determined by $\rho$ *as well as* $f$ (53). Thus, loosely speaking, what is missing in $\Omega^H_{\text{proj}}$ is a symplectic structure for the fluxes $f$.

In sum, not only does $\Omega^H_{\text{proj}}$ depend on the choice of $\varpi$ (corollary 3.2), but it also fails to be non-degenerate (and therefore symplectic).

Both these problems can be solved in one stroke by resorting to the concept of (covariant) superselection sectors. In the Abelian case, this means simply that one "stratifies" the reduced phase space $\Phi//\mathcal{G}$ by subspaces at fixed value of $f$. Notice that in the Abelian case $f$ is gauge invariant and therefore a well-defined quantity on the reduced phase space. As a result, within each superselection sector, $E_{\text{Coul}}$ is also completely fixed by the matter dof and we are therefore in a situation similar to that of the case without boundary. Thus, in the Abelian case, although $\Phi//\mathcal{G}$ is not symplectic, each superselection sector is. In the presence of field-space curvature, the appropriate symplectic structure here is not $\Omega^H_{\text{proj}}$ but rather the projection of $\Omega(\widehat{H},\widehat{H}) \approx \Omega^H + \oint \sqrt{h}\,\text{Tr}(f\,\mathbb{F})$—which is now closed within a superselection sector since there $\mathbb{d}f \equiv 0$ (and $\mathbb{d}\mathbb{F} \equiv 0$ by the Bianchi identity; cf. corollary 3.2). One can show that the resulting symplectic structure is also independent of $\varpi$.

In the non-Abelian case, fixing $f$ would be tantamount to breaking the gauge symmetry at the boundary. Therefore the best one can do is to fix $f$ up to gauge, i.e. demand that $f$ belongs to the set $[f] = \{f = g^{-1}f\,g, \text{for some}\,g \in \mathcal{G}\}$. The restriction of $\Phi$ to those configurations on-shell of the Gauss constraint with $f \in [f]$ is called a *covariant* superselection sector. However, fixing a *covariant* superselection sector is not enough for the Gauss constraint to fully fix $E_{\text{Coul}}$ in terms of the matter field: $f$ can still be varied within $[f]$, even at *fixed* $(A, E_{\text{rad}}, \psi)$. These transformations, that vary $f$ within $[f]$ while leaving the other fields fixed, are called *flux rotations* and are *physical* transformations (gauge transformations would have to uniformly act on the other fields as well). Therefore, loosely speaking, to define a symplectic structure over

---

[37]Here $\Phi//\mathcal{G}$ denotes the symplectic reduction of $\Phi$, which requires both going on-shell of Gauss and modding out gauge transformations.

the (gauge-reduced) covariant superselection sector, one needs to add a symplectic structure for the superselected fluxes $f \in [f]$.

This can be done in a canonical manner, by realizing that $[f]$ is essentially a (co)adjoint orbit in $\mathcal{G}$ and by resorting to the canonical Kirillov–Konstant–Sourieu (KKS) symplectic structure on coadjoint orbits. A properly constructed horizontal variation of the KKS symplectic structure over the fluxes, $\omega^H_{\text{KKS}}$, can then be added to $\Omega^H$. The resulting 2-form $\Omega^H_f = \Omega^H + \omega^H_{\text{KKS}}$ is basic, closed, and projects to a non-degenerate symplectic structure within a reduced covariant superselection sector. The resulting symplectic structure is also independent of the choice of $\varpi$.

We refer to the procedure of adding to $\Omega^H$ the canonically constructed symplectic structure on the fluxes, $\omega^H_{\text{KKS}}$, as the "canonical completion" of the symplectic structure $\Omega^H$. We call it a completion of the symplectic structure—as opposed e.g. to an extension of the phase space (à la "edge mode")—because this procedure fixes the degeneracy of $\Omega^H_{\text{proj}}$ over $\Phi//\mathcal{G}$ without enlarging the space $\Phi//\mathcal{G}$, that is without adding any extra degree of freedom to the phase space.

Mathematically, this procedure is closely related to performing the Marsden–Weinstein symplectic reduction [59] not on the pre-image of the zero-section of the momentum map, but on the pre-image of a coadjoint orbit of the moment map: indeed, from Corollary 3.7 the relevant moment map is $H : \xi \mapsto H_\xi$ such that $H_\xi = \int \sqrt{g} \text{Tr}(E^i D_i \xi + \rho \xi) \approx \oint \sqrt{h} \text{Tr}(f \xi))$. The crucial subtlety is that one still wants the Gauss constraint strictly imposed in the bulk, which means that one focuses on non-zero coadjoint orbits of $H$ that are, so to say, concentrated at the boundary only.

One last remark: although the (completed) reduced symplectic form is independent of a choice of $\varpi$ (analogously, independent of a choice of what one could call a "covariant perturbative gauge-fixing"), the basis in which one describes the physical dof *will* depend on that choice. In particular, which electric degrees of freedom are *precisely* coordinatized by $f$ through (53) depends on the choice of $\varpi$. This is important, for example, when considering how (on-shell of the Gauss constraint) a "flux rotation" alters the bulk electric field $E = E_{\text{rad}} + E_{\text{Coul}}$. Flux rotations, for different choices of $\varpi$ and for the same value of $f$, will have different effects on the bulk electric field (as the "meaning" of $f$—i.e .the component of the electric field that it coordinatizes—changes along with these choices).

# 4 Charges[38]

In the previous sections we have established that dynamical quantities in the quasi-local gauge-reduced phase space—which are by definition gauge invariant—are encoded in the horizontal symplectic structure $\Omega^H$. We have also noticed that the generator of gauge transformations, $H_\xi = \int \sqrt{g} \text{Tr}(\mathsf{G}_f \xi) \approx \oint \sqrt{h} \text{Tr}(f \xi)$, are encoded rather in the remaining part of the symplectic structure, $\Omega^\partial$ (66).

This means that the gauge generators $H_\xi$, which are the (naive) Noether charges for the gauge symmetry, have in general no bearing on the radiative degrees of freedom in the bulk of $R$. Shortly, we will argue that these charges do not encode any particular conservation laws. (These facts notwithstanding, these charges still encode information on the $f$-superselection sector, which is an important physical information; notice, however, how this statement has to be qualified: in non-Abelian theories, neither $f$ nor any of the $H_\xi$ is gauge invariant and therefore observable as such.)

---

[38]In this section we shall rectify some statements made in [19]. In particular, contrary to what we had assumed there, the horizontal symplectic form is *not* invariant under charge-transformations $\chi^\sharp$. Here, we will discuss the origin and consequences of the important obstructions to this statement which had been hitherto missed.

In this section, we are going to clarify these statements, argue that one needs reducible configurations to obtain a gauge-invariant set of charges that satisfy a Gauss' law as well as appropriate conservation laws [21–23], and discuss how these charges are related to certain geometric features of field space and the kernel of the SdW boundary value problem. (Notice that we draw a distinction between the Gauss constraint, which is an elliptic differential equation $DE - \rho = 0$, and the (integrated) Gauss' law, which is an integral relation between the total charge contained in a region and the total electric flux through its boundary, e.g. in electromagnetism $\int \rho = \oint f$.)

At the end of this section, we will briefly comment on the consequences of these observations for the symplectic flow of these charges.

## 4.1 Reducible configurations: an overview

At a configuration $A \in \mathcal{A}$, consider the infinitesimal gauge transformations $\chi_A \in \mathrm{Lie}(\mathcal{G})$ such that

$$\delta_{\chi_A} A \equiv D\chi_A = 0. \tag{72}$$

If a $\chi_A \neq 0$ exists, then $A_i$ is said *reducible* and $\chi_A$ is called a *reducibility parameter* or *stabilizer*. The stabilizers $\chi_A$ depend on the global properties of $\tilde{A}_i$ and constitute a finite dimensional Lie algebra (possibly a zero-dimensional one). Denote this Lie algebra $\mathrm{Lie}(\mathcal{I}_A)$, where $\mathcal{I}_A \subset \mathcal{G}$ is the stabilizer or isotropy group of $A$, i.e. the subgroup of $\mathcal{G}$ composed by the elements $h \in \mathcal{G}$ such that $A^h = A$.

The isotropy group is a covariant notion, in the sense that

$$\mathcal{I}_{A^g} = g^{-1}\mathcal{I}_A g. \tag{73}$$

In general, $\mathcal{I}_A$ is *not* a normal subgroup of $\mathcal{G}$, and $\mathcal{G}_A := \mathcal{G}/\mathcal{I}_A$ is only a (right) quotient and not a group. Infinitesimally, $\mathrm{Lie}(\mathcal{I}_A)$ is a sub-Lie-algebra but generally not an ideal of $\mathrm{Lie}(\mathcal{G})$, and therefore $\mathfrak{G}_A := \mathrm{Lie}(\mathcal{G})/\mathrm{Lie}(\mathcal{I}_A)$ is only a quotient of vector spaces and *not* a Lie algebra. We will indicate elements of $\mathfrak{G}_A$ by $[\xi]_A \equiv [\xi + \chi_A]_A$ or, more often, just $[\xi]$. Since at $A \in \mathcal{A}$, $\chi_A^\sharp \equiv 0$, one has that $[\xi]_A^\sharp = \xi_A^\sharp$ is a well defined vertical vector in $T_A\mathcal{A}$.

In non-Abelian theories, reducible configurations form a meager set,[39] in the same way as those spacetime metrics which admit non-trivial Killing vector fields are "extremely rare" (i.e. form a meager set). In this respect, Abelian theories, such as electromagnetism, are an exception: *all* their configurations have as their reducibility parameter the constant gauge transformation, e.g. $\chi_{\mathrm{EM}} = const \in i\mathbb{R}$.

This means that the action of $\mathcal{G}$ does not act freely on $\mathcal{A}$, and therefore $\mathcal{A}$ cannot be a bona-fide (infinite-dimensional) principal fibre bundle, since it lacks the necessary, homogeneous local product structure: fibres associated to reducible configurations are not isomorphic to $\mathcal{G}$—see figure 8. However, it turns out that $\mathcal{A}$ can be decomposed into "strata" defined by an increasing degree of symmetry, each of which does have (at least locally) a product structure. Indeed, a *slice theorem* shows that $\mathcal{A}$ is regularly stratified by the action of $\mathcal{G}$, and in particular that all the strata are smooth submanifolds of $\mathcal{A}$.

Given the gauge-covariance of the constructions involved in the slice theorem, the stratification of field space survives the gauge-reduction, and is thus geometrically reflected in the structure of the reduced field space. This will give us the opportunity to build gauge invariant (sets of) global charges.

The kernel of the SdW boundary value problem at $A \in \mathcal{A}$ (cf. (25) and (63)) is provided precisely by the reducibility parameters of $A$ (see proposition 2.18). This tyies the SdW kernel

---

[39]A *meagre* set is one whose complement is an everywhere dense set: roughly, an arbitrarily small perturbation takes one out of a meager set. Here, reducible configurations form a meager set according to the standard field-space metric topology on $\mathcal{A}$ (the Inverse-Limit-Hilbert topology [28, 29], see also [60]).

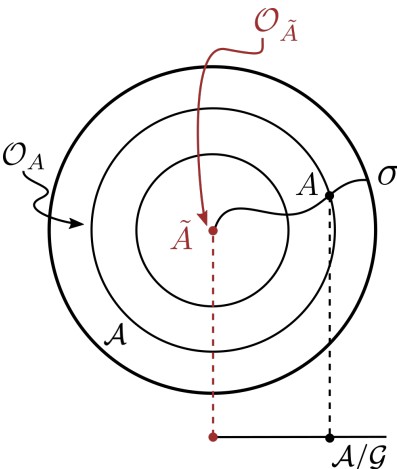

Figure 5: In this representation $\mathcal{A}$ is the page's plane and the orbits are given by concentric circles. The field $A$ is generic, and has a generic orbit, $\mathcal{O}_A$. The field $\tilde{A}$ has a nontrivial stabilizer group (i.e. it has non-trivial reducibility parameters), and its orbit $\mathcal{O}_{\tilde{A}}$ is of a different dimension than $\mathcal{O}_A$. The projection of $\tilde{A}$ on $\mathcal{A}/\mathcal{G}$ therefore sits at a qualitatively different point than that of $A$ (a lower-dimensional stratum of $\mathcal{A}/\mathcal{G}$). Exclusion of the reducible configuration $\tilde{A}$ gives rise to a fibre bundle structure over $\mathcal{A} \setminus \{\tilde{A}\}$; here $\sigma$ represents a section of $\mathcal{A} \setminus \{\tilde{A}\}$. Locally, the concept of section can be generalized to include reducible configurations such as $\tilde{A}$, thus leading to the notion of "slice". This is briefly reviewed in appendix B.

to certain geometrical properties of $\mathcal{A}$: a fact which has two main consequences. First, it means that the SdW kernel is empty almost-everywhere in $\mathcal{A}$ and therefore that the SdW boundary value problem has generically a unique solution. Second, it means that at a generic non-Abelian configuration there is no integrated Gauss law nor a gauge invariant notion of conserved charges. The goal of this section is to analyze and explain these statements and show how one can leverage the a relation between the SdW kernel and the geometry of $\mathcal{A}$. We will take electromagnetism as the epitomic Abelian theory.

## 4.2 Green's functions

Let us consider the properties of the Green's functions $G_{\alpha,x}(y)$ of the SdW boundary value problem entering the definition of the radiative degrees of freedom as well as of the Coulombic ones. For definiteness and future convenience, we will focus on the example of the Coulombic degrees of freedom.

The Green's functions $G_{\alpha,x}(y)$ are defined by the following[40]

$$
\begin{cases}
\mathrm{D}^2 G_{\alpha,x}(y) = \tau_\alpha \delta_x(y), & \text{in } R, \\
\mathrm{D}_s G_{\alpha,x}(y) = 0, & \text{at } \partial R.
\end{cases}
\tag{74}
$$

Physically, this choice of (covariant) Neumann boundary conditions corresponds to the demand that the charged perturbation inserted at $x$ does *not* contribute to the flux $f$ at $\partial R$.

In EM, this choice of boundary conditions is inconsistent and should be amended e.g. by demanding that it creates a constant flux compatible with the integrated Gauss law.[41] This is

---

[40]Here, $\delta_x(y) \equiv \delta(x,y)$ is a Dirac delta distribution. The notation is meant to emphasize the index structure of $\mathrm{Lie}(\mathcal{G})$.

[41]In EM one possible natural boundary condition is $\partial_s G = 1/\mathrm{Vol}(\partial R)$ with $\mathrm{Vol}(\partial R)$ the volume of the region's

because in EM all configuration have as a stabilizer $\chi_{\text{EM}} = const$. However, as we discussed above, *generic* non-Abelian configurations are irreducible and possess no such stabilizer: therefore at these configurations there is *no integrated Gauss law* that the Green's function should respect.

Indeed, the extension of the (Abelian) integrated Gauss law $\int \rho = \int \nabla_i E^i = \oint f$, to the non-Abelian context would be

$$\int \sqrt{g}\,\text{Tr}(\tau_\alpha \rho) \approx \int \sqrt{g}\,\text{Tr}(\tau_\alpha D_i E^i) = -\int \sqrt{g}\,\text{Tr}(E^i D_i \tau_\alpha) + \oint \sqrt{h}\,\text{Tr}(\tau_\alpha f), \qquad (75)$$

but the bulk term on the rightmost side vanishes only if $A$ is reducible and $\tau_\alpha$ is replaced by a reducibility parameter. Notice that reducibility parameters are rigid, i.e. their value at one point determines their value everywhere (they solve a first-order differential equation), and only in such a situation does one lose the functional independence between bulk and boundary integrals. The ensuing functional dependence is what makes the very possibility of having an integrated Gauss law meaningful. Therefore, we conclude that at a generic configuration of a non-Abelian YM theory, there is no (integrated) Gauss law relating total charges and (integrated) electric fluxes.[42]

Using the definition (74) of the Green's function, together with the following non-Abelian generalization of Green's theorem (e.g. [61]),

$$\int_R \sqrt{g}\,\text{Tr}(\psi_1 D^2 \psi_2 - \psi_2 D^2 \psi_1) = \oint_{\partial R} \sqrt{h}\,\text{Tr}(\psi_1 D_s \psi_2 - \psi_2 D_s \psi_1) \quad \forall \psi_{1,2} \in \Omega^0(R, \text{Lie}(\mathcal{G})), \tag{76}$$

one can choose $\psi_1 = \varphi$ and $\psi_2 = G_{\alpha,x}$, to obtain the Coulombic component of the electric field in terms of the charge density $\rho$ and the flux $f$:

$$\varphi_\alpha(x) = \int_R \sqrt{g(y)}\,\text{Tr}\Big(G_{\alpha,x}(y) D^2 \varphi(y)\Big) - \oint_{\partial R} \sqrt{h(y)}\,\text{Tr}\Big(G_{\alpha,x}(y) D_s \varphi(y)\Big)$$

$$\approx \int_R \sqrt{g(y)}\,\text{Tr}\Big(G_{\alpha,x}(y)\rho(y)\Big) - \oint_{\partial R} \sqrt{h(y)}\,\text{Tr}\Big(G_{\alpha,x}(y) f(y)\Big). \tag{77}$$

At reducible configurations, this formula must again be amended by the addition of constant offsets due to the modified boundary condition. The addition of such offsets is possible thanks to the freedom of redefining $G_{\alpha,x}$ by some combination of the reducibility parameters, since they lie in the kernel of (74).

A similar construction allows us to solve the SdW boundary value problem for $\varpi_{\text{SdW}}$. More on this in section 5.

## 4.3 Conserved charges

To talk about conservation laws, we consider now a spacetime process in the presence of matter.

At the light of the previous section, we see that an integrated Gauss law exists at reducible configurations. This suggests the possibility of defining *stabilizer charges* at such configurations

---

boundary $\partial R$, whereas in YM at a $\tilde{A}$ with a single reducibility parameter $\chi$, the following $y$-constant boundary condition plays a similar role $D_s G_{\alpha,x} = \text{Tr}(\chi(x)\tau_\alpha)/\|\chi\|^2_{\partial R}$.

[42]The issue is formally the same as the difficulties present in defining quasi-local conserved quantities in general relativity. Also, notice that bringing the gauge field contribution on the left-hand side of (75) to make the ensuing "integrated Gauss law" satisfied identically is a trick with no bearing on the dressing problem and the definition of the Green's functions for the matter field.

via (75):[43]

$$Q[\chi_A] := \int \sqrt{g}\,\mathrm{Tr}(\rho\,\chi_A) \approx \oint \sqrt{h}\,\mathrm{Tr}(\chi_A f)\,. \tag{78}$$

As a consequence of the Gauss law, these charges are inherently *quasilocal*, insofar the value of $Q[\chi]$ over a closed Cauchy surface $\Sigma$, $\partial\Sigma = \emptyset$, necessarily vanish.

Notice also that, if $\dim(\mathcal{I}_A) = 1$, and we take $\chi_A$ to be of unit[44] norm, then from (73) it follows that $\chi_{A^g} = g^{-1}\chi_A g$ and therefore $Q[\chi_A]$ is gauge invariant. However, if $\dim(\mathcal{I}_A) > 1$, the identification of specific elements of $\mathrm{Lie}(\mathcal{I}_A)$ at different $A$'s is more problematic: for this reason in the reminder of this section we will focus on the case $\dim(\mathcal{I}_A) = 1$ and will comment on its generalization at the end.

Having established a Gauss law and the gauge invariance of the stabilizer charge $Q[\chi_A]$, we now ask whether such charges are also conserved.

Consider a configuration of $\Phi = \mathrm{T}\mathcal{A} \times (\overline{\Psi} \times \Psi)$ whose time evolution in temporal gauge allows a time-*in*dependent extension of $\chi_A$ to a *time* neighbourhood $N$ of $R$, $N = R \times (t_0, t_1)$, that satisfies $\mathrm{D}_i \chi_A = 0$ at every time.[45] Then, the quantity $Q[\chi_A]$ is conserved in the sense that it satisfies a balance law in terms of the matter current [21–23]:

$$0 = \int_N \sqrt{g}\,\nabla_\mu \mathrm{Tr}(\chi J^\mu) = \Delta_{t_1, t_0} Q[\chi] + \int_{t_0}^{t_1} \mathrm{d}t\, F_{\partial R}[\chi]\,, \tag{79}$$

where we introduced the fluxes $F_{\partial R}[\chi] := \oint \sqrt{h}\,\mathrm{Tr}(\chi J_s)$ through $\partial R$. The first equality follows from $\mathrm{D}_\mu \chi_A = 0$ as well as the equation of motion $\mathrm{D}_\mu J^{\mu\alpha} = 0$ (Noether II); instead, the second equality is just an application of Stokes' theorem.

It is important to stress that all integrands in the above balance law are *quantities constructed geometrically* from the properties of $A$ in $N$, with gauge-invariance properties analogous to those of $Q[\chi_A]$. Indeed, the existence and properties of a $\chi_A$ such that $\mathrm{D}_\mu \chi = 0$ are gauge-invariant features of the configuration history $A(t)$. The construction of such quantities would not be possible at non-reducible configurations, where the equation (Noether II) $\mathrm{D}_\mu J^{\mu\alpha} = 0$ does *not* constitute an appropriate replacement of the above.

Notice that the condition $\mathrm{D}_\mu \chi_A = 0$ on the whole of $N = R \times (t_0, t_1)$ implies that $A(t)$ and $E(t) = \dot{A}(t)$ are both reducible, and therefore—via the Gauss constraint—that $\rho$ commutes with $\chi_A$:

$$\mathrm{D}_\mu \chi_A = 0 \quad \Rightarrow \quad \mathrm{D}_i \chi_A = 0, \quad [E_i, \chi_A] = 0 \quad \text{and} \quad [\rho, \chi_A] = 0\,. \tag{80}$$

We conclude this section by noticing that the balance equation expressed in (79) is akin to the conservation of Komar charges for Killing vector fields in general relativity. Similarly, the impossibility of identifying a meaningful non-Abelian charge density over generic, i.e. nonreducible, configurations parallels, in general relativity, the difficulties in identifying conserved stress-energy charges away from backgrounds with Killing symmetries [62]. In general relativity, the Komar charges encode the conservation of energy-momentum and angular-momentum in the test particle approximation over a symmetric background. The physical relevance of this approximation in general relativity is more than well established (think of the theory special

---

[43]Notice that $\mathrm{Tr}(\rho\chi) = \overline{\psi}\chi\psi$. Therefore, this charge can be zero even for $\rho \neq 0$, e.g. if $\chi\psi = 0$ while $\psi \neq 0$. For matter in the fundamental representation, the latter condition is not attainable for $G = \mathrm{SU}(2)$, but it is for larger $N > 2$. This situation was analyzed in [19] through the lens of the Higgs mechanism for condensates.

[44]$\mathrm{Lie}(\mathcal{G})$ is equipped with the following canonical positively-definite inner product: $\langle \xi, \eta \rangle := \int \sqrt{g}\,\mathrm{Tr}(\xi\eta)$.

[45]This is equivalent to asking that the evolution is confined to the stratum $\mathcal{N}_A$—see the end of this section and appendix B. It is possible that such $\chi$'s are uniquely fixed by demanding that they conserve both $A$ and $E_{\mathrm{rad}}$ (and then evolving these solutions in time). Whether these motions are physically relevant (at least in some approximation) is not clear to us. We are also ignoring here the difficulty of identifying a given $\chi \in \mathrm{Lie}(\mathcal{I}_A)$ at different configurations in $\mathcal{N}_A$ when $n = \dim(\mathcal{I}_A) > 1$—see the last paragraph of section 4.5. This difficulty might result in the "mixing" of different stabilizer charges, which is inconsequential for the present argument.

relativity); the same cannot be said for YM. (This difference is due to the extreme weakness of the gravitational attraction compared to the other forces.) Finally, within the present framework, the construction of asymptotic YM charges at null infinity that are more akin to the Bondi, rather than Komar, charges was carried out in [7].

The above analysis has focused on the space and space-time properties of reducible configurations. In the following sections, we will instead focus on their field-space and symplectic properties. Whereas these properties will be analyzed in detail in the case of the Abelian theory, in the non-Abelian case we will limit ourselves at emphasizing the difficulties a generalization would incur.

## 4.4 Charges and symplectic geometry in electromagnetism

As the prototypical example of an Abelian YM theory, we will focus on electromagnetism (EM). As we have already notice, in the *Abelian* case *all* configurations are reducible. It is therefore necessary to incorporate charge transformations in the symplectic treatment of the Abelian theory. In this case, for all $A$, $\mathcal{I}_A \equiv \mathcal{I}_{\mathrm{EM}} \cong G$ is the (normal) subgroup of constant gauge transformations in $\mathcal{G}$ (the "electric-charge" group), and therefore $\mathcal{A}_{\mathrm{EM}}$ has the structure of a bona-fide infinite dimensional fibre bundle for the group $\mathcal{G}_{\mathrm{EM}} := \mathcal{G}/\mathcal{I}_{\mathrm{EM}}$.

Since in EM the electric field $E$ is gauge invariant, the matter-free phase space $\mathrm{T}^*\mathcal{A}_{\mathrm{EM}}$ inherits a bona-fide fibre bundle structure with respect to the same quotient group. In particular, all phase space configurations $(A, E) \in \mathrm{T}^*\mathcal{A}_{\mathrm{EM}}$ are reducible with respect to the constant gauge transformations $\chi_{\mathrm{EM}} = const$. However, in spite of this, none of the matter field configurations for which $\psi \neq 0$ is thus reducible:

$$\delta_{\chi_{\mathrm{EM}}}\psi = -\chi_{\mathrm{EM}}\psi \neq 0. \tag{81}$$

Indeed, $\chi^{\sharp}_{\mathrm{EM}}$ as a vector field on the full phase space $\Phi_{\mathrm{EM}} = \mathrm{T}^*\mathcal{A}_{\mathrm{EM}} \times (\overline{\Psi} \times \Psi)$ reads[46]

$$\chi^{\sharp}_{\mathrm{EM}} = \int (-\chi_{\mathrm{EM}}\psi)^B(x)\frac{\delta}{\delta\psi^B(x)} + (\overline{\psi}\chi_{\mathrm{EM}})^B(x)\frac{\delta}{\delta\overline{\psi}^B(x)} \in \mathrm{T}\Phi_{\mathrm{EM}}. \tag{82}$$

Therefore, although we can define a functional connection on $\mathcal{A}_{\mathrm{EM}}$ for the the quotient gauge group $\mathcal{G}_{\mathrm{EM}}$, in order to use this connection to define horizontal derivatives on $\Phi_{\mathrm{EM}}$, we need to be able to identify elements $[\xi] = [\xi + \chi_{\mathrm{EM}}] \in \mathrm{Lie}(\mathcal{G}_{\mathrm{EM}})$ with elements of $\mathrm{Lie}(\mathcal{G})$. This cannot be done canonically, and a choice of embedding map of vector spaces

$$\kappa : \mathrm{Lie}(\mathcal{G}_{\mathrm{EM}}) \hookrightarrow \mathrm{Lie}(\mathcal{G}), \tag{83}$$

has to be made. (Notice that in the Abelian case, as long as $\kappa$ preserves the vector-space structure of $\mathrm{Lie}(\mathcal{G}_{\mathrm{EM}})$, it will also preserve its (trivial) Lie-algebra structure.)

A simple choice is to represent $\mathcal{G}_{\mathrm{EM}}$ in $\mathcal{G}$ as the so-called group of "pointed[47] gauge transformations," $\mathcal{G}_* := \{g \in \mathcal{G} \text{ such that } g(x_*) = \mathrm{id}\}$ for a certain (arbitrary) $x_* \in R$. I.e. fixing $\kappa(\xi)$ as the only element of $\xi + \mathrm{Lie}(\mathcal{I}_{\mathrm{EM}})$ such that $(\kappa(\xi))(x_*) = 0$.

But there are other possibilities, too. In the following we will denote by $\xi_*$ elements of $\kappa(\mathrm{Lie}(\mathcal{G}_{\mathrm{EM}})) \subset \mathrm{Lie}(\mathcal{G})$, irrespectively of the choice of $\kappa$ that has been made. At the end of this

---

[46]Here, $B$ is a spinorial index in $\mathbb{C}^4$, e.g. the Dirac gamma matrices $\gamma^\mu$ have components $(\gamma^\mu)^{B'}_{\ B}$.

[47]One can always find the respective group of pointed gauge transformations that acts freely on the space of field configurations. Its construction is completely analogous in non-Abelian YM. In all cases $\mathcal{G}_* \subset \mathcal{G}$ is a normal subgroup and $\mathcal{G}/\mathcal{G}_* \cong G$. (Analogous considerations hold in metric general relativity where "pointed diffeomorphisms" are diffeomorphisms that leave a point and a tangent space at that point invariant.) What distinguishes the Abelian case is that the group of pointed gauge transformations is isomorphic to the quotient $\mathcal{G}_A := \mathcal{G}/\mathcal{I}_A$ (for all $A$) which only in this case is a group itself.

section, we will argue that the choice of $\kappa$ is irrelevant—at least within a given superselection sector of $f$.

We now turn our attention to the geometry of field space, and to the definition of a functional connection form.

Recalling that the SdW boundary value problem has $\mathrm{Lie}(\mathcal{I}_{\mathrm{EM}})$ as a kernel, we define $\varpi_{\mathrm{EM}} \in \Omega^1(\mathcal{A}_{\mathrm{EM}}, \mathrm{Lie}(\mathcal{G}))$ as the unique solution to the SdW equation[48] that is valued in $\kappa(\mathrm{Lie}(\mathcal{G}_{\mathrm{EM}})) \subset \mathrm{Lie}(\mathcal{G})$. This connection satisfies the projection and covariance properties (8) with respect to gauge transformations in the image of $\kappa$:

$$\begin{cases} \mathbb{i}_{\xi_*^\sharp} \varpi_{\mathrm{EM}} = \xi_* \\ \mathbb{L}_{\xi_*^\sharp} \varpi_{\mathrm{EM}} = \mathbb{d}\xi_* \end{cases}, \qquad \forall \xi_* \in \kappa(\mathrm{Lie}(\mathcal{G}_{\mathrm{EM}})). \tag{84}$$

The above properties, however, "fail" if $\xi$ is replaced by an element of the stabilizer $\chi_{\mathrm{EM}} = const$:

$$\mathbb{i}_{\chi_{\mathrm{EM}}^\sharp} \varpi_{\mathrm{EM}} = 0 = \mathbb{L}_{\chi_{\mathrm{EM}}^\sharp} \varpi_{\mathrm{EM}}. \tag{85}$$

These two equations can be summarized in the following:

$$\begin{cases} \mathbb{i}_{\xi^\sharp} \varpi_{\mathrm{EM}} = \kappa(\xi) \\ \mathbb{L}_{\xi^\sharp} \varpi_{\mathrm{EM}} = \mathbb{d}\kappa(\xi) \end{cases}, \qquad \forall \xi \in \mathrm{Lie}(\mathcal{G}). \tag{86}$$

In $\mathcal{A}_{\mathrm{EM}}$, these equations are trivial since $\chi_{\mathrm{EM}}^\sharp \equiv 0$ in $\mathrm{T}\mathcal{A}$. But these equations can readily be interpreted within the phase space $\Phi_{\mathrm{EM}}$ as well. In this case $\varpi_{\mathrm{EM}}$ is as usual the pullback of the SdW connection defined over $\mathcal{A}_{\mathrm{EM}}$ and $\cdot^\sharp$ now includes the action of $\mathcal{G}$ on the electric field (which is also trivial) and on the matter fields $\psi$ (which is *non*trivial) (82).

Hence, over $\Phi_{\mathrm{EM}}$, the vector $\chi_{\mathrm{EM}}^\sharp$ does not vanish but is nonetheless in the kernel of (the pullback of) $\varpi_{\mathrm{EM}}$.

Thus, we see that we have geometrically isolated the set of "constant gauge transformations" of EM. Of course, these transformations are precisely those associated with the the total electric charge contained in $R$. In the following, we will rather call them *charge transformations*.

We can now define the horizontal derivatives as usual:

$$\mathbb{d}_\perp A := \mathbb{d}A - \mathrm{d}\varpi_{\mathrm{EM}} \qquad \mathbb{d}_\perp E := \mathbb{d}E \qquad \text{and} \qquad \mathbb{d}_\perp \psi := \mathbb{d}\psi + \varpi_{\mathrm{EM}}\psi, \tag{87}$$

with the understanding that—in the presence of the matter fields—full covariance is only guaranteed with respect to the gauge transformation *modulo* charge transformations. Also, notice how the horizontal derivative of $A$ could be unambiguously defined using a connection valued in $\mathrm{Lie}(\mathcal{G}_{\mathrm{EM}}) = \mathrm{Lie}(\mathcal{G})/\mathrm{Lie}(\mathcal{I}_{\mathrm{EM}})$, since $\mathrm{D}\chi_{\mathrm{EM}} \equiv 0$, but the horizontal derivative of $\psi$ requires a connection valued in $\mathrm{Lie}(\mathcal{G})$, since $\xi$ and $\xi + \chi_{\mathrm{EM}}$ act differently on $\psi$.

Let us analyze the properties of $\theta^\perp = \theta_{\mathrm{EM}}^\perp + \theta_{\mathrm{Dirac}}^\perp$ under gauge and charge transformations. We start by noticing that, contrary to the Noether charges $H^{\mathrm{EM}}[\xi_*] := \theta(\xi_*^\sharp) = \theta^V(\xi_*^\sharp)$, the stabilizer charges $Q_{\mathrm{EM}}[\chi_{\mathrm{EM}}]$ can be defined solely from the *horizontal* symplectic potential:[49]

$$0 \neq Q_{\mathrm{EM}}[\chi_{\mathrm{EM}}] := \int \sqrt{g}(\chi_{\mathrm{EM}}\rho) = \theta^\perp(\chi_{\mathrm{EM}}^\sharp). \tag{88}$$

---

[48]The SdW choice is here made for definiteness, but won't play any particular role in what follows. Other choices of connection can be studied e.g. by considering the corresponding vertical projector $\widehat{V}$ and thus defining $\varpi_{\mathrm{EM}} := \kappa \circ \iota^{-1}(\widehat{V}(\cdot)) \in \Omega^1(\mathcal{A}_{\mathrm{EM}}, \mathrm{Lie}(\mathcal{G}))$, where $\iota \equiv \kappa^\sharp : \mathrm{Lie}(\mathcal{G}_{\mathrm{EM}}) \to V_A \subset \mathrm{T}_A\mathcal{A}_{\mathrm{EM}}$ is the isomorphism between equivalence classes $[\xi + \chi_{\mathrm{EM}}] \in \mathrm{Lie}(\mathcal{G}_{\mathrm{EM}})$ and vertical vector fields in $\mathrm{T}\mathcal{A}_{\mathrm{EM}}$. The SdW connection considered in the text corresponds to the choice of $\widehat{V}$ as the $\mathbb{G}$-orthogonal vertical projector.

[49]Notice that, since $\varpi_{\mathrm{EM}}(\chi_{\mathrm{EM}}^\sharp) = 0$, $Q_{\mathrm{EM}}[\chi_{\mathrm{EM}}] := \theta^\perp(\chi_{\mathrm{EM}}^\sharp)$ is also equal to $\theta(\chi_{\mathrm{EM}}^\sharp)$, to $\theta_{\mathrm{Dirac}}(\chi_{\mathrm{EM}}^\sharp)$, and to $\theta_{\mathrm{Dirac}}^\perp(\chi_{\mathrm{EM}}^\sharp)$.

We notice that the Gauss constraint (53) together with the fact that $0 = \delta_{\chi_{\text{EM}}} A = \mathrm{d}\chi_{\text{EM}}$, implies the (integrated) Gauss law for the stabilizer charges—usually expressed for $\chi_{\text{EM}} = 1$:

$$Q_{\text{EM}}[\chi_{\text{EM}}] \approx \int \sqrt{g}\,(\chi_{\text{EM}} \nabla^i E_i^{\text{Coul}}) \approx \oint \sqrt{h}\,(\chi_{\text{EM}} f) = \chi_{\text{EM}} \oint \sqrt{h}\, f\,. \qquad (89)$$

If $\chi_{\text{EM}}$ is not only a constant in space but also in time, $Q_{\text{EM}}[\chi_{\text{EM}}]$ is a quantity satisfying a balance equation like (79) (electric charge conservation).

To understand the symplectic significance of the charge $Q_{\text{EM}}[\chi_{\text{EM}}]$, the following identity is important:[50]

$$\mathbb{L}_{\chi_{\text{EM}}^\sharp}\,\theta^\perp = \int \sqrt{g}\,(\rho\, \mathrm{d}\chi_{\text{EM}})\,. \qquad (90)$$

This identity establishes that $\theta^\perp$ is *not* invariant under the flow of a *charge* transformation $\chi_{\text{EM}}$, unless $\chi_{\text{EM}}$ is field-*in*dependent. This should be contrasted with the invariance of $\theta^\perp$ under gauge transformations proper (70). Before we make the comparison explicit, let us first follow the previous remark to its conclusions: the invariance of $\theta^\perp$ under the field-*in*dependent charge flow $\chi_{\text{EM}}^\sharp$, for $\mathrm{d}\chi_{\text{EM}} = 0$, implies the following *nontrivial* flow equation:

$$0 = \mathbb{L}_{\chi_{\text{EM}}^\sharp}\,\theta^\perp = \mathrm{d}Q_{\text{EM}}[\chi_{\text{EM}}] + \mathring{\mathbb{i}}_{\chi_{\text{EM}}^\sharp}\,\Omega^\perp \qquad (\mathrm{d}\chi_{\text{EM}} = 0)\,. \qquad (91)$$

Whereas the second equality is an identity that follows solely from Cartan's formula and the definition (88), we stress once again that, in the presence of matter (where the equation is nontrivial), the first equality and therefore the Hamiltonian-flow equation hold if and only if $\chi_{\text{EM}}$ is the *same* throughout $\mathcal{A}_{\text{EM}}$, i.e. only if $\mathrm{d}\chi_{\text{EM}} = 0$ (cf. (90)).

Heuristically, this makes sense: as defined here, a charge $Q_{\text{EM}}[\chi_{\text{EM}}]$ is a measure of a certain physical property of a matter distribution over a (symmetric) background, and the flow equation compares this measure at two neighbouring configurations—but, for this comparison to be meaningful, one cannot change the "measuring rod" (i.e. $\chi_{\text{EM}}$) from one configuration to the other.

Back to the comparison with the *trivial* "flow" equation for gauge-transformations proper. This is implied by (70) when $\mathcal{A}$ actually has the structure of a fibre bundle (so is a proper foliation induced by the action of the gauge group) and $\varpi$ satisfies the connection-form axioms (8) for *all* $\xi \in \text{Lie}(\mathcal{G})$ (as opposed to (85)). In EM, thanks to (84), it is *pointed* gauge transformations which satisfy $\mathbb{L}_{\xi_*^\sharp}\,\theta^\perp_{\text{EM}} \equiv 0$ and thus the trivial flow equation follows:[51]

$$0 \equiv \mathbb{L}_{\xi_*^\sharp}\,\theta^\perp = \mathrm{d}(\mathring{\mathbb{i}}_{\xi_*^\sharp}\,\theta^\perp) + \mathring{\mathbb{i}}_{(\xi - \kappa(\xi))^\sharp}\,\Omega^\perp\,. \qquad (92)$$

This "flow" equation is said to be trivial because each term on the rhs vanishes identically and independently, even when $\mathrm{d}\xi_* \neq 0$.

In sum, that such *charge* transformations $\chi_{\text{EM}}$ are physical, and are thus distinguished from *gauge* transformations $\xi_*$, is not postulated, but derived: the transformations corresponding to $\chi_{\text{EM}}$ are entirely generated by the $\theta^\perp_{\text{EM}}$ components, rather than by the Gauss constraint (which is entirely in $\theta^V_{\text{EM}}$), and thus survive the symmetry reduction process.

Therefore, the formal similarity between the on-shell stabilizer charges $Q_{\text{EM}}[\chi_{\text{EM}}] \approx \oint \sqrt{h}\,(\chi_{\text{EM}} f)$ and the on-shell Noether charge $H^{\text{EM}}_{\xi_*} \approx \oint \sqrt{h}\,(\xi_* f)$, should not obfuscate the

---

[50]This equation can be obtained by Lie-deriving $\theta^H$ (58) using $\mathbb{L}_{\mathbb{X}}\mathrm{d}\bullet = \mathrm{d}\mathbb{L}_{\mathbb{X}}\bullet$, the Leibniz rule, as well as the identities $\mathbb{L}_{\chi_{\text{EM}}^\sharp} A \equiv \delta_{\chi_{\text{EM}}} A = 0$ and (85) (which imply $\mathbb{L}_{\chi_{\text{EM}}^\sharp}\,\theta^H_{\text{EM}} = \mathbb{L}_{\chi_{\text{EM}}^\sharp}\,\theta^H_{\text{EM,Dirac}}$), and (81).

[51]More generally, the two equations above can be summarized in the following equality between the two expressions of $\mathbb{L}_{\xi^\sharp}\,\theta^\perp$:

$$Q_{\text{EM}}[\mathrm{d}\Xi] = \mathrm{d}Q_{\text{EM}}[\Xi] + \mathring{\mathbb{i}}_{\Xi^\sharp}\,\Omega^\perp \quad \text{where} \quad \Xi := \xi - \kappa(\xi) \in \text{Lie}(\mathcal{I}_{\text{EM}})\,.$$

important differences between these two very different quantities. The Noether charges $H_{\xi_*}^{\mathrm{EM}}$ should be thought of as encoding information on the $f$-superselection sector, rather than on the quasi-local radiative degrees of freedom contained in $R$.

**Remark**   Above we have chosen to represent $\mathcal{G}/\mathcal{I}_{\mathrm{EM}}$ in terms of $\mathcal{G}_{\mathrm{EM}}$. This choice is arbitrary not only in the choice of $x_*$, but also due to the fact that other ways exist of representing the quotient $\mathcal{G}/\mathcal{I}_{\mathrm{EM}}$ (e.g., for a cubic region, in terms of the Fourier modes of $\xi$ beyond the zero mode). Different such choices lead to different prescriptions for defining the SdW connection $\varpi_{\mathrm{EM}}$. Consider two such prescriptions, $\varpi_{\mathrm{EM},1}$ and $\varpi_{\mathrm{EM},2}$. Then, since $\varpi_{\mathrm{EM}}$ is itself exact, $\varpi_{\mathrm{EM},2} - \varpi_{\mathrm{EM},2} = \mathbb{d}\sigma$, for some $\sigma \in \Omega^0(\mathrm{Lie}(\mathcal{I}_{\mathrm{EM}}))$. From this, it is easy to see that $\Omega_2^\perp - \Omega_1^\perp = Q_{\mathrm{EM}}[\mathbb{d}\sigma] = \mathbb{d}\sigma \curlywedge \mathbb{d}q_R$ where $q_R = \int \sqrt{g}\rho$ is the total electric charge in $R$. Thus, the two reduced symplectic forms coincide within any given superselection sector (actually, even within larger sectors of constant $q_R$ rather than constant $f$).

## 4.5   Considerations on the non-Abelian generalization

How much of the constructions carried over in the previous section generalize to the non-Abelian case? We leave a comprehensive answer to this question for future work. Here, we limit ourselves to some general considerations on the difficulties one would encounter in this process of generalization.

As observed in the previous part of this section, we already have a candidate for a global set of charges in YM theory: this is the stabilizer charge $Q[\chi_A]$. What is in question is: Are these charges gauge-invariant? Are they the Hamiltonian generator of the charge transformation $\chi_A^\sharp$ for the *horizontal* symplectic structure?

In the rest of this section we will try to identify the difficulties one needs to face when addressing these questions. Albeit rarely, we will at times be compelled to make reference to notions—that we have not introduced—regarding the stratification of $\mathcal{A}$ by the reducible configurations, or the slice theorem [24, 25, 27, 28]. To make this article self-contained, we add a brief summary of these notions in appendix B.

**An example: the vacuum and constant gauge transformations**   Consider the non-Abelian vacuum configuration $A = 0$. Similarly to the EM case, the non-Abelian vacuum is also stabilized by constant gauge transformations, $\mathcal{I}_{A=0} \cong G$ and $\chi_{A=0} = const$; this might suggest that similar considerations to those made in the previous section about EM might be made for YM around the vacuum background $A = 0$. This would recover the notion of global charges proposed in [1, Ch. 7], who singles out "global gauge transformations" of this sort (i.e. $\xi = const$) as having a particular physical significance.

However, complications arise and this simple example is useful to exemplify some of the difficulties one encounters when attempting to generalize the constructions of the previous section to the non-Abelian case.

From a mathematical perspective, taking the directions $\chi_{A=0}^\sharp$ as physical also at non-vacuum configurations means modelling the space of physical configurations $\mathcal{A}/\mathcal{G}$ by the slice through the vacuum configuration[52] $\mathscr{S}_{A=0}$. The notion of "slice" generalizes the notion of "section" of a fibre bundle to the case in which reducible configurations are present. However, at reducible configurations, the notion of slice necessarily differs from the naive intuition behind the notion of section. In particular, the slice $\mathscr{S}_{A=0} \subset \mathcal{A}$ contains the non-trivial orbits of the $A \neq 0$ under the constant transformations $\chi_{A=0}^\sharp$ even if they contain a single representative

---

[52]Mutatis mutandis, any slice through a vacuum configuration $A = 0^g = g^{-1}\mathrm{d}g \in \mathcal{O}_{A=0}$ would do. Although a brief summary will follow, for a more through discussion of the notion of "slice" see appendix B and references therein.

of the (partial) orbit of "non-constant gauge transformations" $\mathcal{G}_{A=0} = \mathcal{G}/\mathcal{I}_{A=0}$ (which do not form a group). The stabilizer charges $Q[\chi_{A=0}]$ then emerge as the Noether charges (Noether I) associated with the now "frozen" background $A = 0$ and (fluctuation) fields that transform in the adjoint of $\mathcal{I}_{A=0} \cong G$. (This treatment has an analogue at all background configurations $\tilde{A}$ with nontrivial stabilizers. These analogue constructions, however, lead to different charge groups, $\mathcal{I}_{\tilde{A}}$).

There are however various reasons to deem this analysis incomplete. One such reason is the arbitrariness in the choice of symmetric background. Another is that, even granting that, in YM, the vacuum configurations in $\mathcal{O}_{A=0}$ *are* special, the use of $\mathscr{S}_{A=0}$ is at best perturbative around $A = 0$, as the following analogy highlights: the analogue in general relativity of using $\mathscr{S}_{A=0}$ as model for $\mathcal{A}/\mathcal{G}$ in YM would correspond to (somehow) choosing one set of e.g. Cartesian coordinates (adapted to the vacuum $g_{\mu\nu} = \eta_{\mu\nu}$) and declaring that translations and rotations with respect to these coordinates have physical significance also at non-flat configurations. Therefore, the analysis offered above can, at best, have an approximate significance in the presence of small perturbations on top of the vacuum background. (As already emphasized at the end of section 4.2, the YM stabilizer charges $Q[\chi]$ are indeed analogous to general relativity's Komar charges for backgrounds with a Killing symmetry; but whereas the physical importance of these charges over approximately symmetric backgrounds is manifest in the low-mass limit of general relativity, to the best of our knowledge it has not been established at all in (any regime of) YM.)

These observations suggest that it is not possible to arrive at a single notion of global charges in YM that is meaningful throughout $\mathcal{A}$. The alternative is to work, as in the rest of this paper, in a differential-geometric language at the level of local tangent spaces, considering only those stabilizer charges defined at one given configuration.

**Symmetry sectors** This takes us to the next complication. Since stabilizer charges exists only at reducible configurations which form a meagre set in $\mathcal{A}$, differentiating quantities such as $Q[\chi_A]$—e.g. to study the associated symplectic flows—is problematic: generic variations of the symmetric base configurations that are not fine-tuned necessarily take us to irreducible configurations that do not admit any stabilizer charge at all. This is another reason why, in the non-Abelian theory, the physical viability of stabilizer charges is unclear.

The above notwithstanding, it still is of mathematical interest to analyze the consequence of stabilizer charges *within* field-space sectors characterized by a certain degree of symmetry, i.e. within given strata $\mathcal{N}_A$ characterized by non-trivial (possibly sub-maximal) stabilizers, $\mathcal{I}_A \supsetneq \{\mathrm{id}\}$.

(One notable case in which focusing on reducible configurations is not physically restrictive is that of Yang-Mills fields at asymptotic null infinity: there, all physically admissible configurations must be—at leading order in $1/r$—in the vacuum configuration. This means that certain asymptotic stabilizers are intrinsically defined, and thus lead to an enlarged group of asymptotic symmetries, that correspond to Strominger's leading soft-charges. Our formalism extends to that context without obstructions: See [7] and references therein.)

Thus, within a single stratum, it is at least meaningful to attempt a generalization of the symplectic analysis we performed for the Abelian stabilizer charges. There are however further obstacles, which we will now highlight by inspecting the various ingredients entering (91) and (92). In the following, all differential operators must be understood as being those intrinsic to a given stratum $\mathcal{N}_A$.

**A $\varpi$ for reducible configurations?** The main tool we need to construct is a generalization of the connection form to a stratum $\mathcal{N}_A$ and thus of a horizontal differential. First, notice that the notion of horizontal differential is useful only if there are nontrivial horizontal directions

within $\mathcal{N}_A$, which is not the case in the bottom stratum $\mathcal{N}_{A=0}$ of the YM vacuum $A = 0$, since[53] $\mathcal{N}_A = \mathcal{O}_{A=0}$. Second, in the non-Abelian theory, $\mathfrak{G}_A = \mathrm{Lie}(\mathcal{G})/\mathrm{Lie}(\mathcal{I}_A)$ generically fails to be a Lie algebra, and so does $\kappa(\mathfrak{G}_A) \subset \mathrm{Lie}(\mathcal{G})$, for $\kappa$ (the non-Abelian analogue of) the embedding map given in (83). Therefore, on $\mathcal{N}_A$, it won't be possible to define an actual connection form that satisfies the covariance property as in (84), and its extension to $\Phi$ will lack a generalization of this property as in (86). In order to find a useful non-Abelian version of (86), it will therefore be important to find a set of definitions leading to a minimal modifications of the projection and covariance properties (8) in the presence of a nontrivial stabilizer group.

**A basic symplectic structure?**     Once a viable generalization of the projection and covariance properties has been found, and thus a $\varpi_{\mathrm{red}} \in \Omega^1(\mathcal{N}_A, \mathrm{Lie}(\mathcal{G}))$ has been defined, one will still have to check whether—through this $\varpi_{\mathrm{red}}$—it is possible to define an appropriately horizontal and gauge-invariant (i.e. basic) symplectic structure that can lead to the analogues of (91) and (92). We expect this will not be, strictly speaking, possible: depending on the specific way $\varpi_{\mathrm{red}}$ generalizes the projection and covariance properties, we expect certain obstructions to invariably appear. Ideally, these obstructions will be encoded in a certain combination of the stabilizer charges, rather than some new objects.

**Field-space constant charge transformations?**     Finally, an essential ingredient of the flow equation for the electric charges (91) is the condition $\mathbb{d}\chi_{\mathrm{EM}} = 0$. A similar condition will have to appear in the non-Abelian case too. It is thus instructive to discuss why its naive generalization, $\mathbb{d}\chi_A = 0$, cannot be correct and what kind of generalization might be available. The failure of the naive generalization follows from the fact that $\mathcal{I}_A$ is preserved throughout $\mathcal{N}_A$ only *up to* conjugation by elements of $\mathcal{G}$ (see appendix B). Since $\mathcal{I}_A$ necessarily changes vertically according to (73), an equation expressing (some form of) constancy of $\chi_A$ can only hold along horizontal directions. Indeed, it turns out that $\mathcal{I}_A$ *is* preserved in the directions that are $\mathbb{G}$-orthogonal to the orbits $\mathcal{O}_A$ (this fact is at the basis of most proves of the slice theorem, see footnote 89 in appendix B). If $\dim(\mathcal{I}_A) = 1$ or $\dim \mathscr{S}_A < 2$,[54] this observation would suffice to find a (weaker) version of the condition $\mathbb{d}\chi_{\mathrm{EM}} = 0$ that applies to the non-Abelian case. However, it is still found to be insufficient if $\dim(\mathcal{I}_A) = n > 1$ (and $\dim \mathscr{S}_A \geq 2$). This is because there is no canonical way to a priori identify *elements* of $\mathcal{I}_A$ and $\mathcal{I}_{A'}$ at two different configurations $A$ and $A'$, even when $\mathcal{I}_A = \mathcal{I}_{A'}$. Therefore, in general, we expect that it would be necessary to introduce a set of bases of $\{\chi_A^{(\ell)}\}_{\ell=1}^n$ of $\mathrm{Lie}(\mathcal{I}_A)$ and of a connection $\nu^\ell{}_{\ell'} \in \Omega^1(\mathcal{N}_A, \mathfrak{gl}(n))$. The curvature of this connection, if geometrically constrained, would provide yet another source of obstruction to the non-Abelian generalization of the flow equation (91).

We postpone any further analysis of these issues to future work.

## 5   The SdW connection from Dirac's dressing prescription

In this section, which is somewhat independent from the rest of the paper, we revisit Dirac's construction for the dressing of the electron [40], and provide considerations about its generalizability (or lack thereof) to the non-Abelian case. This discussion also offers an *independent*, albeit heuristic, route to the introduction of the SdW connection $\varpi_{\mathrm{SdW}}$.

It should be noted from the outset that, following Dirac, we will set up the problem in terms of a Coulombic potential since the very beginning (equation (74)). In the light of section 3, and of equation (63) in particular, it should then not come as a surprise that this will lead us precisely to the *SdW* connection. What *is* unexpected is that Dirac's construction will naturally

---

[53]Cf. footnote 88 in appendix B.

[54]Here we take the slice within each stratum, $\mathcal{N}_A$.

lead us to a field-space connection form at all, and thus to a notion of dressing that involves "field-space Wilson lines," which are field-space non-local objects. We will comment on this point at the end of this section.

The Dirac's dressing construction can be motivated by the need to define a *physical* electron field that is meant to correspond to creation operators associated to the "bare" electron and its Coulombic electric field at once, so that the Gauss law is automatically satisfied. With the purport of being physical, this dressed field is expected to be, and indeed is, gauge invariant. At the same time, however, the dressed field describes a charged electron, and therefore also carries electric charge. This means that the total electric charge Poisson-generates a global shift in the phase of the dressed electron field, which might seem in contrast with the posited gauge invariance of the dressed field. However, as we saw in section 4, these requirements are not mathematically in conflict: constant "gauge transformations" associated to the electric charge correspond to stabilizer transformations that do have a different (geometric) status in $\mathcal{A}_{\mathrm{EM}}$ with respect to generic (local) gauge transformations.

Starting from these ideas, we will now revisit Dirac's construction. We will work from the onset in finite regions and in the non-Abelian setting.

Denoting the dressed field with a hat, $\widehat{\psi}$, the classical condition corresponding to the demand that the corresponding quantum field creates an electron at $x$ together with its electrostatic field is

$$\{E_j^\beta(y), \widehat{\psi}(x)\} = -\left(D_j G_{\alpha,x}\right)^\beta(y) \tau_\alpha \widehat{\psi}(x), \tag{93}$$

where $\{\cdot, \cdot\}$ denotes the Poisson bracket and[55] $G_{\alpha,x} \in \Omega^0(R, \mathrm{Lie}(\mathcal{G}))$ is the $\mathrm{Lie}(\mathcal{G})$-valued Green's function of the SdW boundary value problem, as in (74)—which we report here for convenience:

$$\begin{cases} D^2 G_{\alpha,x}(y) = \tau_\alpha \delta_x(y), & \text{in } R, \\ D_s G_{\alpha,x}(y) = 0, & \text{at } \partial R. \end{cases} \tag{94}$$

Although at this level any other choice of boundary conditions would have worked, the (covariant) Neumann boundary condition is chosen here for future convenience. As observed in section 4.2, this choice of Green's function—valid only at non-reducible configurations[56]—corresponds to the demand that the dressed particle created at $x$ does *not* contribute to the flux $f$ at $\partial R$. This is consistent with the fact reviewed in 4.2 that at non-reducible configurations there is no meaningful integrated Gauss law. A posteriori, with the knowledge acquired from the construction of the SdW connection, it is possible to see that the boundary conditions of (94) are moreover the only ones that make $\widehat{\psi}$ gauge invariant with respect to gauge transformations whose support is not limited to the interior of $R$, extending also to its boundary $\partial R$.

Going back to the definition of the dressed matter field, and working formally ($\simeq$), we consider the Ansatz

$$\widehat{\psi}(x) \simeq e^{\phi[A](x)} \psi(x) \quad \text{for some} \quad \phi[A] \in \mathrm{Lie}(\mathcal{G}). \tag{95}$$

From this Ansatz, by substitution into the requirement (93), we get the condition:

$$\frac{1}{\sqrt{g}} \frac{\delta}{\delta A} \phi = \{E, \phi\} = -DG, \tag{96}$$

or, in full detail,

$$\frac{g_{ji}(y)}{\sqrt{g(y)}} \frac{\delta}{\delta A_i^\beta(y)} \phi^\alpha[A](x) = \{E_j^\beta(y), \phi^\alpha(x)\} = -\left(D_j G_{\alpha,x}\right)^\beta(y), \tag{97}$$

---

[56]See 4.2 for how the boundary value problem should be amended at reducible configurations.

where we used $\Omega(\frac{\delta}{\delta A_i^\beta(y)}) = -\sqrt{g(y)}\, g^{ij}(y)\delta E_j^\beta(y)$ to individuate the Hamiltonian vector field associated to $E_j^\beta(y)$ (notice that this expression is valid also in the presence of boundaries without the need of bulk-supported smearings).

Equation (93) in the form of (97) can then be formally solved through a line integral in configuration space $\mathcal{A}$, that we denote $\int\!\!\!\int^A$:

$$\phi[A]^\alpha(x) = -\int\!\!\!\int^A \int_R \mathrm{d}^D y \sqrt{g(y)}\, g^{ij} \sum_\beta (D_i G_{\alpha,x})^\beta(y) \, \mathbb{d}A_j^\beta(y). \tag{98}$$

Using a more compact notation, this can be written

$$\phi[A](x) = -\int\!\!\!\int^A \int_R \sqrt{g}\,\mathrm{Tr}\Big(D^i G_{\alpha,x}\,\mathbb{d}A_i^\beta\Big)\tau_\alpha. \tag{99}$$

Integrating by parts, one obtains

$$\phi[A](x) = -\int\!\!\!\int^A \left(-\int_R \sqrt{g}\,\mathrm{Tr}\Big(G_{x,\alpha}D^j\mathbb{d}A_j\Big)\tau_\alpha + \oint_{\partial R} \sqrt{h}\,\mathrm{Tr}\Big(G_{x,\alpha}\mathbb{d}A_s\Big)\tau_\alpha\right). \tag{100}$$

Now, to be able to use Green's theorem and simplify this expression, it is natural to introduce

$$\varpi \in \Omega^1(R, \mathrm{Lie}(\mathcal{G})), \tag{101}$$

defined by (8)

$$\begin{cases} D^2\varpi = D^i\mathbb{d}A_i, & \text{in } R, \\ D_s\varpi = \mathbb{d}A_s, & \text{at } \partial R. \end{cases} \tag{102}$$

Hence, using this definition and Green's theorem (76) for $\psi_1 = \varpi$ and $\psi_2 = G_{\alpha,x}$, we obtain the *formal* solution

$$\phi[A](x) = \int\!\!\!\int^A \varpi(x) \tag{103}$$

and $\widehat{\psi} \simeq \exp\left(\int\!\!\!\int^A \varpi\right)\psi$ is the formal general solution to the demands imposed by Dirac's dressing.

This construction provides an independent motivation for the introduction of the SdW connection form $\varpi$—even though at this level, its connection-form properties (8), and in particular its covariance property, are *not* manifest.

But with hindsight knowledge of the connection-form nature of $\varpi$, we introduce the following gauge-covariant expression (i.e. even under field-*dependent* gauge transformations) involving a *field-space Wilson-line*: we call this the dressing factor:[57]

$$\widehat{\psi}(x) = \underbrace{\mathbb{P}\mathbb{e}\mathbb{x}\mathbb{p}\left(\int\!\!\!\int^A \varpi(x)\right)}_{=:\text{dressing factor } e^{\phi[A]}}\psi(x) \tag{104}$$

(see below and especially [19, Sec. 9] for details and crucial subtleties regarding the dressing factor's gauge-covariance and the associated choice of field-space path).

---

[57]This is a 1-dimensional integral along a curve embedded in an infinite dimensional space. It is the latter property that the doublestruck-face of the symbol $\mathbb{P}\mathbb{e}\mathbb{x}\mathbb{p}\int$ is meant to emphasize. Cf. equation (152), where a Wilson line in space—rather than in configuration space—is considered.

In EM, the SdW connection is Abelian and flat (cf. theorem 2.19), i.e. $\varpi = \mathbb{d}\varsigma$ for

$$
\begin{cases}
\nabla^2 \varsigma = \mathrm{D}^i A_i, & \text{in } R, \\
\partial_s \varsigma = A_s, & \text{at } \partial R.
\end{cases}
\tag{105}
$$

Therefore, both the path ordering and the choice of path in $\mathcal{A}$ are inessential. In particular, one can choose the trivial configuration $A^\star = 0$ as a starting point for the field-space line integral so that the resulting expression (seemingly) depends only on the final configuration $A$. Indeed, using the fact that in EM the Green's function $G$ does not depend on $A$, we can perform the integral explicitly and readily find—cf. (104):

$$
\widehat{\psi}(x) = e^{\varsigma(x)}\psi(x) = e^{i\int_R \sqrt{g(y)}\, G(x,y)\partial^i A_i(y)}\psi(x). 
\tag{106}
$$

This provides a generalization to finite and bounded regions of the Dirac dressing, which in $\mathbb{R}^{D=3}$ with isotropic boundary conditions at infinity reads (see [40]):

$$
\widehat{\psi}(x) = e^{i\int \frac{\mathrm{d}^3 y}{4\pi} \frac{\partial^i A_i(y)}{|x-y|}}\psi(x).
\tag{107}
$$

In the non-Abelian setting, we first proposed the expression (104) (without reference to the derivation presented here) in [19]. There, this formula was framed in relation to the work on dressings by Lavelle and McMullan [41–44], and also to the Gribov-Zwanziger framework [63, 64] (see [65] for a review and relation to confinement), and, finally, to the geometric approach to the quantum effective action by Vilkovisky and DeWitt [23, 47–51]. In particular, in [19], we studied in detail the properties and limitations of (104) and we related the limitations to certain obstructions appearing in the previous works [23, 44, 47–49, 63, 64]. More specifically, we showed that: the obstructions found previously come from the curvature of the connection form, which induces a path-dependence ambiguity in the dressing; that this ambiguity can be fixed in a neighbourhood of a given reference configuration $A^\star$ (using field-space geodesics with respect to the so-called Vilkovisky connection [23, 48, 49]); and that, nonetheless, all expressions will still depend on the (gauge-dependent) choice of the reference configuration $A^\star \in \mathcal{A}$ [50, 51]. Finally, note that global existence and uniqueness of the Vilkovisky geodesics from $A^\star = 0$ to a generic $A$, a question related to the non-perturbative existence and uniqueness of the dressing factor, is expected to fail in view of the Gribov problem. We restrain from dissecting these topics here, and refer to [19, Sec. 9] for a thorough discussion.

Instead, we limit ourselves to the following observations: although the notion of a full-blown nonperturbative dressing is not viable in YM due to the involved geometry of $\mathcal{A}$, an infinitesimal version thereof is precisely provided by the SdW horizontal differential. Indeed, *formally*, the total differential of the (gauge invariant) dressed matter field, $\mathbb{d}\widehat{\psi}$ is directly related to the SdW horizontal differential of the bare matter field, modulo the dressing factor:[58]

$$
\mathbb{d}\widehat{\psi} \simeq e^{\phi}(\mathbb{d}\psi + \varpi\psi) = e^{\phi}\mathbb{d}_\perp \psi.
\tag{108}
$$

Since the SdW-horizontal differential has a natural place in any Abelian as well as non-Abelian YM theory, it follows that, in this sense, *the SdW horizontal differential constitutes the closest analogue to the Dirac dressing that generalizes to the non-Abelian YM theory*. In particular, the discussion of symmetry charges of section 4 shows that the dressed fields (or better, their

---

[58]Once again this can be made precise in the Abelian case where $\varpi = \mathbb{d}\varsigma$, where the following is an actual, i.e. not merely formal, equality (formally, $\phi = \varsigma - \varsigma_\star$—in the following we set $\varsigma_\star = 0$):

$$
\mathbb{d}\widehat{\psi} = e^{\varsigma}(\mathbb{d}\psi + \mathbb{d}\varsigma\,\psi) = e^{\varsigma}\mathbb{d}_\perp \psi.
$$

differentials) do carry charges despite being fully gauge invariant (resp. horizontal covariant) objects.

Although uncharged, the photon can also be made gauge invariant by dressing it with the same dressing factor. Not surprisingly this gives rise to the transverse photon. In the non-Abelian setting a dressed, covariantly-transverse, gluon can be defined with the same caveats as for the dressed quark and with a completely analogous relation to the horizontal differentials.

We conclude by directing the reader to [55, 66] for a more algebraic take on dressings and their consequences e.g. for the interpretation of spontaneous symmetry breaking, aka the Higgs mechanism. In relation to the Higgs mechanism within our formalism we refer to the field-space "Higgs connection" discussed in[59] [19, Sec. 7 and 9].

# 6 Gluing

So far we have analyzed the definition of quasilocal degrees of freedom and charges within a given region with boundaries. The goal of this section is to study how these notions behave with regards to the composition, or gluing, of regions.

The first result of this section concerns the gluing of field configurations or, more precisely, of horizontal field-space vectors—i.e. either horizontal perturbations of $A$ or radiative electric fields. The second result builds on the first and concerns the gluing of the symplectic structures.

In other words, we first show that—with knowledge of the choice of connection—two horizontal field configurations can be glued *un*ambiguously. Then, in section 6.7, we apply this result to the gluing of the symplectic structure. We will show that the horizontal symplectic structure do *not* factorize, even though the total symplectic structure can be unambiguously reconstructed from the regional ones. In doing so we will precisely identify *what* the new dof emerging upon gluing are.

These results apply, strictly speaking, only when considering the gluing of (trivial bundles over) simply connected regions into a larger (trivial bundle over a) simply connected region. That is, we neglect topological effects, such as the emergence upon gluing of new Aharonov–Bohm dof.

This possibility is briefly discussed in the simplest possible context in section 6.8.

Nonetheless, despite the simplified context, this result is nontrivial: new dof *do* emerge upon gluing, and can be *uniquely* identified. The fact that new dof emerge upon gluing will not surprise those whose intuition is built through lattice considerations. However, the fact that these dof can be *uniquely* identified, i.e. that no gauge "slippage" is allowed at the interface, might defy their intuition. For this reason we will sidestep the problem of providing a thorough set of conditions for the existence part of the problem of (smooth) gluing.[60]

In this topologically trivial context we will also discuss how the presence of matter fields (on top of regionally-reducible configurations) can introduce ambiguities in the gluing. We relate these ambiguities to 't Hooft's beam splitter thought experiment and to the concept of

---

[59]See in particular *ibid.* point (*x*) of Sec. 9.2.

[60]A thorough discussion of this problem should be set up along the following lines: an appropriate functional space (e.g. the space of smooth functions, or a certain Sobolev space) must be chosen for the configurations $A \in \mathcal{A}$ and their fluctuations $\mathbb{X} \in T\mathcal{A}$ (more generally, analogous choices must be made also for the electric and matter fields). Once this space is given, the existence within the same functional space (restricted over $R^{\pm}$) of the regional horizontal projections $h^{\pm}$ according to the SdW boundary value problem must be checked. Finally, conditions on the regional horizontal projections must be provided so that the resulting glued, global, and horizontal fluctuation $H = H(h^+, h^-)$ also belongs to the originally chosen functional space. If the functional space of choice is the space of smooth functions, $C^{\infty}$, then the main difficulty is ensuring that the glued, global, and horizontal fluctuation $H$ is smooth across $S$. Cf. the next section for the notation.

Direct Empirical Significance for gauge symmetries (DES).

## 6.1 Mathematical statement of gluing

In the following, we will first state and then prove the theorem at the root of all our results on the composition of both the electric field $E$ and of the perturbations $\mathbb{X} \in T\mathcal{A}$ of the gauge potential $A$.

Here we are not interested in global, topological, dof, and will focus on a boundary-less, self-contained model of the universe as a whole. Therefore, we consider for simplicity a global region $\Sigma \cong S^D$, $\partial \Sigma = \emptyset$, which is split into two hemispheres, $R^{\pm} \cong \mathbb{B}^D$ (the $D$-dimensional ball), such that $R^+ \cap R^- = S$ coincides with the equator up to orientations, $S = \pm \partial R^{\pm} \cong S^{D-1}$. I.e.

$$\Sigma = R^+ \cup_S R^- . \tag{109}$$

The two hemispheres serve also as charts over $\Sigma$. Since we are not interested in studying topological dof, we consider the transition function for the gluing of the $A$'s across $S$ to be fixed. For simplicity we will fix this transition function as the identity, which will allow us to introduce quantities corresponding to global electric fields and global perturbations of the gauge potentials. At the end of section 6.1.2, we will comment on the extension to the case of $\partial \Sigma \neq \emptyset$.

To formally encode the separation of regions, we introduce $\Theta_{\pm}$ as the characteristic functions of $R^{\pm}$. Denoting $s_i$ the outgoing co-normal at $S$ with respect to the region $R^+$, one has

$$\partial_i \Theta_{\pm} = \mp s_i \delta_S , \tag{110}$$

where $\delta_S$ is a $(D-1)$-dimensional delta function supported on the interface $S$.

Having assumed a trivial bundle over $\Sigma$, a generic Lie-algebra-valued vector in $T\mathcal{A}$ can be written as $\mathbb{Y} = \int_{\Sigma} Y \frac{\delta}{\delta A} \in T\mathcal{A}$. It is useful to introduce the following notation for the regional decomposition of $\mathbb{Y}$ supported on $\Sigma$:

$$\mathbb{Y} = \mathbb{Y}^+ \oplus \mathbb{Y}^- , \tag{111}$$

where $Y^{\pm} := Y\Theta_{\pm}$ and $\mathbb{Y}^{\pm} = \int_{R^{\pm}} Y^{\pm} \frac{\delta}{\delta A}$.

With this notation, notation we can state the following (see figure 6):

**Theorem 6.1** (General Gluing Theorem). *Given $\Sigma = R^+ \cup_S R^-$ as above, and given $\mathbb{Y} = \mathbb{Y}^+ \oplus \mathbb{Y}^-$ as above; consider three field-space connections $(\varpi, \varpi_{\pm})$ each associated to $\Sigma$ and $R^{\pm}$ respectively, defining the three horizontal/vertical decompositions*

$$Y = H + \mathrm{D}\Lambda \qquad and \qquad Y^{\pm} = h^{\pm} + \mathrm{D}\lambda^{\pm} , \tag{112}$$

*where $\Lambda := \varpi(\mathbb{Y})$ and $\lambda^{\pm} := \varpi_{\pm}(\mathbb{Y}^{\pm})$, and $\varpi(\mathbb{H}) = 0 = \varpi_{\pm}(\mathbb{h}^{\pm})$. Then, formally, $H$ is uniquely determined by $h^{\pm}$.*

Notice that $h^{\pm} \neq H\Theta_{\pm}$ and $\lambda^{\pm} \neq \Lambda\Theta_{\pm}$, i.e. that *regional restrictions and horizontal projections fail to commute*. This is a consequence of the nonlocality of the functional connections $(\varpi, \varpi_{\pm})$ and the reason why the Gluing Theorem is nontrivial.

Under the hypotheses of the theorem, the "commutator" between these two operations is provided by the regional vertical adjustments $\xi^{\pm} := \lambda^{\pm} - \Lambda\Theta_{\pm}$:

$$H(h^+, h^-) = (h^+ + \mathrm{D}\xi^+)\Theta_+ + (h^- + \mathrm{D}\xi^-)\Theta_- . \tag{113}$$

The precise form of these vertical adjustments depends on the specific choice of connections $(\varpi, \varpi_{\pm})$. If all three connections are SdW connections, then the $\xi^{\pm}$'s can be determined through explicit formulas. Indeed, as it turns out, the General Gluing Theorem will be proven in the following section as an immediate consequence of the analogous statement for the SdW decompositions, the SdW Gluing Lemma:

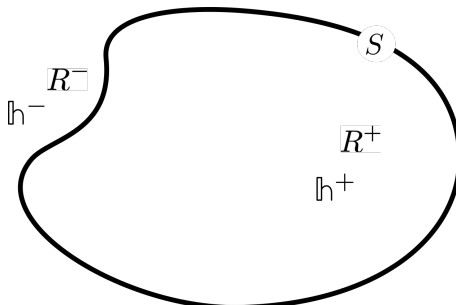

Figure 6: The two subregions of $\Sigma$, i.e. $\Sigma^{\pm}$, with the respective horizontal perturbations $\mathbb{h}^{\pm}$ on each side, along with the separating surface $S$.

**Lemma 6.2** (SdW Gluing Lemma). *Consider the premises of the General Gluing Theorem 6.1 in the case where $(\varpi, \varpi_{\pm})$ are all SdW connections, so that*

$$\mathrm{D}^i H_i = 0 \qquad and \qquad \begin{cases} \mathrm{D}^i h_i^{\pm} = 0, & in\ R^{\pm} \\ s^i h_i^{\pm} = 0, & at\ S \end{cases} \qquad (SdW). \qquad (114)$$

*Assume also that $A$, $A^{\pm} := A\Theta_{\pm}$ and ${}^S A := \iota_S^* A$ are irreducible as gauge potentials over $\Sigma$, $R^{\pm}$ and $S$ respectively.*

*Then, if $H$ is $C^1$ across $S$, the vertical-adjustments $\xi^{\pm} = \xi^{\pm}(h^+, h^-) \in \Omega^0(R^{\pm}, \mathrm{Lie}(G))$ of formula (113) are uniquely determined by the regional SdW value problems*

$$\begin{cases} \mathrm{D}^2 \xi^{\pm} = 0, & in\ R^{\pm} \\ \mathrm{D}_s \xi^{\pm} = \Pi, & at\ S \end{cases} \qquad (SdW), \qquad (115)$$

*for $\Pi \in \Omega^0(S, \mathrm{Lie}(G))$ given by*

$$\Pi = -\left(\mathcal{R}_+^{-1} + \mathcal{R}_-^{-1}\right)^{-1} \mu \qquad (116)$$

*and $\mu \in \Omega^0(S, \mathrm{Lie}(G))$ determined by the following SdW boundary value problem intrinsic to the boundary $S$ ($\partial S = \emptyset$):*

$${}^S \mathrm{D}^2 \mu = {}^S \mathrm{D}^a \iota_S^* (h^+ - h^-)_a. \qquad (117)$$

*Here, ${}^S \mathrm{D}_a := (\iota_S^* \mathrm{D})_a$ is the covariant derivative intrinsic to $S$ and ${}^S \mathrm{D}^2 = h^{ab}\ {}^S \mathrm{D}_a\ {}^S \mathrm{D}_b$ is the covariant Laplace operator on $S$, with $h_{ab} = (\iota_S^* g)_{ab}$ the induced metric there. Finally the operators $\mathcal{R}_{\pm}$ appearing in (116) are the 'generalized Dirichlet-to-Neumann pseudo-differential operators' attached to $S$, but geometrically associated to each region (see the next section for the precise definitions of these operators, equation (128)).*

Notice that *neither* the General Gluing Theorem *nor* the SdW Gluing Lemma will attempt an in-depth analysis of the conditions that, from smooth $h^{\pm}$, allow us to reconstruct a global $H$ which is smooth across the interface $S$. Indeed, as emphasized above, our main interest is to prove that the gluing procedure is—whenever meaningful—fully unambiguous (at irreducible configurations and modulo topological ambiguities).

We make the following remark on the conditions of Lemma 6.2 in the *non-Abelian case*: as emphasized by our discussion of global charges, section 4.5, the generic and physically most natural condition is subsumed by an irreducible *bulk* configuration. Then, if the bulk configuration is irreducible, the requirement of irreducibility for ${}^S A = \iota_S^* A$ is also open: a choice of $S$ such that ${}^S A$ is irreducible always exists, and perturbations in the position of $S$ preserve this property. In the Abelian case, on the other hand, all configurations are reducible by constant "gauge transformations." This reducibility moreover carries physical significance,

since it is—in all cases—related to the total charge contained in the region. In section 6.4 we will explore the physical meaning of a *bulk* stabilizer ambiguity due to a reducible bulk configuration $A$ in the presence of matter.

**Road map of the proof**   Let us chart a roadmap of the proof of the two propositions above, which will be given in greater detail in the next section. The proof consists of four steps. The first three focus on reconstructing the $\xi^\pm$ in the SdW case; the last one bootstraps the general case from the SdW case:

1. First, assuming that a global $H$ which is at least $C^1$ at $S$ exists, we will deduce restrictions on the difference $h_+ - h_-$ at $S$. Combined with the horizontality of the regional $h^\pm$, the requirement of smoothness gives us conditions on the longitudinal and transverse derivatives of $(\xi^+ - \xi^-)$ at the boundary:

   (a) In the absence of boundary stabilizers, the longitudinal condition allows us to solve for the difference

   $$\mu := -(\xi^+ - \xi^-)_{|S}\,, \tag{118}$$

   in terms of the interface *mismatch* $(h^+ - h^-)_{|S}$, which is parallel to the boundary due to (114). This leads to equation (117).

   (b) The transverse condition states the equality of the derivatives normal to the boundary, allowing us to introduce

   $$\Pi := D_s\xi^+_{|S} = D_s\xi^-_{|S} \qquad \text{(SdW)}\,. \tag{119}$$

2. Second, the SdW horizontality of the global $H$ provides us with one extra condition on the bulk part of the $\xi^\pm$'s, stating that the $\xi^\pm$ must be (covariantly) harmonic in their own regional domains. Together with (119), this leads to the SdW boundary value problem for the $\xi^\pm$'s (115).

3. Third, we show that an equation fixing $\Pi$ in terms of $\mu$ exists (116), thus allowing us to prove that the information above suffices to uniquely and fully reconstruct the $\xi^\pm$ in terms of $h^+$ and $h^-$. The relationship between $\Pi$ and $\mu$ can be loosely understood as a conversion of Dirichlet ($\mu$) to Neumann ($\Pi$) boundary conditions for the $\xi^\pm$'s entering the SdW boundary value problem (115). This is why (116) involves a combination of generalized Dirichlet-to-Neumann pseudo-differential operators $\mathcal{R}_\pm$ attached to the boundary (but geometrically associated to each region). The details on the nature of these operators are postponed to the next section.

4. Finally, we show that the proof of the general case can always be reduced to the proof of the SdW case, thus showing that the General Gluing Theorem follows from the SdW Gluing Lemma.

The next two sections are devoted to the proof (and explanation) of the above formulas.

### 6.1.1  Formal proof of the SdW Gluing Lemma 6.2

*Proof.* Assuming a global $H$ which is $C^1$ across $\Sigma$ exists, from (113) we deduce that the following relation must hold at the interface:

$$(h_i^+ - h_i^-)_{|S} = -D_i(\xi^+ - \xi^-)_{|S}\,. \tag{120}$$

This equation not only imposes a series of conditions on our unknown $\xi^\pm$, but also demands the interface mismatch of our variables $h_i^\pm$ to be of a pure-gradient form. This condition is

restrictive and will be discussed in more detail in section 6.2. For now, we take it for granted and focus on the consequences of this equation on $\xi^{\pm}$.

We now decompose (120) into its transverse and longitudinal components with respect to $S$. Since the component of $h^{\pm}$ transverse to $S$ vanishes because of regional horizontality (114), contracting (120) with $s^i$ we obtain that the normal derivatives of $\xi^{\pm}$ at $S$ must match

$$\mathrm{D}_s(\xi^+ - \xi^-)_{|S} = 0. \tag{121}$$

Therefore, taking the boundary divergence of the pullback of (120) to $S$ (i.e. effectively contracting its pullback with ${}^S\mathrm{D}^a$) we find that the difference

$$\mu := -(\xi^+ - \xi^-)_{|S} \tag{122}$$

is solely determined, in the absence of stabilizers $\chi_S$ of ${}^SA = \iota_S^* A$, by the mismatch of the two horizontals at the boundary according to the following SdW boundary value problem *intrinsic to the boundary $S$ ($\partial S = \emptyset$)*:

$$ {}^S\mathrm{D}^2\mu = {}^S\mathrm{D}^a \iota_S^*(h^+ - h^-)_a. \tag{123}$$

Assuming that ${}^SA = \iota_S^* A$ is boundary-*irreducible*, this equation has a unique solution and this concludes step 1 of the proof outlined above.

Now we move to step 2: assuming that the global region $\Sigma$ has no boundaries, by smearing the global horizontality condition, $\mathrm{D}^i H_i = 0$, with $H$ given by (113), we obtain:

$$\int_\Sigma \mathrm{Tr}\Big[\sigma\,\mathrm{D}^i\big((h_i^+ + \mathrm{D}_i\xi^+)\Theta_+ + (h_i^- + \mathrm{D}_i\xi^-)\Theta_-\big)\Big] = 0, \tag{124}$$

for any smooth test function $\sigma \in C^\infty(\Sigma, \mathrm{Lie}(G))$. Now, thanks to the identity $\partial_i\Theta_\pm = \mp s_i\delta_S$ (110) and to the regional horizontality conditions $\mathrm{D}^i h_i^\pm = 0 = s^i h_i^\pm{}_{|S}$ (114), we get:

$$\int_{R^+} \mathrm{Tr}\big(\sigma\,\mathrm{D}^2\xi^+\big) + \int_{R^-} \mathrm{Tr}\big(\sigma\,\mathrm{D}^2\xi^-\big) - \oint_S \mathrm{Tr}\big(\sigma\,s^i\mathrm{D}_i\big(\xi^+ - \xi^-\big)\big) = 0, \tag{125}$$

where the last term above already vanishes due to (121). From the arbitrariness of $\sigma$, we obtain the last, bulk, condition mentioned in step 2 of the outline above.

We thus deduce that, if a global $H \in C^1(\Sigma, \mathrm{Lie}(\mathcal{G}))$ exists, then the $\xi^{\pm}$ must satisfy the following elliptic boundary value problem

$$\begin{cases} \mathrm{D}^2\xi^\pm = 0, & \text{in } R^\pm, \\ s^i\mathrm{D}_i\big(\xi^+ - \xi^-\big) = 0, & \text{at } S, \\ (\xi^+ - \xi^-) = -\mu(h^+, h^-), & \text{at } S, \end{cases} \tag{126}$$

where $\mu(h^+, h^-)$ is the unique solution to (123). This concludes step 2. Now we must use the appropriate PDE tools to show that this boundary value problem determines $\xi^{\pm}$ in terms of the regional horizontal perturbations $\mathbb{h}^{\pm}$. This is step 3.

For step 3, we proceed as follows: start by setting

$$\Pi := s^i(\mathrm{D}_i\xi^\pm)_{|S}, \tag{127}$$

from the second equation in (126). Note that in possession of $\Pi$, we can determine $\xi^{\pm}$ by solving the boundary value problem given by (127) and the first equation of (126). Then step 3 can be reformulated as the problem of fixing $\Pi$ in terms of $\mu(h^+, h^-)$.

Notice that, given $\Pi$, $\xi^{\pm}$ are uniquely determined up to stabilizers, i.e. up to elements $\chi^{\pm} \in C^\infty(\Sigma, \mathrm{Lie}(G))$ such that $\mathrm{D}_i\chi^{\pm} = 0$ which are nontrivial only at reducible configurations.

In the topologically simple case that we have analyzed so far, this is the only ambiguity present in the determination of $\xi^\pm$. We postpone the discussion of reducible configurations and of nontrivial topologies until sections 6.4 and 6.8, respectively.

Now, to determine $\Pi$, we introduce generalized *Dirichlet-to-Neumann operators* $\mathcal{R}_\pm$ (see e.g. [67] and references therein). In each region, such operators map Dirichlet conditions for a (gauge-covariantly) harmonic function to the corresponding (gauge-covariant) Neumann conditions. In brief, for a given bounded region, $\mathcal{R}$ functions as follows: a given harmonic function with Dirichlet conditions—these conditions are the input of $\mathcal{R}$—will possess a certain normal derivative at the boundary; i.e. will induce certain Neumann conditions there—these conditions are the output of $\mathcal{R}$. But let us be more explicit.

In general, for a manifold with boundary $S$ and outgoing normal $s^i$, we define the Dirichlet-to-Neumann operator $\mathcal{R} \in \mathrm{Aut}(\Omega^0(S, \mathrm{Lie}(\mathcal{G})))$ by

$$\mathcal{R}(u) := s^i \mathrm{D}_i(\zeta_u)_{|S}, \tag{128}$$

where $\zeta_u$ is the unique (gauge-covariantly) harmonic Lie-algebra-valued function defined by the elliptic Dirichlet boundary value problem: $\mathrm{D}^2 \zeta_u = 0$ with $(\zeta_u)_{|S} = u$. Notice that the subscript $u$ encodes the *Dirichlet* boundary condition employed. Using superscripts to denote (gauge-covariant) Neumann boundary conditions, we would have by definition $\zeta^{\mathcal{R}(u)} \equiv \zeta_u$. Moreover, since (in the absence of stabilizers) the corresponding Neumann problems also have unique solutions, $\mathcal{R}$ is invertible, i.e. $\zeta^\Pi \equiv \zeta_{\mathcal{R}^{-1}(\Pi)}$. At irreducible configurations, we can thus define $\mathcal{R}_\pm$ associated to $R^\pm$ with boundaries $\partial R^\pm = S$, and their inverses $\mathcal{R}_\pm^{-1}$.

Now, from (127) and the fact that $\xi$ is itself (gauge-covariantly) harmonic from the first equation of (126), we have

$$\xi^\pm = {}^{(\pm)}\zeta^{\pm\Pi} \equiv {}^{(\pm)}\zeta_{\mathcal{R}_\pm^{-1}(\pm\Pi)}, \tag{129}$$

where the back-superscript $(\pm)$ indicates whether the respective covariantly harmonic functions ${}^{(\pm)}\zeta$ are defined over $R^+$ or $R^-$, respectively. We will now use the last equation of (126) to fix $\Pi$ uniquely. Once this is done, (129) contains all the information we sought for the gluing.

Notice that there is a $\pm$ sign in the argument of $\mathcal{R}_\pm^{-1}$ in (129). This sign is due to the fact that, at $S$, $s^i \mathrm{D}_i \xi^+ = s^i \mathrm{D}_i \xi^-$ but $s^i$ is the outgoing normal on one side and the ingoing normal on the other, so the conditions $s^i \mathrm{D}_i \xi^\pm = \Pi$ fix opposite Neumann conditions on the two sides. By the linearity of $\mathcal{R}$ we have

$$\mathcal{R}_\pm^{-1}(\pm\Pi) = \pm\mathcal{R}_\pm^{-1}(\Pi). \tag{130}$$

Hence, since by defintion $(\zeta_u)_{|S} = u$, together with (129) and (130) we have

$$(\xi^+ - \xi^-)_{|S} = \mathcal{R}_+^{-1}(\Pi) - \mathcal{R}_-^{-1}(-\Pi) = \left(\mathcal{R}_+^{-1} + \mathcal{R}_-^{-1}\right)(\Pi). \tag{131}$$

This gives us a relation between the (gauge-covariant) Neumann boundary condition $\Pi$ and the difference of the Dirichlet boundary conditions $\xi_{\pm|S}$.

This relation finally allows us to provide a formula that fixes $\Pi$ in terms of the boundary discrepancy of the regional horizontals $(h^+ - h^-)_{|S}$. That is, we insert (131) into the last of the equations (126) to obtain:

$$\left(\mathcal{R}_+^{-1} + \mathcal{R}_-^{-1}\right)(\Pi) = -\mu(h^+, h^-). \tag{132}$$

This is the equation that fixes $\Pi$ in terms of $\mu(h^+, h^-)$. Since its solution is unique—as we will discuss in a moment—it also fixes $\xi^\pm$ uniquely through (129), thus subsuming the entire set of equations (126). This concludes step three.

For the uniqueness statement for $\Pi$ to be meaningful, it is important to check that the operator $(\mathcal{R}_+^{-1} + \mathcal{R}_-^{-1})$ is invertible. That this is (formally) the case follows from $\mathcal{R}_\pm$ being

positive self-adjoint operators, and from the relative sign appearing on the left-hand-side of (132)—a consequence of the sign in (130).

To show that the generalized Dirichlet-to-Neumann operators $\mathcal{R}_{\pm}$ are self-adjoint and have positive spectrum we proceed as follows. Consider again $\zeta_u \neq 0$ to be the unique solution to the problem $D^2\zeta_u = 0$ in the bulk and $(\zeta_u)_{|S} = u$ at the boundary. Then, for any Lie-algebra valued functions $u, v$ on the boundary, one has

$$
\int_{R^+} \sqrt{g}\, g^{ij} \mathrm{Tr}(D_i \zeta_u D_j \zeta_v) = -\int_{R^+} \sqrt{g}\, \mathrm{Tr}(\zeta_u D^2 \zeta_v) + \oint_S \sqrt{h}\, s^i \mathrm{Tr}(\zeta_u D_i \zeta_v)
$$

$$
= \oint_S \sqrt{h}\, \mathrm{Tr}(u \mathcal{R}_+(v)) = \oint_S \sqrt{h}\, \mathrm{Tr}(\mathcal{R}_+(u) v). \tag{133}
$$

Notice that the first step in (133) follows from an integration by parts and properties of the commutator under the trace (i.e. from the ad-invariance of the Killing form).[61] The last line of (133) proves the self-adjointness of $\mathcal{R}_+$ with respect to the natural inner product $\langle u, v \rangle_S = \oint_S \sqrt{h}\, \mathrm{Tr}(uv)$, while setting $u = v$ in (133), gives positivity:

$$
\oint_S \sqrt{h}\, \mathrm{Tr}(u \mathcal{R}_+(u)) \geq 0. \tag{134}
$$

At *irreducible* configurations, the equality holds if and only if $\zeta_u = 0$ and therefore if and only if $u = 0$. Similar manipulations lead to the analogous conclusion for $\mathcal{R}_-$.

We have thus showed that: *if a global SdW horizontal $H \in C^1(\Sigma, \mathrm{Lie}(\mathcal{G}))$ exists, then it is uniquely determined by the SdW regional horizontals $h^{\pm}$ via* (113), (115) and (116). This concludes the proof of the SdW Gluing Lemma 6.2. □

**Summary**  Here is what our gluing theorem means, in words. In a given region, say $R_+$, the vertical $\xi_+$ which translates between the global and regional horizontals, $H_{|R_+} = h_+ + D\xi_+$, is defined as a harmonic function with Neumann boundary conditions (with respect to the *covariant* differential operator D). The Neumann conditions are implicitly defined by the difference of horizontals at the boundary, but since this difference would only give a Dirichlet boundary condition, one must apply the Dirichlet-to-Neumann boundary operator $\mathcal{R}_+$. Nonetheless, we can summarize: $\xi_{\pm}$ are the unique harmonic functions with Neumann conditions defined by the difference of horizontals at the boundary. Each such doublet will identify a unique global horizontal $H$ compatible with the doublet of horizontals, $(h_+, h_-)$.

In the next section we show that the SdW connection is just a crutch: the gluing theorem holds more generally.

### 6.1.2  Proof of the General Gluing Theorem 6.1

The proof of SdW Gluing Lemma of course relies on the particular choice of the SdW connection. But as long as there exists a 1-1 correspondence between the horizontal vectors of one connection to the horizontal vectors of another, uniqueness will go through.

To avoid confusions, in this section we denote the global and regional SdW connections as $(\varpi_{\mathrm{SdW}}, \varpi^{\pm}_{\mathrm{SdW}})$ and the arbitrary connections of the statement of the General Gluing Theorem

---

[61]The following identity is valid for any smearing $\sigma \in C^{\infty}(\Sigma, \mathrm{Lie}(G))$:

$$
\mathrm{Tr}\Big(-\sigma\, \partial^i D_i \zeta + g^{ij}[A_i, \sigma] D_j \zeta\Big) = \mathrm{Tr}\Big(-\sigma\, \partial^i D_i \zeta - g^{ij}\sigma[A_i, D_j \zeta]\Big) = \mathrm{Tr}\Big(-\sigma D^2 \zeta\Big).
$$

6.1 as $(\varpi', \varpi'_\pm)$ so that the corresponding horizontal/vertical decompositions are also primed. Unprimed decompositions refer to the SdW connection.

Let us emphasize once again that the three connections $(\varpi', \varpi'_\pm)$ can be completely unrelated: unlike $\varpi_{SdW}$ and $\varpi^\pm_{SdW}$, they might not all descend from the same geometric criterion.

Now, according to the primed horizontal/vertical decomposition, equation (111) stays the same, whereas (112) and (113) are rewritten with primes. E.g.

$$H' = (h'_+ + D\xi'_+)\Theta_+ + (h'_- + D\xi'_-)\Theta_-. \tag{135}$$

Our goal is to show that, given $h'_\pm$, then $H'$ is uniquely determined. We start by SdW-decomposing $h'_\pm$, thus obtaining:

$$h'_\pm = h_\pm + D\lambda_\pm, \tag{136}$$

where $\lambda_\pm := \varpi^\pm_{SdW}(\mathbb{h}'_\pm)$ and $\varpi^\pm_{SdW}(\mathbb{h}_\pm) = 0$. Now, from the SdW Gluing Lemma, we formally compute the unique SdW-horizontal $H$ such that

$$H = (h_+ + D\xi_+)\Theta_+ + (h_- + D\xi_-)\Theta_-; \tag{137}$$

here, $\varpi_{SdW}(\mathbb{H}) = 0$ and the $\xi_\pm = \xi_\pm(h^+, h^-)$ are given by the SdW Gluing Lemma. Now, decomposing $H$ according to $\varpi'$ we obtain:

$$H = H' + D\Lambda', \tag{138}$$

where $\Lambda' := \varpi'(\mathbb{H})$. Hence, combing all formulas together, we find that the $\xi'_\pm$'s of equation (135) are given by:

$$\xi'_\pm = \xi_+ - \lambda_+ - \Lambda'\Theta_\pm. \tag{139}$$

Therefore, if $H'$ is to be a horizontal field according to $\varpi'$, we can find unique vertical adjustments $\xi'_\pm$ to (135). This concludes the formal proof of General Gluing Theorem.

Finally, let us comment on the role of the condition $\partial\Sigma = \emptyset$. This condition ensures that the only boundary of $R^\pm$ is the interface $S = R^+ \cap R^-$. Relaxing this condition by e.g. introducing "radial gluing" of spherical shells introduces only minor variations on the above construction. This is the case unless boundaries intersect at corners, as e.g. in the case of two topological balls glued to form a larger ball. This is most clearly highlighted from the perspective of the SdW Gluing Lemma, in which case one must require further (corner) boundary conditions for the equation determining the mismatch $\mu := -(\xi^+ - \xi^-)_{|S}$. We will not attempt an analysis of this situation here beyond the preliminary observations offered at the end of the next section.

## 6.2 Continuity at $S$: towards a dimensional tower of conditions on $h^\pm$

Recall that, whereas the normal component of the continuity condition for $H$ (120) is a condition on the $(\xi^+ - \xi^-)_{|S}$ only, its parallel component to $S$ not only encodes a relation between $(\xi^+ - \xi^-)_{|S}$ and $(h^+ - h^-)_{|S}$, but also requires $(h^+ - h^-)_{|S}$ to be a pure gradient parallel to $S$. This is a necessary and sufficient condition on $h^\pm$ for there to exist a continuous global horizontal field $H$ corresponding to their composition.

In this section we will discuss a more constructive procedure to understand this condition on $(h^+ - h^-)_{|S}$. This procedure can be iteratively applied to the "boundaries of the boundaries", opening a door to the discussion of the more general gluing schemes involving corners.

In a gauge theory, the space of the pullbacks to $S$ of the fields in $\mathcal{A}$ defines a new "boundary configuration space", ${}^S\mathcal{A}$ which is isomorphic to the space of gauge fields intrinsic to $S$:

$${}^S A := \iota_S^* A \in {}^S\mathcal{A}. \tag{140}$$

Moreover, the induced metric on $S$ defines a supermetric ${}^S\mathbb{G}$ on ${}^S\mathcal{A}$. From this, one can define an SdW connection $\varpi_S$ on ${}^S\mathcal{A}$ and hence, via pullback, on the "phase space" of boundary fields $\mathrm{T}^*({}^S\mathcal{A})$. Now, thanks to the second of the equations (114), i.e. $s^i\mathbb{h}_i = 0$, the difference between two *generic*[62] horizontal perturbations $\mathbb{h}^\pm$ defines, without any loss of information, a vector field intrinsic to the boundary:

$$
{}^S\mathbb{Y} := \oint_S \iota_S^*(h^+ - h^-)\frac{\delta}{\delta({}^S A)} \in \mathrm{T}_{({}^S A)}({}^S\mathcal{A}). \tag{141}
$$

Now, the boundary field-space vector ${}^S\mathbb{Y}$ can be decomposed via $\varpi_S$ into its horizontal and vertical parts *within* ${}^S\mathcal{A}$:

$$
{}^S\mathbb{Y} = {}^S\mathbb{H} + \left({}^S\xi\right)^{\sharp_S}, \tag{142}
$$

where the $\cdot^{\sharp_S}$ operation is the $S$-intrinsic analog of $\cdot^\sharp$. Given equations (141) and (142), then it becomes clear that the parallel component of the continuity condition for a fiducial boundary (120),[63] is equivalent to demanding that ${}^S\mathbb{Y}$ has no horizontal component, i.e. ${}^S\mathbb{H} = 0$. Of course, in this case, the ${}^S\xi$ of (142) is identified with the $(\xi^- - \xi^+)$ of (120).

From these observations we conclude that *the parallel continuity condition is satisfied if and only if ${}^S\mathbb{Y}$ is purely vertical, that is if and only if ${}^S\mathbb{Y} = \varpi_S^{\sharp_S}\left({}^S\mathbb{Y}\right)$.* If this is the case, this last equation is only a more formal way to write (120), with $(\xi^+ - \xi^-)_{|S} = -\varpi_S({}^S\mathbb{Y})$ being a rewriting of (123).

We conclude this section by observing that the parallel continuity condition bears an interesting possibility. Note that if $S$ itself had corners, i.e. if it was subdivided into regions $S^\pm$ sharing a boundary, we could have repeated the same treatment for two possible horizontal differences, $({}^Sh)^+ - ({}^Sh)^-$, themselves arising from the difference of horizontals in a manifold of one higher dimension, as expressed in (142). This chain of descent to the boundaries of boundaries might become useful in discussions of more complex gluing patterns involving corners; a necessary extension for building general manifolds from fundamental building blocks. We conclude this section by noticing that this chain of descent is reminiscent of the BV-BFV formalism [58, 68, 69], but we will leave an investigation of these matters to future work.

## 6.3 Gluing of gauge potential fluctuations

We are now ready to apply the above results to the gluing of the perturbations of the gauge potential $A$. We include matter in the next section, and apply the construction to the elecric field in section 6.5.

Therefore, we consider

$$
\mathbb{X} = \int X\frac{\delta}{\delta A} \in \mathrm{T}_A\mathcal{A}, \tag{143}
$$

and decompose it, and its regional restrictions, into their SdW-horizontal and SdW-vertical components

$$
X = H + \mathrm{D}\Lambda \quad \text{and} \quad X^\pm = h^\pm + \mathrm{D}\lambda^\pm. \tag{144}
$$

Physically, whereas $\Lambda$ and $\lambda^\pm$ encode the "pure gauge" components of $X$ in $\Sigma$ and $R^\pm$ respectively, $H$ and $h^\pm$ encode their physical components. Therefore, the gluing question can be rephrased as the following: given only the regional gauge invariant perturbations $h^\pm$, is the global gauge invariant perturbation[64] $H$ *uniquely* reconstructed, provided it can be reconstructed at all?

---

[62]I.e. that do *not* have to necessarily satisfy the continuity condition (120).

[63]Fiducial interfaces are interfaces at which no fixed boundary condition is imposed.

[64]Notice that the theorem involves the perturbations of $A$ (elements of $\mathrm{T}\mathcal{A}$) over a globally smooth, fixed, background configuration $A$.

The theorem of the previous sections states that—whenever possible—*the reconstruction of a continuous H from $h^\pm$ is indeed unique*, and no additional information is needed to perform the gluing.

In particular, the theorem provides an explicit formula (116) for the reconstruction of the gauge transformations $\xi^\pm$ that relate the regional and global horizontals according to

$$H = (h^+ + \mathrm{D}\xi^\pm)\Theta_+ + (h^- + \mathrm{D}\xi^-)\Theta_- \,, \tag{145}$$

where the $\xi^\pm$ were fully determined in (132) and (129), i.e. by a covariant Laplace equations with boundary conditions determined in terms of the mismatch $\iota_S^*(h^+ - h^-)$.

However, the derivation assumed the mismatch $\iota_S^*(h^+ - h^-)$ to be a pure (gauge-covariant) gradient intrinsic to $S$. As explained in the previous section, whether this is the case can be checked by considering an SdW connection $\varpi_S$ intrinsic to $S$, and verifying whether $\iota_S^*(h^+ - h^-)$ is purely vertical with respect to $\varpi_S$. If this mismatch is not purely boundary-vertical, then there would be a physical discontinuity in the magnetic flux across $S$, i.e. in $F_{ab}$ ($a, b$ are tangential indices over $S$).[65]

With reference to EM, it is interesting to observe that such a discontinuity is *not* the consequence of a distributional surface current density on $S$, which would rather contribute a discontinuity in $s^i F_{ia}$ corresponding to the tangential magnetic field. Rather, it is the consequence of a distributional surface density of magnetic monopole charges. Indeed, in the same way a discontinuity in the electric flux across a surface is due to a nonvanishing surface density of electric charges, a discontinuity in the magnetic flux is due to a nonvanishing surface density of magnetic monopoles. But, postulating the configuration space of Yang-Mills theory to be fundamentally given by the space of smooth (or at least once-differentiable) connections $\mathcal{A}$, we are implicitly excluding this possibility from the onset: the algebraic validity of the Bianchi identities $\mathrm{D}F = 0$ excludes the existence of magnetic monopoles[66]—and thus guarantees that a physically allowed $H$ is continuous across $S$.

## 6.4 Gluing with matter: reducible configurations and charges

In this section, we will briefly discuss caveats of our gluing theorem due to reducibility. First, we briefly comment on the changes brought about the presence of a reducibility condition on the boundary that does not extend into the region. In that case, $\mu$ defined in (122), whose solution in terms of the difference in boundary horizontals is described in (123), is defined only up to the boundary stabilizers: $\mu \to \mu + {}^S\chi$. This degeneracy propagates to the determination of $\Pi$, in (132)—sending $\Pi \mapsto \Pi + \left(\mathcal{R}_+^{-1} + \mathcal{R}_-^{-1}\right)^{-1}({}^S\chi)$—and thereby to the final solution of the $\xi^\pm$ in (126). Thus the total solution $(\xi_+, \xi_-)$ acquires a physically significant degeneracy labeled by the boundary stabilizers. The degeneracy is physically significant since, for each choice of $h_\pm$, ${}^S\chi$, we obtain a distinct global $H$. That is, we obtain a $H({}^S\chi, h_\pm)$ that is not gauge-related to $H({}^S\chi', h_\pm)$.

---

[65]More precisely, in a neighbourhood of $S$, the relation between the curvature and the perturbation is: $F_{ab}(A + X) - F_{ab}(A) = [F_{ab}(A), \Lambda] + {}^S\mathrm{D}_{[b}H_{a]} + \mathcal{O}(X^2)$, where the first term on the right-hand side is an inconsequential perturbation in the gauge (vertical) direction and the second is the physical perturbation. Thus, only if $(h^+ - h^-)_a = \mathrm{D}_a\Xi$ does ${}^S\mathrm{D}_{[b}(h^+ - h^-)_{a]} = [F_{ab}, \Xi]$ feed into the gauge ambiguity; otherwise, a physical discontinuity in the parallel curvature will emerge. In this case, existence fails. But, once again, we do not aim here to give a complete characterization of existence.

[66]Notice that, the discontinuity in the components $s^i F_{ia}$ of the magnetic field at $S$ induced by the presence of surface currents is more subtle from a gluing perspective since it does not necessarily stem from a discontinuity of $h_i$ (it could also be due to a discontinuity in its normal derivative). Given any vector field $u$ in a neighbourhood of $S$ that is tangent to $S$, and recalling that $h_s = 0$ by the horizontality condition, one has that the perturbation of $F_{su}^\pm \equiv s^i u^j F_{ij}^\pm$ is given by $s^i u^j \mathrm{D}_i h_j^\pm = \mathrm{D}_s h_u^\pm - h_j^\pm (\$_u s)^j$.

The presence of a boundary stabilizer that is not extendible into the bulk is typical of asymptotic boundaries.[67] At finite boundaries, and in non-Abelian theories, this condition is only slightly "less generic" than the presence of a bulk stabilizer. This latter case is the one we will now focus on. It is most relevant in the Abelian theory, where such a bulk stabilizer is always present and there is no mismatch between bulk and boundary stabilizers.

In vacuum, the difference due to $\chi$ will then have no effect on the physical states. In the presence of matter, gluing is more subtle. Let us see how this goes.

First, some notation: Let $\mathbb{h}^{\pm} = \mathbb{h}_A^{\pm} + \mathbb{h}_{\psi}^{\pm}$ and $\mathbb{H} = \mathbb{H}_A + \mathbb{H}_{\psi}$, be horizontals, which decompose according to

$$\begin{cases} H_A = (h_A^+ + \mathrm{D}\xi^+)\Theta_+ + (h_A^- + \mathrm{D}\xi^-)\Theta_- \\[2ex] H_{\psi} = (h_{\psi}^+ - \xi^+\psi)\Theta_+ + (h_{\psi}^- + \xi^-\psi)\Theta_- \end{cases} \tag{146}$$

and e.g.

$$\mathbb{H} = \mathbb{H}_A \oplus \mathbb{H}_{\psi} = \int H_A \frac{\delta}{\delta A} + \int H_{\psi} \frac{\delta}{\delta \psi}. \tag{147}$$

As above, we are here implicitly using the SdW connection to assess horizontality. It is important to note that the matter horizontal components $\mathbb{h}_{\psi}^{\pm}$ are then, in a sense, parasitic on the gauge-field: they are just the matter perturbations corrected by the vertical displacement provided by the gauge sector. Namely, for a fermion field in the fundamental representation of $\mathcal{G}$ [19],

$$H_{\psi} = X_{\psi} - \varpi(\mathbb{X}_A)X_{\psi}, \tag{148}$$

where $\mathbb{X}_{\psi}$ and $\mathbb{X}_A$ denote arbitrary (not necessarily horizontal) matter and gauge-potential perturbations respectively. In other words, $\mathbb{H}_{\psi}$ and $\mathbb{h}_{\psi}^{\pm}$ do not satisfy horizontality conditions of their own. In section 5, we provided an interpretation of this in terms of Dirac dressings.

Then, we see that $\mathbb{H}$ (and $\mathbb{h}^{\pm}$) is horizontal (regionally horizontal, respectively) if and only if $\mathbb{H}_A$ ($\mathbb{h}_A^{\pm}$, respectively) is. This means in particular that the above procedure aimed at the determination of $\xi^{\pm}$ is completely insensitive to the presence of matter, and can be applied in the same way.

Now, all previous results on gluing go through seamlessly unless either one of the *regional* configurations of the gauge potential, i.e. $A^{\pm} = A_{|R^{\pm}}$, is reducible. On such configurations, a modification of the connection-form, $\tilde{\varpi}$, must be employed, and this comes with certain added difficulties and obstructions to the usual properties of $\varpi$—see sections 4.4 and 4.5. For what concerns this section, the main point is that at a reducible configurations $\tilde{A}$ an ambiguity is present in defining a pure-gauge transformation $\xi^+$ from a fluctuation of $\tilde{A}$ (parallel to the given stratum).

If, say, $A^+ = \tilde{A}^+$ is reducible, then the resulting ambiguity in the reconstruction of $\xi^+$ will have no effect on the reconstruction of the global horizontal gauge potential $H_A$, but it *will* generically render the reconstruction of the horizontal matter field $H_{\psi}$ ambiguous. This is always the case in QED, where we can always add constants $\chi_{\mathrm{EM}}^{\pm}$ to the reconstructed $\xi^{\pm}$ and where a constant phase shift will affect the Dirac fermions, unless they vanish. In a non-Abelian theory, the zoology of the solution is more complicated, and will depend on the gauge group as well as the type of matter fields (fundamental, adjoint, etc).

For definiteness and simplicity, we will henceforth suppose that only the regional configuration $A^+ = \tilde{A}^+$ is reducible by a single reducibility parameter, i.e. $\chi^+$ such that $\tilde{\mathrm{D}}\chi^+ = 0$, while $A^-$ is not reducible. The hypothesis that the stabilizer of $\tilde{A}^+$ is one-dimensional is quite

---

[67]As such, even at finite boundaries, it can be possibly interpreted as a (kinematical) *isolation* condition between two subsystems. With this interpretation, the above gluing ambiguity is maybe less surprising: if two subsystems are properly isolated there could be more ways of gluing them together.

strong, and it would be interesting to explore its relaxation (cf. the last paragraph of section 4.5).

Anyway, with these restrictions, we see that the the solution $\xi^+$ to the gluing boundary value problem (126) is defined only up to the addition of terms proportional to $\chi^+$. That is, there is a continuous 1-parameter family of solutions for $\xi^+$ that we write, by choosing an arbitrary origin $\xi_o^+$ and introducing the parameter $r$ (depending on the charge group), as

$$\xi_r^+ := \xi_o^+ + r\chi^+ \qquad r \in \mathbb{R} \text{ or } \mathbb{C}. \tag{149}$$

Then, two distinct possibilities are given: either $\psi$ vanishes at $S$ or it does not. The second case allows us to glue the two perturbations together if and only if we can find an $r$ such that

$$\xi_r^+ \psi_{|S} = \xi^- \psi_{|S}. \tag{150}$$

With the continuity hypothesis for the original global field perturbation $\mathbb{X} = \mathbb{X}_A + \mathbb{X}_\psi$, this equation would then fix the global ambiguity, but for generic values of $\psi_{|S}$ no solution exists.[68] If no solution exists, it means that the two perturbations are not glueable, i.e. they do not descend from a global smooth perturbation.

Conversely, in the first case, which is realized if $\psi^+$ vanishes at $S$, the gluing of the two perturbations $\mathbb{h}^\pm$ is possible for any $r$ but will give rise to *distinct horizontal global perturbations*. These should be interpreted as physically distinct alternatives, thus leading—for the first time in our analysis so far—to an actual ambiguity in the gluing procedure. This ambiguity is due to the concomitant presence of a reducible gauge potential and of charged matter.

To see how this comes about, we observe that in the presence of this stabilizer, there exists a 1-parameter family of global horizontal perturbations corresponding to each of the $\xi_r^+$, i.e. $\mathbb{H}^r = \mathbb{H}_A^r \oplus \mathbb{H}_\psi^r$, is given by

$$H_A^r \equiv H_A^o \qquad \text{and} \qquad H_\psi^r = H_\psi^o + r\chi^+\psi\,\Theta_+, \tag{151}$$

where the same notation as in (147) was used.

Now, two possible situations are given: either $\chi^+$ stabilizes $\psi^+$ throughout $R^+$, or it does not.

If $\psi^+$ is also stabilized,[69] then uniqueness of the reconstructed global radiative mode is untouched: even if the regional gauge transformations $\xi^\pm$ are ambiguous, $\mathbb{H}$ of (146) will not be since in this case $\mathbb{H}^r \equiv \mathbb{H}^o$. The generally quite restrictive condition of $\chi^+$ stabilizing $\psi^+$ trivially applies if matter is absent from $R^+$, in which case $\mathbb{H} = \mathbb{H}_A$ is clearly unaffected by $\chi^+$ such that $\tilde{D}\chi^+ = 0$.

If $\psi^+$ is *not* stabilized by $\chi^+$, on the other hand, the resulting $\mathbb{H}^r$ are indeed *distinct* from one another. This setup formalizes the beam splitter thought experiment devised by 't Hooft [70] (see also [71]), and can be used to provide a concrete example for the considerations of Wallace and Greaves, characterizing "symmetries with direct empirical significance" (DES) [72].

As a proof of concept that the ensuing states are regionally indistinguishable but globally distinct, let us consider the following simplified scenario in the Abelian theory, closely related to 't Hooft's beam splitter (figure 7): let $R^\pm$ contain one charged particle each, located

---

[68]These compatibility requirements between $\chi^+$ and $\psi^+$ could be further formalized in terms of the kernel of the Higgs functional connection introduced in [19]. However, the presence of distributional charged matter at $S$—as manifested over e.g. an idealized conducting plate—generally blocks the possibility of a smooth gluing *of the electric field, E,* discussed in the following section.

[69]For matter fields $\psi \neq 0$ throughout $R^+$ which are in the fundamental representation, $G = \mathrm{SU}(N \geq 3)$ is needed; see [19, Sec.7].

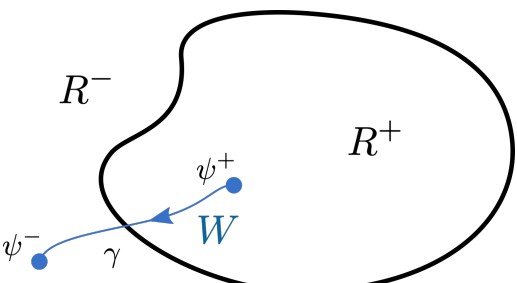

Figure 7: We consider the gluing of two regions $R^+$ and $R^-$ containing two charged (point) particles $\psi^+$ and $\psi^-$ connected by a Wilson line $\gamma$. In $\Sigma$, the observable $W$ (152) is gauge invariant. Nonetheless, its gluing is ambiguous in the presence of a regional stabilizer (153). This ambiguity is closely related to 't Hooft's beam splitter thought experiment and highlights the fact that global phase transformations (which are part of the stabilizer) are physical transformations distinguished from pure gauge transformations. Our formalism allows to make this distinction also within a finite and bounded regional, i.e. at the quasi-locally.

at $x^\pm \in R^\pm$, and thus $A^+$ admits a reducibility parameter $\chi^+ = const$. Denoting the particle's spinorial configurations[70] by $|\psi^\pm\rangle$, we consider then the following global *gauge-invariant* Wilson-line observable between two charged particles (with obvious notation):

$$W = \langle \psi^- | \exp\left( \int_\gamma A \right) | \psi^+ \rangle, \tag{152}$$

where $\gamma$ is some path connecting across $S$ the positions $x^\pm \in R^\pm$ of the charged particles $|\psi^\pm\rangle$. Now, if we "unglue" the two regions, perform the (infinitesimal) charge transformation $\chi^+$ in $R^+$ and glue back (which as we saw above is a seamless operation), we will find that: whereas $\exp\left( \int_\gamma A \right)$ and $|\psi^-\rangle$ have not changed at all, $|\psi^+\rangle$ has changed by the (infinitesimal) amount $\delta_{\chi^+} |\psi^+\rangle = -\chi^+ |\psi^+\rangle$; in turn this means that the global, gauge invariant, observable $W$ is able to distinguish the two global states, since generically

$$\delta_{\chi^+} W = -\langle \psi^- | \exp\left( \int_\gamma A \right) \chi^+ | \psi^+ \rangle \neq 0. \tag{153}$$

In sum: in the presence of matter, the Wilson line $W$ is a gauge-invariant functional that is sensitive to the ambiguity present in the gluing procedure, i.e. in the determination of the vertical adjustments $\xi^\pm(h^+, h^-)$. As per our gluing theorem, this ambiguity is in one-to-one correspondence with choices of regional stabilizer, $\chi_\pm$, and is only relevant in the presence of matter (since $\delta_{\chi^+} A^+ = \partial \chi^+ \equiv 0$, but in general $\delta_{\chi^+} \psi^+ = -\chi^+ \psi^+ \neq 0$).

Of course, this construction is strictly related to the ability of defining a charge for the $|\psi^+\rangle$ on the reducible background $A^+ = \tilde{A}^+$, and is in line with the claim that *(regional) stabilizers must be attributed a different status than generic gauge transformation*, as discussed in section 4 (see in particular the last two paragraphs of 4.4 before the Remark).

## 6.5 Gluing of the electric field

We now turn our attention to the gluing of the electric field $E$.

---

[70]The bra-ket notation is employed to ease the writing of the following formulae, it does not refer to any quantum treatment.

We start by recalling the representation of the electric field as a configuration-space vector $\mathbb{E}$:

$$\mathbb{E} = \int E_i \frac{\delta}{\delta A_i} \in T_A \mathcal{A} \subset \Phi. \tag{154}$$

In this section, for ease of notation, we shall treat $E$ as a one-form, i.e.—consistently with the above—$E$ will stand for $E_i := g_{ij}E^j$. Thus (the components) of the global and regional SdW decompositions of $\mathbb{E}$ (see equation (52) in Section 3.1) are

$$E = E_{\text{rad}} + \mathrm{D}\varphi \quad \text{and} \quad E^{\pm} = E_{\text{rad}}^{\pm} + \mathrm{D}\varphi^{\pm}. \tag{155}$$

We emphasize that, as was the case with $h^{\pm}$ and $\lambda^{\pm}$ in relation to $H$ and $\Lambda$ in (114), $\varphi^{\pm}$ is *not* the regional restriction[71] of $\varphi$ to $R^{\pm}$, and similarly $E_{\text{rad}}^{\pm}$ is *not* the regional restriction of $E_{\text{rad}}$ to $R^{\pm}$; instead,

$$\begin{cases} \varphi = (\varphi^+ - \eta^+)\Theta_{\pm} + (\varphi^- - \eta^-)\Theta_- \\ E_{\text{rad}} = (E_{\text{rad}}^+ + \mathrm{D}\eta^+)\Theta_+ + (E_{\text{rad}}^- + \mathrm{D}\eta^-)\Theta_- \end{cases} \tag{156}$$

where, according to the theorem of section 6.1, the $\eta^{\pm}$ are fully determined by the mismatch of $(E_{\text{rad}}^+ - E_{\text{rad}}^-)_{|S}$; the appropriate behaviour of $\varphi$ merely follows. Notice also that the electric flux $f$ through $S$ corresponds precisely to $\Pi = f$ of (116) in our main gluing theorem in section 6.1.

In the case of the electric field, we do not interpret the SdW vertical component $\varphi$ of $\mathbb{E}$ as a pure-gauge quantity, but as a Coulombic component of the electric field, while $E_{\text{rad}}$ is what we called its radiative component.

Therefore, equation (156) states that the Coulombic/radiative split of the electric field depends on the choice of region in which the split is performed.

To understand this phenomenon, it is particularly instructive to consider first the case without matter. We also recall that we have assumed, for simplicity, that the simply connected global region $\Sigma = R^+ \cup_S R^-$ has no boundary, i.e. $\partial \Sigma = \emptyset$. From the above equations, it then follows that $\varphi \equiv 0$, and therefore, according to (156) $\varphi^{\pm} = \eta^{\pm}$. Since $\eta^{\pm}$ are entirely functions of $E_{\text{rad}|S}^{\pm}$, it follows that all components of the global electric field are determined solely by its regional radiatives.

Indeed, for a globally radiative electric field (i.e. no global boundary and no charges), $E = E_{\text{rad}}$ and $E_{|R^{\pm}} = E_{\text{rad}}^{\pm} - \mathrm{D}\eta^{\pm}$ with $\eta^{\pm}$ functionals of $(E_{\text{rad}}^+ - E_{\text{rad}}^-)_{|S}$ only.[72] Thus, in this case, once *both* regional radiatives are known, even the *regional* Coulombic components are completely determined—including the electric flux $f$ through $S$, which is thus no longer an independent degree of freedom once the radiative modes are accessible in *both* regions.

Thus, in this case—when the larger (glued) region $\Sigma$ has no boundary,—the *regional radiative modes encode the totality of the dof in the joint system.*

In particular, the conclusion reached in section 3.4 from a regional viewpoint that $f$ through $S$ must be superselected is—as expected—a mere artifact of excluding observables in the complement of that region.

The addition of charged matter does not change this conclusion.

In sum, once the radiative modes are given in both regions, the role of the flux $f$ at $S$—i.e. to regionally fix $\varphi^{\pm}$—is taken over by $(E_{\text{rad}}^+ - E_{\text{rad}}^-)_{|S}$. Thus $f$—which is often claimed to embody the "new boundary degrees of freedom" [73] or their momenta [12]—also constitutes a piece of redundant information for the final result of the gluing. Heuristically, we could say

---

[71]Having run out of symbols, we could not use the same capitalized vs. lower case variables to indicate that relationship.

[72]Again, as already stressed, it is important to note that regional restriction and horizontal projection do not commute, thus e.g.: $E_{\text{rad}}^{\pm} \neq E_{\text{rad}|R^{\pm}}$.

that $f$ only shows up when encoding one subregion's ignorance of the other, i.e. when we do not have access to both radiatives, $E_{\text{rad}}^{\pm}$.

Explicitly, playing the role of $\Pi$ in the theorem of section 6.1, the flux is given by

$$f = \left(\mathcal{R}_+^{-1} + \mathcal{R}_-^{-1}\right)^{-1} \left({}^S\mathrm{D}^2\right)^{-1} {}^S\mathrm{D}^a \iota_S^*(E_{\text{rad}}^+ - E_{\text{rad}}^-)_a. \tag{157}$$

This conclusion is only challenged in the presence of nontrivial cohomological 1-cycles in the Cauchy surface, a point exemplified in section 6.8.

Concerning the analogues of the continuity conditions explored in section 6.2 for the gauge potential, we observe that on-shell the electric field is continuous across $S$ if and only if there is no *distributional* charge density there. Such a charge density would create a discontinuity in the fluxes $E_{s|S}^{\pm} \equiv f^{\pm}$. No analogous physical discontinuity can be found in the components of the electric field parallel to $S$. Moreover, if there is no charge density and therefore $E$ is continuous, the difference $(E_{\text{rad}}^+ - E_{\text{rad}}^-)_{|S}$ is the same as the difference $(\mathrm{D}_i\varphi^+ - \mathrm{D}_i\varphi^-)_{|S}$. Since the latter is always of the pure-gradient form, the radiative parts of a continuous electric field satisfy (on-shell of Gauss) the analogue of (120).

## 6.6 On the energy of radiative and Coulombic modes

The radiative/Coulombic split of $E$ satisfies a monotonocity property, which roughly states that *in a composite region* $\Sigma = R^+ \cup_S R^-$, *a larger portion of the energy is attributed to the radiative part of the electric field than it is in the disjoint union of* $R^+$ *and* $R^-$; *the converse holds for its Coulombic part*. This section is devoted to establishing and interpreting this result.

Let us start by writing the energy $\mathcal{H}$ contained in $\Sigma$. We decompose this energy into its electric (kinetic) and magnetic (potential) parts,

$$\mathcal{H} = \mathcal{E} + \mathcal{B} = \int_\Sigma \sqrt{g}\,\mathrm{Tr}(E^i E_i) + \int_\Sigma \sqrt{g}\,\tfrac{1}{2}\mathrm{Tr}(F^{ij}F_{ij}). \tag{158}$$

Since $F$ is fully determined by the background value of $A$ (which undergoes no SdW splitting), we will henceforth focus on the electric contribution. This can be written more abstractly as

$$\mathcal{E} = ||\mathbb{E}||^2 = ||\mathbb{E}||_+^2 + ||\mathbb{E}||_-^2, \tag{159}$$

with $||\cdot||$ and $||\cdot||_\pm$ the $\mathbb{G}$-norms over $\Sigma$ and $R^\pm$ respectively. E.g. $||\mathbb{E}||_+^2 = \mathbb{G}_{R^+}(\mathbb{E}, \mathbb{E}) = \int_{R^+} \sqrt{g}\,g^{ij}\,\mathrm{Tr}(E_i E_j)$.

Consider now the radiative/Coulombic decomposition of $E$, and recall that it corresponds to a horizontal/vertical orthogonal decomposition with respect to the $\mathbb{G}$ supermetric. Then,

$$||\mathbb{E}||^2 = ||\mathbb{E}_{\text{rad}}||^2 + ||\varphi^\sharp||^2 =: \mathcal{E}_{\text{rad}} + \mathcal{E}_{\text{Coul}}, \tag{160}$$

and similarly on $R^\pm$.

Applying the same decomposition to the second gluing formula of (156) gives

$$\begin{aligned}
||\mathbb{E}_{\text{rad}}||^2 &= ||\mathbb{E}_{\text{rad}}^+ + (\eta^+)^\sharp||_+^2 + ||\mathbb{E}_{\text{rad}}^- + (\eta^-)^\sharp||_-^2 \\
&= ||\mathbb{E}_{\text{rad}}^+||_+^2 + ||\mathbb{E}_{\text{rad}}^-||_-^2 + ||(\eta^+)^\sharp||_+^2 + ||(\eta^-)^\sharp||_-^2 \\
&\geq ||\mathbb{E}_{\text{rad}}^+||_+^2 + ||\mathbb{E}_{\text{rad}}^-||_-^2.
\end{aligned} \tag{161}$$

From the additivity of $\mathcal{E}$ (159), the gluing formula (156) and the equation above, it follows

that the total Coulombic contribution correspondingly decreases by the same amount:[73]

$$
\begin{aligned}
||\varphi^\sharp||^2 &= ||(\varphi^+)^\sharp||_+^2 + ||(\varphi^-)^\sharp||_-^2 - ||(\eta^+)^\sharp||_+^2 - ||(\eta^-)^\sharp||_-^2 \\
&\leq ||(\varphi^+)^\sharp||_+^2 + ||(\varphi^-)^\sharp||_-^2.
\end{aligned}
\tag{162}
$$

We have thus proved (and qualified) our statement above.

So, if to the radiative part of $E$ we ascribe the kinetic energy of the radiative modes, the following question arises: which new radiative field strengths are included in $\Sigma$ that are not present in the disjoint union of $R^+$ and $R^-$?

The answer lies at the interface $S$: the regional Coulombic and vertical adjustments, $\eta^\pm$ and $\xi^\pm$, respectively, from the global perspective are additions to the radiative sector of $\Sigma$ with respect to the radiative sectors of $R^\pm$. Although supported on the whole regions $R^\pm$ respectively, these new components, are completely determined by the mismatch at $S$ of the two regional radiative modes, $\mathbb{E}^\pm_{\mathrm{rad}|S}$ (or $h^\pm_{|S}$, resp). In other words, the new global radiative field strength that emerges on $\Sigma$ upon gluing $R^\pm$ is entirely determined by the standard regional radiative modes at the boundary.

In formulas:

$$
\mathcal{E} = \mathcal{E}_{\mathrm{rad}} = \mathcal{E}_{\mathrm{rad}}^+ + \mathcal{E}_{\mathrm{rad}}^- + \oint_S \sqrt{h}\,\mathrm{Tr}\Big(f\big(\mathcal{R}_+^{-1} + \mathcal{R}_-^{-1}\big)f\Big),
\tag{163}
$$

where $f$ should be understood as given by (157) and there we used the following relation $\mathcal{E}^\pm_{\mathrm{Coul}} = ||(\varphi^\pm)^\sharp||_\pm^2 = \oint_S \sqrt{h}\mathrm{Tr}(f\,\mathcal{R}_\pm^{-1}f)$ that is easily deducible from the definitions and results of section 6.1 (also, a similar computation will be carried out in more detail in the next section).

We summarize these results in the following[74]

**Proposition 6.3.** *Assuming the same geometrical setting relevant for the General Gluing Theorem 6.1, the following radiative/Coulombic energy balance holds:*

$$
\mathcal{E}_{\mathrm{rad}} - (\mathcal{E}_{\mathrm{rad}}^+ + \mathcal{E}_{\mathrm{rad}}^-) = (\mathcal{E}_{\mathrm{Coul}}^+ + \mathcal{E}_{\mathrm{Coul}}^-) - \mathcal{E}_{\mathrm{Coul}} = \oint_S \sqrt{h}\,\mathrm{Tr}\Big(f\big(\mathcal{R}_+^{-1} + \mathcal{R}_-^{-1}\big)f\Big) \geq 0,
$$

*with $f = f(E_{\mathrm{rad}}^+, E_{\mathrm{rad}}^-)$ as in (157). The equality sign holds if and only if $f = 0$.*

It is important to stress that the new global contribution to the radiative energy is not encoded in either region, since it depends on the mismatch at $S$ of the two regional components (157). Thus, in this precise sense, we can claim that there is an additional component to the global radiative field strength: it results from the gluing and arises from the relation of the two subsystems at their common boundary.

## 6.7 Gluing of the symplectic potentials

It is now straightforward to study the gluing of the SdW-horizontal symplectic potential. As above, we focus on the situation where a $D$-dimensional simply connected hypersurface without boundary $\Sigma \cong S^D$ is split into two regions $R^\pm \cong B^D$ glued at $S = \partial R^\pm \cong S^{D-1}$, i.e. $\Sigma = R^+ \cup_S R^-$.

---

[73]This follows from the comparison of the following two expressions

$$
\begin{cases}
||\mathbb{E}||^2 &= ||\mathbb{E}_{\mathrm{rad}} - \varphi^\sharp||^2 = ||\mathbb{E}_{\mathrm{rad}}||^2 + ||\varphi^\sharp||^2 = ||\mathbb{E}_{\mathrm{rad}}^+||_+^2 + ||\mathbb{E}_{\mathrm{rad}}^-||_-^2 + ||(\eta^+)^\sharp||_+^2 + ||(\eta^-)^\sharp||_-^2 + ||\varphi^\sharp||^2 \\
||\mathbb{E}||^2 &= ||\mathbb{E}||_+^2 + ||\mathbb{E}||_-^2 = ||\mathbb{E}_{\mathrm{rad}}^+ - (\varphi^+)^\sharp||_+^2 + ||\mathbb{E}_{\mathrm{rad}}^- - (\varphi^-)^\sharp||_+^2 = ||\mathbb{E}_{\mathrm{rad}}^+||_+^2 + ||(\varphi^+)^\sharp||_+^2 + ||\mathbb{E}_{\mathrm{rad}}^-||_-^2 + ||(\varphi^-)^\sharp||_+^2.
\end{cases}
$$

[74]The last statement follows from the positivity of $\mathcal{R}$, see the proof of the SdW Gluing Lemma 6.2.

In this case, from (58), the total symplectic potential reads

$$\theta = \int_\Sigma \sqrt{g}\,\Big\{\mathrm{Tr}\Big(E^i \mathrm{d}A_i\Big) - \overline{\psi}\gamma^0 \mathrm{d}\psi\Big\} \approx \int_\Sigma \sqrt{g}\,\Big\{\mathrm{Tr}\Big(E^i_{\mathrm{rad}} \mathrm{d}_\perp A_i\Big) - \overline{\psi}\gamma^0 \mathrm{d}_\perp \psi\Big\} = \theta^\perp, \qquad (164)$$

where $\theta \approx \theta^\perp$ since $\partial \Sigma = \emptyset$.

Now, $\theta$ can also be decomposed into $\theta = \theta^+ + \theta^-$ simply by factorizing the integration domain in the first expression above,

$$\theta^\pm = \int_{R^\pm} \sqrt{g}\,\Big\{\mathrm{Tr}\Big(E^i \mathrm{d}A_i\Big) - \overline{\psi}\gamma^0 \mathrm{d}\psi\Big\}. \qquad (165)$$

Each of these regional contributions can be written in the SdW decomposition following (58):

$$\theta^\pm \approx \int_{R^\pm} \sqrt{g}\,\Big\{\mathrm{Tr}\Big(E^{\pm\,i}_{\mathrm{rad}} \mathrm{d}_{\perp(\pm)} A_i\Big) - \overline{\psi}\gamma^0 \mathrm{d}_{\perp(\pm)}\psi\Big\} \pm \oint_S \sqrt{h}\,\mathrm{Tr}\big(f\,\varpi_\pm\big), \qquad (166)$$

where $\perp(\pm)$ denotes that the SdW decomposition intrinsic to $R^\pm$ has been respectively used, and the sign of the last term depends on the fact that, in $f = s^i E_{i|S}$, the normal $s^i$ to $S$ is outgoing for $R^+$ and ingoing for $R^-$. Thus, we find

$$\theta \approx \theta^{\perp(+)} + \theta^{\perp(-)} + \oint_S \sqrt{h}\,\mathrm{Tr}\big(f(\varpi_+ - \varpi_-)\big). \qquad (167)$$

The results of section 6.1, and in particular equation (123), can be applied[75] to $\varpi_\pm$ to obtain

$$(\varpi_+ - \varpi_-)_{|S} = -(^S\mathrm{D}^2)^{-1S}\mathrm{D}^a \iota_S^*(\mathrm{d}_{\perp(+)} A - \mathrm{d}_{\perp(-)} A)_a \equiv -\frac{^S\mathrm{D}[\mathrm{d}_\perp A]_S^\pm}{^S\mathrm{D}^2}. \qquad (168)$$

Here, $(\varpi_\pm)_{|S}$ means that the connection—which is valued in $\mathrm{Lie}(\mathcal{G}) = C(R^\pm, \mathrm{Lie}(G))$—is evaluated at the boundary $S = \partial R^\pm$, i.e. at points $x \in \partial R^\pm$. We have also introduced a new short-hand symbol for the interface mismatch of a given regional quantity $\bullet$, namely $[\bullet]_S^\pm$. For more compact notation, we have also schematically denoted the inverse operator by a fraction $(^S\mathrm{D}^2)^{-1}(\bullet) := \frac{\bullet}{(^S\mathrm{D}^2)}$. Similarly, we recall[76] (157)

$$\Big(\mathcal{R}_+^{-1} + \mathcal{R}_-^{-1}\Big)(f) = -(^S\mathrm{D}^2)^{-1S}\mathrm{D}^a \iota_S^*(E^+_{\mathrm{rad}} - E^-_{\mathrm{rad}})_a \equiv -\frac{^S\mathrm{D}[E_{\mathrm{rad}}]_S^\pm}{^S\mathrm{D}^2}. \qquad (169)$$

Hence, combining these results, and remembering that $\mathcal{R}_\pm$ is self-adjoint, we find the following result for the gluing of the symplectic potential:

**Theorem 6.4** (Gluing of the symplectic potential). *Consider the same geometrical setting relevant for the General Gluing Theorem 6.1. Denote, the YM regional symplectic potentials associated to $\Sigma$ and $R^\pm$ by $\theta$ and $\theta^{(\pm)}$ respectively, and their SdW-horizontal counterparts by $\theta^\perp$ and $\theta^{\perp(\pm)}$ respectively. Then, as a corollary of the SdW Gluing Lemma,*

$$\theta \overset{\partial\Sigma=\emptyset}{\approx} \theta^\perp = \theta^{\perp(+)} + \theta^{\perp(-)} + \oint_S \sqrt{h}\,\mathrm{Tr}\left(\frac{^S\mathrm{D}[E_{rad}]_S^\pm}{^S\mathrm{D}^2}\Big(\mathcal{R}_+^{-1} + \mathcal{R}_-^{-1}\Big)^{-1}\frac{^S\mathrm{D}[\mathrm{d}_\perp A]_S^\pm}{^S\mathrm{D}^2}\right). \qquad (170)$$

*Since both $\Omega^\perp := \mathrm{d}\theta^\perp$ and $\Omega^{\perp(\pm)} := \mathrm{d}\theta^{\perp(\pm)}$ are basic and closed, each of the terms on rhs above is projectable on the reduced phase space $\mathcal{A}/\mathcal{G}$. Since $\Omega^\perp$ projects to the full reduced symplectic structure (cf. section 3.4 and [20]), each of the terms of the rhs encodes a "pair" of reduced canonical dof. The last terms, in particular, encodes the "new" radiative dof emerging upon gluing.*

---

[75]This is entirely compatible with the standard definition of $\varpi$, which can be seen by noticing that given a vector $\mathbb{Y} \in \mathrm{T}_A\mathcal{A}$: $\mathbb{i}_\mathbb{Y}\mathrm{d}_\perp A_i = H_i$, $\mathbb{i}_\mathbb{Y}\varpi = \Lambda$, and similarly $\mathbb{i}_\mathbb{Y}\mathrm{d}_{\perp(\pm)}A_i = h_i^\pm$, $\mathbb{i}_\mathbb{Y}\varpi_\pm = \lambda^\pm$.

[76]The notation used for (157) has been here (slightly) adapted to fit with the notation used in the rest of this section. We apologize with the reader for the inconvenience.

In other words, $\theta \approx \theta^\perp$ is a functional *only* of the regional *radiative* electric fields $E^\pm_{rad}$ and the regional *SdW-horizontal* differentials $\mathbb{d}_{\perp(\pm)}A$—i.e. it does *not* require any knowledge of the regional Coulombic or pure-gauge dof. Nonetheless, $\theta^\perp$ does not factorize in terms of its regional SdW-horizontal counterparts, $\theta \approx \theta^\perp \neq \theta^{\perp(+)} + \theta^{\perp(-)}$. This non-factorizability has its root in the nonlocality of the horizontal/vertical decomposition. Its physical consequence is the emergence of new radiative dof upon gluing, which express the relational nature of the gauge theory across the interface. As discussed in section 6.6, the *mismatch* of the horizontal/radiative modes at the interface $S$ plays—from the global perspective—the role of a new horizontal/radiative dof which is not present in *either* region.

As emphasized in the previous section, upon gluing, we are—consistently—no longer required to superselect, or otherwise fix or refer to the electric flux $f$ trough $S$ (which was conjugate to the pure-gauge part of the gauge potential): $f$ can now be reconstructed from the mismatch of the *electric* radiative modes (157).

In sum, the horizontal symplectic structure of Yang-Mills theory fails, as expected, to factorize into regional horizontal symplectic structures. This is because there are global horizontal modes can only be reconstructed from the two regional radiative modes as functionals of their nontrivial *mismatch* at the common interface $S$.

## 6.8 Example: 1-dimensional gluing and the emergence of topological modes

In this final section we work out a simple example, implementing the gluing of 1-dimensional intervals. Two cases are given: two closed intervals are glued into a larger interval, and one interval is glued on itself to form a circle. This second case falls outside the simply-connected setup we adopted for the rest of the paper. Nonetheless, this case allows us to easily discuss, without introducing a host of new technologies, the emergence of new global (or "topological") degrees of freedom associated to the non trivial cohomology of the circle.

### 6.8.1 Gluing into an interval

Let us start by considering two closed intervals $I^+ = [0, 1]$ and $I^- = [-1, 0]$, that we shall glue together to form a new closed interval $I = [-1, 1]$. We shall see that, since on the interval the gauge potential must be pure gauge, the regional horizontal perturbations must vanish—a fact consistently encoded by our gluing formula. Although somewhat trivial, this example helps us set the stage for the gluing into a circle.

We first characterize the 1-dimensional gauge fields and their horizontal perturbations. One dimensional gauge fields are always locally pure gauge,

$$A^\pm = g^{-1}_\pm \mathrm{d}g_\pm, \tag{171}$$

for $g_+(x) = \mathrm{Pexp} \int_0^x A$ on $I^+$ and similarly on $I^-$, where we choose $g_-$ such that $g_-(0) = \mathbb{1}$ too ($x = 0$ is where the gluing takes place). Since in one dimension $s^i h_{i|S} = 0$ implies $h_{|S} = 0$, SdW-horizontal perturbations $\mathbb{h}^\pm$ in $I^\pm$, according to (114) must satisfy the equations

$$D^\pm h^\pm = 0, \quad \text{and} \quad h^\pm_{|\partial I^\pm} = 0, \tag{172}$$

which can be rewritten in terms of $\tilde{h}^\pm := g_\pm h^\pm g^{-1}_\pm$ as $\partial \tilde{h}^\pm = 0$ and $\tilde{h}^\pm_{|\partial I^\pm} = 0$. Now, these equations can be solved to give $\tilde{h}^\pm = 0$ and hence

$$\mathbb{h}^\pm = 0. \tag{173}$$

This is solely an immediate consequence of the pure gauge character of all 1-dimensional configurations, and therefore all perturbations over topologically trivial regions must be purely vertical.

Applying these results on the horizontal/vertical decomposition of fields on the interval to the electric field, we deduce that on the interval all electric fields are purely Coulombic. As per section 6.5, without any knowledge of regions outside of the interval $I^+ \equiv R^+$, this is entirely characterized by the charge content of the interval and by $f$ at its boundary $S$. The latter encodes our ignorance of the outside of the region.

Let us now analyze the gluing. Again, the global horizontal vector is denoted by

$$H = (h^+ + D\xi^+)\Theta_+ + (h^- + D\xi^-)\Theta_- = D\xi^+\Theta_+ + D\xi^-\Theta_-, \tag{174}$$

as in (113). The relevant equations for gluing arise as in (126), with a couple of new features: (*i*) there is no analogue to the last equation of (126), since $h_i$ has only one component that is transverse to the zero-dimensional gluing surface $S$; and (*ii*) we have to add one equation per global boundary of the interval $I = [-1, 1]$, since the total horizontal vector has now (two) endpoint boundaries, $\partial\Sigma \equiv \partial I = \{-1\} \cup \{+1\} \neq \emptyset$.

Thus,

$$\begin{cases} D^2\xi^\pm = 0, & \text{in } I^\pm, \\ D(\xi^+ - \xi^-) = 0, & \text{at } \partial I^+ \cap \partial I^- = \{0\}, \\ D\xi^\pm = 0, & \text{at } \partial I = \{\pm 1\}. \end{cases} \tag{175}$$

Now, again, by defining $\tilde{\xi}^\pm := g_\pm \xi^\pm g_\pm^{-1}$, we can turn the covariant derivatives into ordinary ones. This allows us to readily solve these equations. In fact, the bulk equations (the first of (175)) tell us that

$$\tilde{\xi}^\pm = \pm\tilde{\Pi}^\pm x + \tilde{\chi}^\pm, \tag{176}$$

where $\tilde{\chi}^\pm$ are constant functions valued in Lie($G$) corresponding to two arbitrary reducibility parameters of the vanishing configuration $\tilde{A}^\pm = 0$. This is a concrete example of the discussion in the previous section.

Now, the second equation of (175) sets $\tilde{\Pi}^+ = -\tilde{\Pi}^-$, and the third one sets them equal to zero. Since the $\tilde{\chi}_\pm$ don't affect the value of the regional horizontal fields, we hence conclude that in this case the unique solution to the gluing problem at hand is $\xi^\pm = 0$ which readily leads to $\mathbb{H} = 0$, consistently with the general regional result (173). This concludes the gluing of two intervals $I^\pm$ into a larger one $I = [-1, 1]$.

### 6.8.2 Gluing into a circle

We now move on to the second case, where one interval, $I = [-\pi, \pi] \ni \phi$, has its ends glued to form a unit circle. To keep the two cases notationally distinct, we have denoted an element of the circle by $\phi$, as opposed to $x$ of the interval in the previous case. This case requires a little more care.

The idea is to split $I$ into two intervals which overlap around $\phi = 0$, e.g. on the interval $U_\epsilon := (-\epsilon, \epsilon)$. Thus we consider $I^- = [-\pi, \epsilon)$ and $I^+ = (-\epsilon, \pi]$, so that we can glue at $\phi = \pm\pi$ according to the procedures of the above section, while matching the overlap of charts around $\phi = 0$ to close the interval into a circle.

This allows us to separate the problem of gluing from the problem of covering the circle. The latter is accomplished by overlapping open charts, with transition functions which appropriately match the gauge configuration.

Let us start by analyzing the background configuration $A^\pm$ on $I^\pm$. We assume, as in the previous sections, that the configurations $A^\pm$ join smoothly at $\phi = \pm\pi$.

As above, $A^\pm$ are pure gauge, i.e. $A^\pm = g_\pm^{-1} dg_\pm$ with $g_+(\pi) = g_-(-\pi)$. On the other hand, on $U_\epsilon$, the configurations $A^\pm$ do not have to be equal; they need only be related by the action of a gauge transformation $\kappa$, the transition function. Since we are in 1-dimension, this does not constitute a restriction; one simply has $\kappa = g_-^{-1} g_+$.

Now, we move on to consider the horizontal perturbations. We shall find that the relevant horizontality equations for $\mathbb{h}^\pm$ involve boundary conditions only at $\phi = \pm\pi$, and the one for $\mathbb{H}$ does not involve boundary conditions at all. In particular no boundary conditions are imposed at the open-extrema of the intervals $I^\pm$. This is not because the intervals are open, but rather because there are no boundaries from the perspective of the global $\mathbb{H}$. But let us be more detailed.

We start from the observation that on the overlap region $U_\epsilon$, generic perturbations $\mathbb{X}^\pm$ must be gauge related through $X^+ = \mathrm{Ad}_\kappa X^-$. This means that, using the appropriate partitions of unity over $S^1$, there is no difficulty, nor ambiguity, in the patching of the SdW inner products over $I^+$ and $I^-$: we obtain an inner product over $S^1$ between two perturbations $\mathbb{X}^\pm$ and $\mathbb{Y}^\pm$ that satisfy the overlap condition we have just described. Recalling that SdW-horizontality is the requirement of being orthogonal to any purely vertical vector with respect to the SdW supermetric, we see that the horizontality condition for $\mathbb{H}$ does *not* involve boundary conditions at the non-glued boundaries of $I^\pm$, i.e. at $\phi = \pm\epsilon$. Of course, this was an expected result from the closed nature of the manifold on which $\mathbb{H}$ resides.

Focusing now on horizontal perturbations, it is easy to see that this discussion doesn't change the fact that $\mathbb{h}^\pm = 0$, since the manifold on which they reside still has boundaries at $\phi = \pm\pi$. Note moreover that $\mathbb{h}^\pm = 0$ implies that their matching on $U_\epsilon$ is automatic. However, this discussion leads us to a horizontality condition for $\mathbb{H}$ that is distinct from the one found for the gluing into an interval (175). Indeed, in the present case, we find

$$
\begin{cases}
\mathrm{D}^2 \xi^\pm = 0\,, & \text{in } I^\pm\,, \\
\mathrm{D}(\xi^+ - \xi^-) = 0\,, & \text{at } \phi = \pm\pi\,,
\end{cases}
\tag{177}
$$

with *no* extra conditions at $\phi = \pm\epsilon$. Hence, it is readily clear that the solutions for $\xi^\pm$ are here much less restricted than they were in the closed interval case considered above: in this case we find that

$$
\xi^\pm = g_\pm^{-1}(\tilde\Pi\phi + \tilde\chi^\pm)g_\pm\,,
\tag{178}
$$

with the same, possibly non-vanishing, $\tilde\Pi$ for both the $\pm$ choices. From this we obtain,

$$
H = g_\pm^{-1}\tilde\Pi g_\pm\,.
\tag{179}
$$

As for the background, matching the perturbed configurations in $U_\epsilon$ comes at no cost (since $\mathbb{h}_\pm = 0$).

In summary, we see that the gluing procedure has no unique solution in this case, as a consequence of the absence of a second "outer" boundary for the interval (which is glued into a circle). The second outer boundary is instead replaced by the chart matching.[77] We thus obtain a one-parameter family of solutions parametrized by an element $\tilde\Pi \in \mathrm{Lie}(G)$. This element constitutes the perturbation of the Wilson-loop observable around the circle (Aharonov-Bohm phase), which is precisely the unique physical degree of freedom present there. The existence of this new topological mode is of course related to the non-contractibility ($\pi_1(S_1) = \mathbb{Z} \neq 0$) of the circle.

Application of these results to the gluing of the electric field on the circle leads to the following analogous result: the Coulombic adjustments $\eta^\pm$—formally corresponding to the

---

[77]The decoupling of chart transitioning and horizontal gluing can be made into a more general feature. For instance, had we wished to cut up the circle into three segments, we would divide the interval $[0, 2\pi]$ into three sets, $I_1 = [0, 2\pi/3], I_2 = [2\pi/3, 4\pi/3], I_3 = [4\pi/3, 2\pi]$, with $\mathbb{h}_i \in I_i$. Then we can cover the circle with three charts $U_{1,2,3}$, given in larger, but largely overlapping, domains: $D_1 = [0, 4\pi/3], D_2 = [\pi/3, 2\pi], D_3 = [4\pi/3, \pi/3]$. Then $\mathbb{h}_1$ and $\mathbb{h}_2$ glue entirely within the $U_1$ chart domain $D_1$; $\mathbb{h}_2$ and $\mathbb{h}_3$ similarly glue in $D_2$; and $\mathbb{h}_3, \mathbb{h}_1$ glue in $D_3$. In this way, one decouples the chart matching from the horizontal gluing; we can cyclically glue all $\mathbb{h}_i$'s first and find the appropriate chart transition later, independently. In that case, it is the cyclicity of the equations that yields one less condition. This type of concatenating construction can be extended to higher dimensional manifolds.

vertical adjustments $\xi^\pm$ in the gauge-potential case—encode the global *radiative* mode of the electric field on the circle. This global radiative mode is *regionally* of a pure Coulombic form. Then the analogue of $\tilde{\Pi}$ in equation (178) for $\eta^\pm$ is not free, but fixed by the electric flux $f = E_{s|S}$ through the gluing interface. In other words, a locally Coulombic field can be supported by the topology of the circle without any charged source; this is the conjugate dof to the Aharonov–Bohm phase, and what the electric analogue of $\tilde{\Pi}$ physically stands for.

In sum, this 1-dimensional example provides a proof of principle that topological dof of the Aharonov-Bohm kind are not lost in our formalism, but rather emerge as ambiguities in the gluing procedure, ambiguities which are not there in topologically trivial situations.

This consideration only partly endorses the attribution of "new edge mode degrees of freedom" to boundaries [12,73]. Namely, it grants such status only to those, *finitely many* degrees of freedom which encode information about a (global!) nontrivial first cohomology.[78]

# 7 Outlook

We conclude this article by mentioning a few physically relevant questions that we expect our quasilocal framework will address and clarify. We will also take the opportunity to briefly comment on the relationship of the present quasilocal framework with other formalisms proposed in the literature.

**Comparison to edge modes**   The protagonist of this study is the functional connection on field space, $\varpi$, characterized by its projection and covariance properties. In hindsight and to our knowledge, the first appearance of an object possessing those two properties in the context of the symplectic geometry of YM in the presence of boundaries is [12] (see also [13–16, 74] among others). In contrast to the present work, the connection of [12] was built out of *new* gauge-covariant fields; that is, by enlarging the configuration space of the theory and with no field-space geometrical interpretation in mind. These new fields were called "edge modes" since their existence is arguably revealed only at $\partial R$. In the following, we will denote by $\varpi_{\mathrm{DF}}$ the functional connection that corresponds to the construction of [12].

In the case of YM theory (that work considers also the case of general relativity), edge modes were posited to be group-valued, i.e. of the form $\tilde{g}(x) \in G$, and to transform under gauge transformation as $\tilde{g} \mapsto \tilde{g}g$ (on the right). This meant that $\varpi_{\mathrm{DF}} = \tilde{g}^{-1}\mathbb{d}\tilde{g}$ could serve as a (flat) field-space connection[79] and that the following *extended* symplectic potential was horizontal and gauge-invariant: $\theta_{\mathrm{ext}} = \theta_{\mathrm{YM}} - \oint \mathrm{Tr}(f\,\varpi_{\mathrm{DF}})$. Notice that $\theta_{\mathrm{ext}}$ is—on-shell of the Gauss constraint—formally identical to our $\theta^H = \theta - \theta^V$. But the analogy stops there.

Indeed, $\theta_{\mathrm{ext}}$ is labeled *extended* with respect to $\theta$ because it contains the new fields $\tilde{g}$, whereas $\theta^H$ contains *less* modes than $\theta$ and is defined *intrinsically* to the phase space $\mathrm{T}^*\mathcal{A}$. In many ways, the construction of $\theta_{\mathrm{ext}}$ can be understood as a "Stuckelberg-ization" of the gauge symmetry[80] (at the boundary), as can be inferred from the fact that the gauge charges $H_\xi$ (which have no place in $\theta^H$) reappear as charges associated to the "global" symmetries of

---

[78]Of course, this distinction and the ensuing identification of finitely many topological modes cannot be performed at the regional level.

[79]The reader should be aware that many expressions used in this paragraph cannot be found in [12], which is not framed in terms of principal fiber bundles in field space: we are using our language and conventions, to describe their results.

[80]In the fibre-bundle $P \to \Sigma$ description of YM, the edge modes $\tilde{g}(x)$ are nothing else than the bundle's fibre coordinates (in some arbitrary gauge)—and $\varpi_{\mathrm{DF}}$ is the Maurer-Cartan form on the infinite dimensional bundle provided by $\mathcal{A}$. This relationship between edge modes and coordinates is even clearer in general relativity, where the analogue of the fields $\tilde{g}(x)$ are maps $\tilde{X} : \Sigma \to \mathbb{R}^3$ (or from $M \to \mathbb{R}^4$) which are actual coordinates in the sense of differential geometry.

the new fields $\tilde{g}$, i.e. $\tilde{g} \mapsto h^{-1}\tilde{g}$ (on the left). In fact, this simple observation can be made mathematically precise, thus revealing a hidden residual gauge-dependence of the edge mode construction. This analysis, as well as a detailed comparison of edge modes with the present geometric framework, is available in [20].

In light of these considerations, it seems to us that the intrinsic geometric approach put forward in this article is more minimal and more insightful than the one based on group-valued edge modes. Indeed, it only relies on geometric properties that are already present inside standard Yang-Mills theory, and avoids introducing boundary conditions or new fields. This idea is taken to its logical conclusions [20], where the geometric approach developed in this paper is used to show that the reduced phase space $\Phi/\mathcal{G}$ is foliated by canonically-defined symplectic spaces associated to superselection sectors of fixed electric $f$ (see section 3.4 for a brief review).

Therefore, we take the position that there is no a priori reason to introduce the group-valued edge-modes of [12] in YM theory for the study of quasilocal degrees of freedom, charges, or gluing—all of which we have been able to analyze in greater detail from a purely field-space geometrical standpoint. (Having said that, edge modes can nonetheless be useful to model the idealized coupling of a bulk YM theory with other, physical degrees freedom leaving on a codimension-1 surface).[81]

**Edge modes and $\varpi$ in Chern-Simons theory**   At the boundary of a bulk Chern-Simons theory (CS), it is well-known that a boundary Wess-Zumino-Witten theory (WZW) emerges, whose dof are analogous to the edge modes $\tilde{g}(x)$. But, in relation to gauge, the action and the symplectic structures of YM and CS are very different ($BF$ theory offers yet another example, in many ways more similar to CS than YM). The Lagrangian density of YM theory is point-wise gauge invariant, the same is not true for CS; moreover, in YM there exists a (natural) polarization of the symplectic potential which is gauge-invariant (under field-*in*dependent gauge transformations), whereas the same is not true for CS—this lack of invariance was used in [58] to derive the WZW from CS. ) These remarks suggest that it is totally conceivable that edge modes are required in CS but not in YM; [77] make a similar point. However, to settle this point, it is necessary to give a treatment of CS theory through the formalism put forward in this paper; [78] might provide some useful tools to this purpose.

**Comparison to Lattice Gauge Theory**   The introduction of boundaries in (quantum) Lattice Gauge Theory (LGT) requires one to cut open a series of lattice links (see e.g. [4,79] where a second option—cutting along links—is also considered). At the 1-valent vertex of an open link, gauge invariance must necessarily be broken (unless the link carries a vanishing electric flux). This is most easily seen in the spin-network basis of lattice gauge theory [80]. The result of this breaking of gauge symmetry, it is claimed, is that new would-be-gauge degrees of freedom have to be introduced at the 1-valent vertex of LGT. But let us consider two case-studies: a lattice $G = \mathrm{SU}(2)$ and U(1) gauge theories.

Let us start by $G = \mathrm{SU}(2)$. Then, the lattice links are associated with a spin $j \in \frac{1}{2}\mathbb{N}$ (an irrep of $G$) which labels the eigenvalues of the modulus square of the quantum electric flux through a surface dual to the link, $\mathrm{Tr}(f^2) = j(j+1)$; the vertices are labeled by SU(2) invariant tensors (intertwiners); and the 1-valent vertices at the end of an open link carry, as new dof, the SU(2) magnetic indices $m \in \{-j, -j+1, \ldots, j\}$. This means that the boundary states at

---

[81]E.g. to a superconductor confined on a conducting surface [75]. See also [76] for a different coupling to boundary fields, this time represented by spinorial fields.

In the literature one finds other two motivations for the introduction of edge modes that we haven't mentioned so far: the first is based on an analogy with Chern-Simons theory, the second one with (quantum) Lattice Gauge Theory. Their analysis is instructive.

an open link are given by $\|j, m\rangle \in \mathcal{H}_j$. These magnetic numbers are claimed to be a quantum version of the edge modes. Before coming back to this claim let us discuss the other case, U(1).

If $G = $ U(1), lattice links are associated with an integer $n \in \mathbb{Z}$ (an irrep of $G$) which labels the eigenvalues of the quantum electric flux through a surface dual to the link, with the sign of $n$ encoding whether the flux is ingoing or outgoing (relative to the orientation of the link); at the vertices, gauge invariance means that the sum of these oriented flux quantum numbers must vanish (this is Gauss' law). Boundaries are where open (half) links end; if these half-link carry a nontrivial flux with $n \neq 0$, then gauge invariance is manifestly broken there. However, since all irreps of U(1) are 1-dimensional, no extra dof (beside the magnitude of the flux) is present there. Therefore, in the $G = $ U(1) there is no analogous candidate for the quantum edge modes, which according to the construction of [12] should always be present. Why?

The issue is that the magnetic numbers in the $G = $ SU(2) do *not* correspond to the edge modes of [12], but rather to the (quantum) direction that the electric flux is pointing towards in the internal (gauge) space. This can be seen from the fact that the electric flux operator on a given link is proportional to the $\mathfrak{su}(2)$-generator: $\widehat{f}^\alpha = s_i \widehat{E}^{i\alpha} \propto J^\alpha$. This is why, in the U(1) case no analogue of the magnetic numbers is necessary: the internal space is trivial.[82]

Moreover, as argued by [4], in this framework the value of the electric flux $n$ (or $j$) at the boundary is superselected. This means that they (Poisson-)commute with any other observable in the theory, i.e. fluxes become nondynamical and should have no conjugate variables. This is clearly in contrast to what happens in the edge-mode framework, where the edge modes are the conjugate variables to the fluxes themselves.

Now, according to Kirillov's coadjoint orbit method,[83] the Hilbert space $\mathcal{H}_j$ arises as the quantization of the canonically-given symplectic form associated to the coadjoint orbit fixed by $\mathrm{Tr}(f^2) = j(j+1)$ (at a given link) [81]. Therefore the LGT computation nicely matches the classical and continuum construction of [20] (summarized in section 3.4). Once again, this construction requires no edge modes, and rather relies on the restriction of $\theta$ to sectors at fixed $\mathrm{Tr}(f^2)$ (in the Abelian case this is precisely $\theta^H$).[84] This interpretation is further confirmed by computations of entanglement entropy (see below).

Although our construction nicely parallels the LGT phase space in the way presented above, it seems to us that relating gluing in the two pictures is less straightforward.

We showed that in the continuum there is no ambiguity in the gluing procedure and that all dof can be reconstructed by solving certain elliptic boundary value problems. On the lattice, on the other hand, there is no true analogue of the elliptic boundary value problems that enter the gluing formula—which de facto require infinitely fine-grained knowledge of all the continuous modes of the fields involved. Moreover, gluing is highly ambiguous since one can in principle introduce a gauge "slippage" at the gluing of every open link: these missing modes are essentially new Aharonov-Bohm phases not present in the open lattice. And this leads us to the point of contact between the two formalisms: in our study of gluing in 1+1 dimensions (see section 6.8) we found that new Aharonov-Bohm dof are indeed seen to appear when the glued manifold has a nontrivial topology (like a circle).

Let us clarify our argument with a more concrete example. Consider first electromagnetism in $\Sigma = R^+ \cup R^-$, and consider a Wilson loop $L$ which is cut in two by the interface $S = \partial R^\pm$: $L = L^+ \cup L^-$ with $L^\pm \subset R^\pm$. Although there is no way to reconstruct the Aharonov-Bohm

---

[82]The embedding of the link in $\Sigma$, i.e. the lattice discretization itself, projects the electric field $E^i$ in a particular spacial direction.

[83]The name coadjoint orbit method comes from the following: $\eta_f = \mathrm{Tr}(f \cdot)$ is an element of the vector space dual to $\mathrm{Lie}(G)$ whose coadjoint orbit is parametrized by elements $\tilde{g} \in G$ according to $\mathrm{Ad}^*_{\tilde{g}} \eta_f = \mathrm{Tr}((\tilde{g}^{-1} f \tilde{g}) \cdot)$.

[84]The DF extended symplectic structure, which includes the new edge modes, rather than relying on Kirillov's canonical symplectic structure associated to a coadjoint-orbit of a given flux $f \in \mathrm{Lie}(G)$, relies on the canonical symplectic structure associated to $\mathrm{T}^*G$. This is how new dof $\tilde{g}$ are introdued which are conjugate to the $f$. See [20] for details.

phase $\phi(L)$ around $L$ from gauge invariant information associated to its two open "halves" $L^\pm$, this information *is* gauge-invariantly encoded in $R^\pm$. Indeed, turning $L^\pm$ into closed loops $\overline{L}^\pm = L^\pm \cup \ell \subset R^\pm$ by closing $L^\pm$ with a common open Wilson line along the boundary, $\ell \subset S$ and $\partial \ell = \partial L^\pm$, one obviously finds $\phi(L) = \phi(\overline{L}^+) + \phi(\overline{L}^-) \pmod{2\pi}$. Given the nonlinear nature of non-Abelian YM theory, this trick would not work there; however, our gluing result shows that (at least at the linearized level) having access to *all* the gauge invariant information in $R^\pm$ would allow unambiguous gluing even in the non-Abelian theory. However, this information is not available on the lattice, where information about the field configuration is de facto limited to the knowledge of a finite number of Wilson loops.

Therefore, it seems consistent to understand these results as saying that: from the perspective of our framework, LGT behaves as a gauge theory defined on a topologically (highly) nontrivial 1-dimensional manifold, where gluing is *non*-unique and new dof *do* emerge. (This distinction in the quasilocal properties of continuum and lattice gauge theories might have consequences for approaches to quantum gravity, like Loops and Spinfoams, that maintain as fundamental both gauge-like variables and a polymerized, i.e. lattice-like, notion of quantum spacetime [82].)

**Lorentz covariance of the horizontal symplectic form**     Our formalism is founded on a $D+1$ decomposition of spacetime, which manifestly breaks Lorentz invariance. In this regard, it is crucial to appreciate a rather trivial point: prior to the formalism itself, it is the focus on a $(D-1)$-dimensional surface $S$ that breaks global Lorentz invariance.

Indeed, the natural spacetime structure associated with $S$ is given by a pair of disconnected causal domains $J^\pm$ within the globally hyperbolic spacetime $M \cong \Sigma \times \mathbb{R}$. These are the domain of dependence of the regions $R^\pm \subset \Sigma$. But whereas different choices of $R^\pm$ might determine the same $J^\pm$, all these equivalent choices share the same boundary $S = \partial R^\pm$—which means we should write $S = S(J^\pm)$. Thus, even if the spacetime $M$ is a flat Minkowski space, Lorentz invariance is manifestly broken by our focus on $S$, which indeed picks a privileged rest frame (provided $\Sigma \supset R^\pm$ is a simultaneity hypersurface).

More generally, the above causal spacetime geometry suggests that a better notion of spacetime covariance is given by the freedom to foliate the causal domains $J^\pm$. In this regard we think that an interesting future direction consists in studying the quasilocal dynamics within $J^\pm$ by means of the horizontal (and in particular the SdW) decomposition of the gauge fields. This is also the right (covariant) framework to talk about entanglement entropy—discussed below.

We notice that this type of study requires a straightforward generalization of the present formalism to more general foliations with nontrivial lapse (and possibly shift, see [7]), as well as a way to deal with the divergences associated with a vanishing lapse at $S$.

**Superselection Sectors and the Asymptotic Limit**     It has been argued that, in the asymptotic limit $\partial R \to \infty$, $f = f_\infty$ is superselected and that its superselection has highly nontrivial and somewhat puzzling consequences such as the spontaneous breaking of Lorentz symmetry in Quantum Electrodynamics on a Minkowski spacetime (or, indeed, an asymptotically flat one) [10, 83–85].

In the works dealing with the asymptotic case, the superselection of $f_\infty$ follows from the remark that $f_\infty = E_{s|\infty}$ at *infinity* is spacelike separated from, and hence commutes with, *all* the local operators of the theory (since they must have a finite support).

In the case of a finite-region, we argued that $f$ is also superselected (see section 3.4 for a summary, [4] for a lattice perspective, and [20] for a complete treatment in the continuum).

However, whereas in the standard argument for the asymptotic superselection the latter follows from an argument of complete knowledge (one has that $f_\infty$ commutes with *all* local

observables), in the finite case we argued for the superselection of $f$ on a basis of our *ignorance*: adopting a quantum lingo, we are "tracing over" all observables in the complement of the region of interest.

This interpretation is supported by our results on the gluing problem presented in section 6. There we have shown that from a global perspective (one that is not intrinsic to $R$), the flux $f$ at a finite $\partial R$ functionally depends on the quasilocal radiative dof supported *both* on $R$ and on its complement. Therefore, we conclude, the physical origin of the *regional* superselection of $f$ is indeed the "tracing" over the dof contained in the complementary region to $R$ in $\Sigma$.

From this stance, the Lorentz symmetry breaking in Quantum Electrodynamics—which follows from the superselection of $f_\infty$—appears as a consequence (an artifact?) of taking the idealized limit $\partial R \to \infty$ too seriously: i.e. not merely as a large-distance expansion, but as a limit that "pushes" the complementary region to $R$ out of existence.

We find it compelling that this observation resonates with the previous one, on the breaking of Lorentz invariance: in the finite case, it is the presence of a finite boundary (and the tracing out of dof outside it) that *directly* causes *both* the superselection of $f$ and the breaking of Lorentz invariance.

A detailed discussion of the finite-region superselection of $f$ is provided in [20]. However, to fully bridge with the asymptotic case, a detailed study of the role played by the boundary (and fall-off) conditions for the (asymptotic) fields is needed—see e.g. [6, 11]. This work begun in [7], where null-infinity was analyzed, but we leave a more detailed analysis of these ideas to future work.

**Entanglement Entropy**     Another question that we expect our formalism can help clarify concerns the nonstandard properties of entanglement entropy of gauge systems [86]. In gauge theories, the entanglement entropy turns out to quantify not only the standard, "distillable", (quantum and classical) correlations between local excitations, but also a more exotic "edge" (or "contact") component. The latter component is classical, and descends from the probability distribution for finding the super-selected flux $f$ in a certain configuration—i.e. in a certain superselection sector [3, 4, 73, 87, 88].

Given our understanding of the interplay between gauge, fiducial[85] interfaces, and gauge symmetry, it is clear that the present formalism will shed light on the interpretation and computation of the edge component to the entanglement entropy. Indeed, it turns out that the probability distribution of a superselection sector of $f$, as computed in [73, 88], comes precisely from a (Euclidean spacetime) analogue of formula (163) for the Coulombic contribution to the energy (there, the Euclidean action) of an $f$-superselection sector. It is also worth noticing that the Euclidean action featured in the computation of the entanglement entropy by the replica trick is the Euclideanization of the Lorentzian action in the Rindler causal domain.

In [89], a computation of the contact term is proposed which starts from a comparison between a globally gauge-fixed path integral and its regional counterparts. The main ingredient of this computation is the Forman-BFK formula for the factorization of (zeta-regularized, Faddeev-Popov) functional determinants of Laplacians [67, 90, 91] (the relevance of this ingredient to calculations of black-hole entropy was already identified[86] by Carlip [93]). This formula features precisely the Abelian analogue of the operator $(\mathcal{R}_+^{-1} + \mathcal{R}_-^{-1})$ that is central to our gluing formula. Indeed, interpreting horizontal modes as corresponding to the perturbatively gauge fixed ones, our gluing formula gives a precise non-degenerate[87] Jacobian for the

---

[85]Fiducial interfaces—i.e. interfaces at which no fixed boundary condition is imposed—are crucial to the generic definition of entanglement entropy, but for gauge theories they were not easily implementable in previous set-ups (see e.g. the "brick wall" of [73, 88]).

[86]See also [92] for an even earlier application of Forman's results to the gluing, or "sewing", of string amplitudes.

[87]However, subtleties are expected to arise for non-simply-connected manifolds and at reducible background configurations.

transformation of the global radiatives to the regional radiatives, whose determinant yields the relevant factor in the factorization of the path integrals.

More generally, we notice that our formalism is well-suited not only for a broad generalization of the ideas of [89] on the computation and interpretation of the contact term of the (3d Abelian) Yang-Mills theory, but also for inscribing them in a larger theoretical landscape, viz. in the geometry of the Yang-Mills field space.

The first evidence that this is the right direction comes from an analysis of the LGT entanglement entropy computed in [87] with our framework: the non-distillable part of the entropy precisely reflects the foliation of the reduced phase space by symplectic superselection sectors analyzed in [20] and summarized in section 3.4.

**Corners and Gluing** So far we have considered only gluing patterns in which two regions are glued along their *whole* boundaries. More generally, one should consider cases in which the gluing happens on portions of the boundaries bounded by corner surfaces, and the boundary of those corners, and so on until the 0-dimensional boundary terminates the descent. In particular, these more general gluing patterns are necessary to build topologically nontrivial manifolds from topologically simple building blocks (e.g. in the case of triangulated manifolds, or of the trinion decomposition of Riemann surfaces). This is therefore an important topic that deserves deeper study. In section 6.2, we noticed that the continuity condition parallel to the interface $S$ is a verticality condition *in the space of boundary fields*, where the boundary field in question is the difference of the *pull-backs* of the regional horizontals onto $S$. In this scenario, it seems that a chain of descent could apply for horizontal/vertical decomposition at boundaries of boundaries, etc. with analogies to the nested structures featured in the BV-BFV formalism (when interfaces of multiple codimensions are considered) [58, 68, 69]. More generally, it would be valuable to have a precise mapping between, on one side, our reduction and gluing formalisms, which are based on ideas of symplectic reduction, and, on the other, those formalisms such as BV-BFV which are instead based on the "opposite" ideas of (homological, BRST) resolution of the gauge symmetry—such as the BV-BFV formalism of [58, 68, 69], and the theory of factorization algebras of [94].

**The symplectic flow of non-Abelian stabilizer charges** We have argued, in section 4.3, that the only (nontrivial) geometrically-determined set of quasilocal charges that survives symplectic reduction is given by stabilizer charges $Q[\chi_A]$. These charges are only defined at reducible configurations and, in YM, only special configurations are reducible. Reducible configurations constitute "meager" submanifolds of the configuration space $\mathcal{A}$ and are organized along geometrical structures called strata (this is in analogy to metrics admitting Killing vector fields in general relativity, see [24, 25, 27] and [28] for the same constructions in YM). Hence, in YM, the study of the symplectic flow associated with these symmetries must be performed intrinsically to these lower strata of $\mathcal{A}$, where the charges are defined (physically, this corresponds to a restriction to a sector of the gauge-field configurations determined by a given symmetry property—e.g. rotationally invariant solutions in general relativity).

However, in the non-Abelian case, a definition of a connection-form in these strata is not forthcoming, as discussed in section 4.5. Moreover, in the non-Abelian case, the stabilizers are necessarily field-dependent, and thus the relationship between the flow of the stabilizer transformations and the stabilizer charges, as their would-be-Hamiltonian-generators, is potentially obstructed. A more detailed study of the geometry of the strata is needed to better characterize this obstruction and fully clarify its relation to (that is, the curvature of an associated connection, if it can be defined there).

# Acknowledgements

We are thankful to Florian Hopfmüller as well as to Ali Seraj and Hal Haggard for valuable comments and feedback on an earlier version of this work. We also thank William Donnelly for encouraging us to study nontrivial topologies and in particular the 1+1 dimensional example. Finally, our gratitude goes to an anonymous referee whose insightful observations and questions allowed us to considerably improve this manuscript to its present form.

**Author contributions**    All authors contributed equally to the present article.

**Funding information**    HG was supported by the Cambridge International Trust. During the duration of this work, AR was supported first by the Perimeter Institute and then by the European Union's Horizon 2020 programme. Research at Perimeter Institute is supported in part by the Government of Canada through the Department of Innovation, Science and Economic Development Canada and by the Province of Ontario through the Ministry of Economic Development, Job Creation and Trade. This project has received funding from the European Union's Horizon 2020 research and innovation programme under the Marie Skłodowska-Curie grant agreement No 801505.

# A    A quick translation into common notation for (59)

To conclude, we provide a quick bridge to a more common notation for (59) (e.g. [56] or [57]). Let $\mathbb{X}_{1,2} = \int (X_i^\alpha)_{1,2} \frac{\delta}{\delta A_i^\alpha}$ be two tangent vectors on configuration space. In interpreting them as two infinitesimal variations, we denote their components with the more common notation $(X_i^\alpha)_{1,2} \equiv \delta_{1,2} A_i^\alpha$. Then, the horizontal-vertical decomposition of $\delta_{1,2} A$ is given by

$$\delta_{1,2} A_i = (h_{1,2})_i + D_i \eta_{1,2}, \tag{180}$$

where $h_{1,2}$ is the horizontal part of $\delta_{1,2} A$ and $D\eta_{1,2}$ its vertical part.

On-shell of the Gauss constraint and in vacuum, to obtain a complete basis of variations over $T^*\mathcal{A}$, we define the field space vectors

$$\delta_{1,2} := (\delta_{1,2} A, \delta_{1,2} E) = (h_{1,2}, \eta_{1,2}, \varepsilon_{1,2}^{\text{rad}}, \delta_{1,2} f), \tag{181}$$

where we denoted $\varepsilon_{1,2} = \delta_{1,2} E_{\text{rad}}$, and traded the variation of the Coulombic part of the electric field for that of $f$. Then,

$$\begin{cases} \Omega^H(\delta_1, \delta_2) = \int \sqrt{g} \, \text{Tr}\Big((\varepsilon_1)^i (h_2)_i - (\varepsilon_2)^i (h_1)_i\Big), \\ \Omega^\partial(\delta_1, \delta_2) \approx \oint \sqrt{h} \, \text{Tr}\Big(f[\eta_1, \eta_2] + \delta_1 f \, \eta_2 - \delta_2 f \, \eta_1\Big). \end{cases} \tag{182}$$

# B    A brief overview of the slice theorem

Denote $\tilde{A}$ a reducible configuration and by $\tilde{\chi}$ or $\tilde{\chi}_{\tilde{A}}$ one of its reducibility parameters. Let us start by the simple observation that since $(\mathbb{i}_{\tilde{\chi}^\sharp} dA)_{|\tilde{A}} = \delta_{\tilde{\chi}} \tilde{A} = 0$, it follows from the definition (4) that at these configurations of $\mathcal{A}$, $\tilde{\chi}^\sharp_{|\tilde{A}} \in T_{\tilde{A}}\mathcal{A}$ vanishes, thus establishing the degeneracy of the gauge orbit $\mathcal{O}_{\tilde{A}} \subset \mathcal{A}$. Therefore, $\mathcal{A}$ is not quite a bona fide fibre bundle, and its base manifold is in fact a stratified manifold, see figure 8.

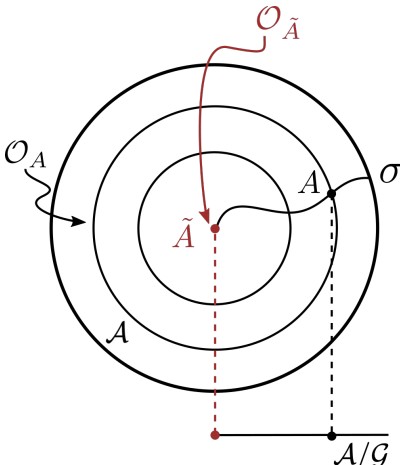

Figure 8: In this representation $\mathcal{A}$ is the page's plane and the orbits are given by concentric circles. The field $A$ is generic, and has a generic orbit, $\mathcal{O}_A$. The configuratoin $\tilde{A}$ has a nontrivial stabilizer group (i.e. it has non-trivial reducibility parameters), and its orbit $\mathcal{O}_{\tilde{A}}$ is of a different dimension than $\mathcal{O}_A$. The projection of $\tilde{A}$ on $\mathcal{A}/\mathcal{G}$ therefore sits at a qualitatively different point than that of $A$ (a lower-dimensional stratum of $\mathcal{A}/\mathcal{G}$). Exclusion of the reducible configuration $\tilde{A}$ gives rise to a fibre bundle structure over $\mathcal{A} \setminus \{\tilde{A}\}$; here $\sigma$ represents a section of $\mathcal{A} \setminus \{\tilde{A}\}$. Whereas $\sigma$ defines a slice through $A$, the slice through $\tilde{A}$ (not depicted) is an open disk centred at $\tilde{A}$.

To be more precise, the definition of a fibre bundle requires a local product structure, and while $\mathcal{A}$ does not have that product structure, it can be decomposed into submanifolds that do. These manifolds are called the *strata* of $\mathcal{A}$, and this result is known as a *slice theorem* for $\mathcal{A}$ [24, 25, 27, 28].

A stratum $\mathcal{N}_A \subset \mathcal{A}$ consists of those connections that have the same stabilizer of $A$ up to conjugacy by $\mathcal{G}$. E.g. generic configurations are irreducible, i.e. have $\mathcal{I}_A = \{\text{id}\}$, and therefore belong to the same (top) stratum; the (bottom) stratum of the vacuum configuration $\mathcal{N}_{A=0}$ is instead constituted by those configurations of maximal stabilizer[88] $\mathcal{I}_{A=0} \cong \mathcal{G}$, i.e. $\mathcal{N}_{A=0} = \mathcal{O}_{A=0} = \{g^{-1}dg, g \in \mathcal{G}\}$. Intermediate strata have an increasing degree of symmetry, $\{\text{id}\} \subset \mathcal{I}_A \subset G$. The slice theorem shows that $\mathcal{A}$ is regularly stratified by the action of $\mathcal{G}$. In particular, all the strata are smooth submanifolds of $\mathcal{A}$.

A "slice" is a notion that reverts to the usual definition of a section on a fibre bundle, but in the presence of stabilizers, it differs in important ways. More precisely, at slice $\mathscr{S}_A$ at $A \in \mathcal{A}$ is an open submanifold of $\mathcal{A}$ containing $A$ such that [27, Def. 1.1]: (*i*) the *entire* $\mathscr{S}_A$ is invariant under $\mathcal{I}_A$, i.e. for all $g \in \mathcal{I}_A$, $R_g \mathscr{S}_A = \mathscr{S}_A$; (*ii*) an orbit will interesect with $\mathscr{S}_A$ only for the stabilizers, i.e. $(R_g \mathscr{S}_A) \cap \mathscr{S}_A \neq \emptyset$ iff $g \in \mathcal{I}_A$, (*iii*) most importantly, the part of the group that is not in the stabilizer heuristically provides the fibres of its own kind of sub-bundle; namely, for an open neighbourhood around the identity coset $\mathcal{U}_A \subset \mathcal{G}_A := \mathcal{G}/\mathcal{I}_A$, and a section $\kappa : \mathcal{U}_A \to \mathcal{G}$, the following map $\Gamma$ is a local diffeomorphism:

$$\Gamma : \mathcal{U}_A \times \mathscr{S}_A \to \mathcal{A}, \qquad ([g], s) \mapsto R_{\kappa([g])} s. \tag{183}$$

---

[88]Contrary to general relativity, there is only one configuration (up to gauge) with maximal stabilizer in YM—at least over a simply connected space, e.g. $R \cong \mathbb{R}^D$, for $G$ a semisimple Lie group. Indeed, suppose $A$ is maximally symmetric, and denote $\{\chi_A^{(\ell)}\}_{\ell=1}^n$ a basis of $\text{Lie}(\mathcal{I}_A) \cong \text{Lie}(G)$, that is $D\chi_A^{(\ell)} = 0$. This implies $[F_A, \chi_A^{(\ell)}] = 0$ at every point in space. Now, since the $\dim(\text{Lie}(\mathcal{I}_A)) = \dim(\text{Lie}(G))$, and since the $\chi^{(\ell)}$ are all linearly independent at every point in space (this is because the equation $D\chi = 0$ is first order), we conclude that $[F_A(x), \text{Lie}(G)] = 0$. If $G$ is semisimple, this means $F_A = 0$, and hence, using that $R$ is simply connected, one concludes that $A = g^{-1}dg = 0^g \in \mathcal{O}_{A=0}$.

So-called "slice theorems" ensure that slices exist at all $A \in \mathcal{A}$ (cf. [28, 29], see also [60]).[89]

At a non-reducible configuration $A$, the stabilizer $\mathcal{I}_A = \{\text{id}\}$ and the definition of a slice collapses to that of a local section. The space of such configurations is open and dense inside of $\mathcal{A}$, i.e. it is generic. At a reducible configuration $\tilde{A}$, however, new features emerge. Call $\mathcal{V}_{\tilde{A}}$ a small open neighbourhood of $\tilde{A}$ (the image of $\Gamma$). Then, the demand (*i*)—that the entire $\mathscr{S}_{\tilde{A}}$ is stable under the action of $\mathcal{I}_{\tilde{A}}$—takes two different meanings depending on whether we focus on neighbouring configurations which take the stratum as an ambient manifold or which take the entire $\mathcal{A}$ as the ambient manifold, i.e. in $\mathcal{V}_{\tilde{A}} \cap \mathcal{N}_{\tilde{A}}$ or in $\mathcal{V}_{\tilde{A}}$.

On the one hand, off the stratum, condition (*i*) means that a suitable $\mathscr{S}_{\tilde{A}}$ contains also the non-trivial orbit of a generic $A \in \mathcal{V}_{\tilde{A}} \setminus \mathcal{N}_{\tilde{A}}$ with respect to $\mathcal{I}_{\tilde{A}}$. So the slice of a reducible configuration is of "higher dimension" than that of a generic configuration.[90] This phenomenon ulitmately underlies our identification of charges.

On the other hand, on the stratum, condition (*i*) means that the slice cuts through the orbits within $\mathcal{N}_{\tilde{A}}$ in a non-generic manner, that is to ensure that for $\tilde{A}' \in \mathscr{S}_{\tilde{A}} \cap \mathcal{N}_{\tilde{A}}$, $\mathcal{I}_{\tilde{A}'}$ is equal to $\mathcal{I}_{\tilde{A}}$ and not just conjugate to it. The existence of $\mathscr{S}_{\tilde{A}}$ means that this "special" cuts exist; and indeed they are usually constructed by exponentiating an orthogonality condition with respect to a gauge-compatible supermetric $\mathbb{G}$ on $\mathcal{A}$ (cf. [28, 29], see also [60]).[91]

## C  List of Symbols

**Space and time**

| | |
|---|---|
| d | De-Rahm differential on $\Sigma$ |
| $g_{ij}, \sqrt{g}$ | the space-like metric on $\Sigma$ and the square-root of its determinant and the square-root of its determinant |
| $h_{ab}, \sqrt{h}$ | the induced metric on $\partial R$, $h_{ab} := (\iota^*_{\partial R} g)_{ab}$ and the square-root of its determinant |
| $N$ | a time-neighbourhood of $R$, $N = R \times (t_0, t_1)$ |
| $R$ | a (compact) subregion of $\Sigma$, possibly with boundary. It is assumed to have trivial topology, $\mathring{R} \cong \mathbb{R}^D$, and smooth boundary |
| $s^i$ | the outgoing normal to $\partial R$ |
| $\Sigma$ | a $D$-dimensional Cauchy hypersurface of spacetime |
| $\int$ | integral over $R$ |
| $\oint$ | integral over $\partial R$ |
| $\nabla$ | the space(-time) Levi-Civita connection |
| $\wedge$ | the wedge product between differential forms (often omitted) |

---

[89]The difficulties in proving the slice theorem all stem from the infinite-dimensional nature of field space: one must show that the orbits are embedded manifolds, and that they are "splitting" (i.e. the total tangent space splits into the tangent to the orbit a closed complement) and that the Riemann exponential map (for some auxiliary gauge-compatible supermetric $\mathbb{G}$) is a local diffeomorphism. One then constructs the slice—whose tangent complements the vertical directions at the given configuration—by exponentiating some neighbourhood of the zero section of the normal bundle to the orbit (the subbundle of $T\mathcal{A}$ which is $\mathbb{G}$-normal to the orbit in question). The $\mathcal{G}$-invariance of $\mathbb{G}$ guarantees that the slice has the necessary properties above. All of this must be done with due consideration of the relevant convergence properties for spaces with the appropriate Holder and Sobolev norms, within a given differentiability class. It is beyond the scope of this paper to exhibit these details (cf. [28]). This appeal to a super-metric shows once again the naturalness of the SdW notion of horizontality.

[90]In finite dimensions, one would have $\dim(\mathscr{S}_A) = \dim(\mathcal{A}) - \dim(\mathcal{O}_A)$ and $\dim(\mathcal{O}_A) = \dim(\mathcal{G}) - \dim(\mathcal{I}_A)$. In the present context, however, all these dimensions are actually infinite except that of $\mathcal{I}_A$ which is finite and bounded from above by $\dim(G)$.

[91]The exponential is equivariant, and "transports" the relevant properties above at $A$ to any other $A'$ in the slice. We discussed a completely analogous construction of transverse sections, this time at generic configurations, in [19, Sect. 9] under the name of Vilkovisky-DeWitt dressing. See there for details.

### Yang-Mills and matter fields

| | |
|---|---|
| $A$ | a Lie$(G)$-valued gauge field configuration ($A \in \mathcal{A} = \Omega^1(R, \text{Lie}(G))$ (a gauge potential over $\Sigma$, in temporal gauge) |
| $\mathcal{A}$ | the space of all field configurations $A$ |
| $(A, E)$ | coordinates on the cotangent bundle of $\text{T}^*\mathcal{A}$ |
| $\text{D}$ | the gauge-covariant differential $\text{D} = \text{d} + A$ |
| $E$ | the electric field (the momentum conjugate to $A$). In temporal gauge, $E = \dot{A}$. See "symplectic geometry" below for the definition of $E_{\text{rad}}$ and $E_{\text{Coul}}$ |
| $F$ | the field-strength of $A$ (magnetic field), $F = \text{d}A + A \wedge A$ |
| $f$ | the electric flux through $\partial R$, $f = E_s \equiv s_i E^i$ |
| $g$ | a (finite) gauge transformation, i.e. an element of $\mathcal{G}$ |
| $G$ | the charge group (finite dimensional, e.g. $G = \text{SU}(N)$) |
| $\mathcal{G}$ | the gauge group (infinite dimensional, $\mathcal{G} = \mathcal{C}^\infty(R, G)$) |
| $\text{G}$ | the Gauss constraint $\text{G} := \text{D}_i E^i - \rho \approx 0$ (see also (53) for $\text{G}_f^{\text{tot}}$ and $\text{G}_f^\partial$) |
| $G_{\alpha,x}(y)$ | the Green's function of the "SdW boundary-value problem" (see (74)) |
| $J^\mu$ | the Lie$(G)$-valued current $J_\alpha^\mu = \overline{\psi}\gamma^\mu \tau_\alpha \psi$, $J^\mu = (\rho, J^i)$ |
| $R_g$ | the action of $\mathcal{G}$ on $\mathcal{A}$ (or $\Phi$, see below) |
| $\text{Tr}$ | a short-hand for the appropriately normalized Killing form on $\mathfrak{g}$ |
| $\gamma^\mu$ | Dirac's gamma matrices |
| $\xi, \eta, \dots$ | an infinitesimal "field-dependent" gauge transformations, i.e. elements of $\Omega^0(\mathcal{A}, \text{Lie}(\mathcal{G}))$ (more generally elements of $\Omega^0(\Phi, \text{Lie}(\mathcal{G}))$, see below). One says $\xi$ is field-*in*dependent if $\text{d}\xi = 0$ i.e. $\xi$ is a constant over $\mathcal{A}$ and can thus be identified with an element of $\text{Lie}(\mathcal{G})$. If $\text{d}\xi \neq 0$, $\xi$ is said field-dependent |
| $\rho$ | the Lie$(G)$-valued matter charge density (see $J^\mu$) |
| $\tau_\alpha$ | a basis of Lie$(G)$ normalized so that $\text{Tr}(\tau_\alpha \tau_\beta) = \delta_{\alpha,\beta}$ |
| $\Phi$ | the total phase space: $\Phi = \text{T}^*\mathcal{A} \times (\Psi \times \overline{\Psi})$ |
| $\psi$ | a matter field; for definiteness, often taken to be a charged Dirac spinor in the fundamental representation of $\mathcal{G}$ |
| $\overline{\psi}$ | the conjugate spinor, $\overline{\psi} = i\psi^\dagger \gamma^0$ |
| $\Psi, \overline{\Psi}$ | the spaces of $\psi$'s and $\overline{\psi}$'s respectively |
| $[\cdot, \cdot]$ | the Lie bracket in $\mathfrak{g}$ |

### Field space geometry

| | |
|---|---|
| $\text{d}$ | the (formal) exterior differential over $\mathcal{A}$ (it commutes with $\text{d}$, and satisfies $\text{d}^2 \equiv 0$) |
| $\text{d}_H$ | the horizontal differential adapted to a covariant horizontal distribution $H = \ker(\varpi)$. Heuristically, it is given by the "covariant" differential $\text{d}_H = \text{d} + \varpi$ |
| $\text{d}_\perp$ | the horizontal differential specific to $\varpi = \varpi_{\text{SdW}}$ |
| $\mathbb{E}$ | the field-space vector on $\mathcal{A}$ built out of $E$ by means of $\mathbb{G}$, $\mathbb{E} = \int g_{ij} E^i \frac{\delta}{\delta A_j}$. The vectors $\mathbb{E}_{\text{rad}}$ and $\mathbb{E}_{\text{Coul}}$ are similarly defined from $E_{\text{rad}}$ and $E_{\text{Coul}}$. See "symplectic geometry" below for their definition |
| $\mathcal{F}$ | the vertical foliation, i.e. $\mathcal{F} = \{\mathcal{O}_A\}$ |
| $\mathbb{F}$ | the curvature of $\varpi$, it encodes the anholonomicity of the horizontal distribution $H \subset \text{T}\mathcal{A}$ |
| $\mathbb{F}_{\text{SdW}}$ | the curvature of $\varpi_{\text{SdW}}$ |
| $\mathbb{G}$ | the "kinetic super-metric" on $\mathcal{A}$ built through the natural $L^2$ metric on $\Sigma$ together with the Killing form $\text{Tr}$ |

| | |
|---|---|
| $H$ | a transverse complement to $V$ in $T\mathcal{A}$, $H \oplus V = T\mathcal{A}$. For brevity, it often stands for $H_{\mathbb{G}}$ (see below) |
| $H_{\mathbb{G}}$ | the orthogonal complement to $V$ with respect to $\mathbb{G}$, $H_{\mathbb{G}} = V^{\perp}$ |
| $\widehat{H}, \widehat{V}$ | projectors from $T\mathcal{A}$ to $H$ and $V$ respectively |
| $\mathbb{h}$ | a field-space horizontal vector (field), $\mathbb{h}_A \in H_A$ |
| $\mathring{\mathbb{i}}$ | the field-space inclusion operator, e.g. $\mathring{\mathbb{i}}_{\mathbb{X}}\mathbb{d}\phi = \mathbb{X}(\phi)$ for all $\phi \in \Omega^0(\mathcal{A})$ |
| $\mathbb{L}$ | the field-space Lie derivative. Acting on field-space forms, $\mathbb{L}_{\mathbb{X}} = \mathring{\mathbb{i}}_{\mathbb{X}}\mathbb{d} + \mathbb{d}\mathring{\mathbb{i}}_{\mathbb{X}}$ (Cartan's formula) |
| $\mathcal{O}_A$ | the orbit of $A$ under the action of $\mathcal{G}$, a subspace of $\mathcal{A}$ |
| $V$ | the vertical subspace of $T\mathcal{A}$, i.e. $V = T\mathcal{F}$ |
| $\widehat{V}$ | see $\widehat{H}$ |
| $\mathbb{X}, \mathbb{Y}, \dots$ | a field-space vector (field), $\mathbb{X} \in \mathfrak{X}^1(\mathcal{A})$, e.g. $\mathbb{X} = \int \mathrm{d}x X_i^{\alpha}(x) \frac{\delta}{\delta A_i^{\alpha}(x)}$ |
| $\varpi$ | a connection 1-form on $\mathcal{A}$, $\varpi \in \Omega^1(\mathcal{A}, \mathrm{Lie}(\mathcal{G}))$. It is adapted to a choice of decomposition $T\mathcal{A} = H \oplus V$ in the sense that $H = \ker(\varpi)$ and $\varpi^{\sharp} = \widehat{V}$. It satisfies the defining properties (8). It can also stand for the pull-back of $\varpi$ to $\Phi$. Often, after section 2.2, $\varpi$ can stand for $\varpi_{\mathrm{SdW}}$ |
| $\varpi_{\mathrm{SdW}}$ | the Singer-DeWitt connection 1-form, i.e. the connection 1-form uniquely adapted to the orthogonal decomposition of $\mathcal{A}$ with respect to $\mathbb{G}$ |
| $\varsigma$ | it is a "potential" for $\varpi$, i.e. $\varpi = \mathbb{d}\varsigma$. This potential exists only under restrictive hypothesis ($G$ Abelian and $\mathbb{F} = 0$) |
| $\cdot^{\sharp}$ | it is the infinitesimal version of $R_g$, it maps a field-independent $\xi \in \mathrm{Lie}(\mathcal{G})$ to a vertical field-space vector, $\xi_A^{\sharp} \in V_A$. This maps extends canonically to field-dependent gauge transformations |
| $[\![\cdot, \cdot]\!]$ | the field space Lie bracket between vector fields $\mathbb{L}_{\mathbb{X}}\mathbb{Y} = [\![\mathbb{X}, \mathbb{Y}]\!]$ |
| $\curlywedge$ | the formal antisymmetric tensor (wedge) product between field-space differential forms |

## Symplectic geometry

| | |
|---|---|
| $E_{\mathrm{rad}}, E_{\mathrm{Coul}}$ | the (functional) components of $E$ entering $\theta^H$ and $\theta^V$ respectively |
| $H_{\xi}$ | the (naive) Noether charge, defined as $H_{\xi} = \mathring{\mathbb{i}}_{\xi^{\sharp}}\theta$ |
| $\mathcal{S}, \mathcal{T}, \dots$ | bulk-supported real-valued function(al)s on $\Phi$, i.e. function(al)s on $\Phi$ which do not depend on the value of the fields in an (arbitrary) collar neighbourhood of $\partial R$ |
| $\mathbb{X}_{\mathcal{S}}, \mathbb{X}_{\mathcal{T}}, \dots$ | the Hamiltonian vector fields associated with $\mathcal{S}, \mathcal{T}, \dots$ |
| $\theta$ | the sum $\theta = \theta_{\mathrm{YM}} + \theta_{\mathrm{Dirac}} \in \Omega^1(\Phi)$. It is the off-shell symplectic potential of YM theory with matter |
| $\theta_{\mathrm{Dirac}}$ | the off-shell symplectic potential of the matter sector of Yang-Mills theory, a 1-form on $\Psi \times \overline{\Psi}$ |
| $\theta_{\mathrm{YM}}$ | the tautological 1-form on $T^*\mathcal{A}$, it is the off-shell symplectic potential of pure Yang-Mills theory |
| $\varphi$ | for the SdW decomposition of $E$ into $E_{\mathrm{rad}}$ and $E_{\mathrm{Coul}}$, one finds $E_{\mathrm{Coul}}^i = g^{ij}\mathrm{D}_j\varphi$ |
| $\Omega$ | the off-shell symplectic form of Yang-Mills theory with matter, $\Omega \in \Omega^2(\Phi)$. Obvious variations are $\Omega_{\mathrm{YM}} = \mathbb{d}\theta_{\mathrm{YM}}$ and $\Omega_{\mathrm{Dirac}} = \mathbb{d}\theta_{\mathrm{Dirac}}$ |
| $\theta^H, \theta^V$ | respectively the horizontal and vertical parts of $\theta$ with respect to a given decomposition $T\mathcal{A} = H \oplus V$. Clearly $\theta = \theta^H + \theta^V$ |
| $\Omega^H$ | is the differential $\Omega^H := \mathbb{d}\theta^H$ (it is necessarily horizontal, but it is not necessarily the horizontal part of $\Omega$) |
| $\Omega^{\partial}$ | it is the differential $\Omega^V := \mathbb{d}\theta^V$ (on-shell of the Gauss constraint it is a pure-boundary term) |

**Reducible configurations**

| | |
|---|---|
| $\mathcal{A}_{\mathrm{EM}}$ | the configuration space of the electromagnetic theory taken as the prototypical example of an Abelian YM theory |
| $\mathcal{G}_A$ | the quotient $\mathcal{G}_A = \mathcal{G}/\mathcal{I}_A$. It is not a group unless $\mathcal{I}_A$ is a normal subgroup of $\mathcal{G}$ (which is a non-generic property) |
| $\mathcal{G}_{\mathrm{EM}}$ | the quotient $\mathcal{G}_{\mathrm{EM}} = \mathcal{G}/\mathcal{I}_{\mathrm{EM}}$. It is a group |
| $\mathcal{G}_*$ | a subgroup of $\mathcal{G}$ homomorphic to $\mathcal{G}_{\mathrm{EM}}$. In this case, $\kappa : \mathcal{G}_{\mathrm{EM}} \to \mathcal{G}_* \subset \mathcal{G}$ is a group homomorphism. The choice of $\mathcal{G}_* \subset \mathcal{G}$ is not unique |
| $\mathfrak{G}_A$ | the quotient of vector spaces $\mathfrak{G}_A = \mathrm{Lie}(\mathcal{G})/\mathrm{Lie}(\mathcal{I}_A)$ |
| $\mathcal{I}_A$ | the stabilizer (or "isotropy") group of $A$, i.e. the subgroup of $\mathcal{G}$ given by those $g \in \mathcal{G}$ such that $A^g = A$ |
| $\mathcal{I}_{\mathrm{EM}}$ | the sub Lie algebra of constant gauge transformations in electromagnetism, $\mathrm{Lie}(\mathcal{I}_{\mathrm{EM}}) \cong i\mathbb{R}$. These transformations stabilize all $A \in \mathcal{A}_{\mathrm{EM}}$ |
| $\mathcal{N}_{\tilde{A}}$ | the subspace of $\mathcal{N}$ composed of all configurations $A$ with stabilizer conjugate to that of $\tilde{A}$. This space is called a (lower) "stratum" of $\mathcal{A}$ (the "top" stratum, which is dense in $\mathcal{A}$, is given by the set of generic configurations with trivial stabilizer; conversely the "bottom" stratum has maximal stabilizer $\mathcal{I}_A \cong G$ and is given by the single orbit $\mathcal{O}_{A=0}$) |
| $Q[\chi_A]$ | the stabilizer charge, $Q[\chi_A] = \int \sqrt{g}\, \mathrm{Tr}(\rho \chi_A)$, which is defined at reducible $A \in \mathcal{N}$ |
| $\mathscr{S}_A$ | a "slice" through $A$. The notion of "slice" generalizes the notion of section at reducible configurations |
| $Q_{\mathrm{EM}}[\chi_{\mathrm{EM}}]$ | the stabilizer charge in electromagnetism. $Q_{\mathrm{EM}}[\chi_{\mathrm{EM}}] = \chi_{\mathrm{EM}} \int \sqrt{g}\, \rho$ is the total electric charge in $R$ (times the constant $\chi_{\mathrm{EM}}$) |
| $\kappa$ | a section $\kappa : \mathcal{U}_A \to \mathcal{G}$ where $\mathcal{U}_A$ is a neighbourhood of the identity coset $[\mathrm{id}] \in \mathcal{G}_A$. With an abuse of notation, we use the same symbol for what is actually the tangent map $\mathrm{T}\kappa : \mathfrak{G}_A \to \mathrm{Lie}(\mathcal{G})$ |
| $[\xi]_A, [\eta]_A, \dots$ | an element of $\mathfrak{G}_A$, $[\xi]_A = [\xi + \chi_A]_A$. It is often simply denoted by $[\xi]$ |
| $\chi_A$ | an element of $\mathrm{Lie}(\mathcal{I}_A)$ |
| $\chi_{\mathrm{EM}}$ | an element of $\mathrm{Lie}(\mathcal{I}_{\mathrm{EM}})$ |

**Gluing**

| | |
|---|---|
| $^S\mathrm{D}_a$ | the gauge-covariant Levi-Civita derivative on $S$ associated to $h_{ab}$ |
| $^S\mathrm{D}^2$ | the gauge-covariant Laplace operator on $S$, $^S\mathrm{D}^2 := h^{ab\,S}\mathrm{D}_a\,^S\mathrm{D}_b$ |
| $\mathbb{H}_A, \mathbb{H}_\psi$ | the components along the gauge-potential and matter-field directions respectively of a SdW-horizontal field-space vector $\mathbb{H} \in \mathrm{T}\Phi$ |
| $H_A, H_\psi$ | the components of $\mathbb{H}_A$ and $\mathbb{H}_\psi$ |
| $h_{ab}$ | the induced metric on $S$, $h_{ab} := (\iota_S^* g)_{ab}$ |
| $\mathcal{R}_\pm$ | the (generalized) Dirichlet-to-Neumann pseudo-differential operator associated with the SdW boundary value problem |
| $R^\pm$ | the two complementary regions in which $\Sigma$ is split, $\Sigma = R^+ \cup_S R^-$ |
| $S$ | the common boundary $S = \pm \partial R^\pm$ |
| $\Sigma$ | the whole Cauchy surface, assumed simply-connected and boundary-less |
| $\bullet^\pm, \bullet_\pm, {}^\pm\bullet$ | indicates which of the regions $R^\pm$ the given object $\bullet$ is associated to (this should *not* be confused with the restriction to a certain region of a globally defined object) |
| $\bullet_{\mid R^\pm}$ | restriction of a globally defined object $\bullet$ to the region $R^\pm$ |
| $[\bullet]_S^\pm$ | the boundary mismatch, defined on regional one-form-valued objects (typically $\bullet \in \Omega^1(R^\pm, \mathrm{Lie}(\mathcal{G}))$) as $\iota_S^*(\bullet^+ - \bullet^-)$ |

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
