# Peer review of "The quasilocal degrees of freedom of Yang-Mills theory"

_SciPost Physics, doi:SciPost Phys. 10, 130 (2021)_

## Round 1 · Referee Report · Prahar Mitra (Referee 1) · 2020-5-6

Strengths

  1. The paper explores an important aspect of gauge theories, namely the degrees of freedom in a local region $R$ of a Cauchy slice $\Sigma$. In particular, they discuss the splitting of the dof into radiative and Coulombic modes. The results are important for the discussion of entanglement entropy of subregions in gauge theories and in the study of asymptotic symmetries where such splittings are studied often.

  2. The paper also discusses how dof in two separate regions $R_1$ and $R_2$ are related to those in $R_1 \cup R_2$ via a gluing procedure. This result is important for the study of entropy inequalities in the context of gauge theories.

  3. The discussion is very thorough. A lot of new formalism has to be used to derive the results of this work and most of the details are presented cleanly.

Weaknesses

  1. The paper discusses the idea of "configuration space" and regularly works with vectors and forms on this space. This is a fairly non-standard manifold and it would be useful to provide explicit formulae when possible - particularly in section 1.3 where a large part of the formalism is introduced (for instance, for $\varpi$). The authors refer to another paper where this formalism has been explicitly spelled out, but where possible it would be good to add additional equations for self-containedness.

  2. The overall set of ideas presented in this paper are closely related to the study of asymptotic symmetries at null infinity that are being studied now, especially in the discussion of the symplectic form and charges. In particular, how do the discussions of this paper change if $\Sigma$ is null?

  3. The authors state that the observable associated to the electric flux $f$ (the pure gauge mode) on the boundary is superselected. This is known to be untrue in the context of "memory effects" where most dynamical processes causes a change in the boundary pure gauge mode. Such a change is also experimentally measurable (in theory!). I suspect that the essential difference between the two is that the authors study a compact $\Sigma$. It might be nice to include this feature in the outlook in section 5.2.

  4. It is emphasized throughout this paper that the discussion differs from others in that no gauge choice is made. However, isn't choosing the SdW form to distinguish between horizontal and vertical modes precisely equivalent to working in the Coulomb gauge, $D^i A_i = 0$? It might be good to clarify this point (if I am confused about this, other readers might also be).

Report

In this paper, the authors discuss the (important) issue of degrees of freedom in gauge theories, especially in the presence of boundaries. The discussion relies on a heavy (but ultimately very useful) mathematical tool of "configuration space. This formalism is introduced in section 2 and then applied to gauge theories in section 3. Section 4 is the most important section in that it describes precisely how dof in disjoint subspaces of $\Sigma$ are glued together and its uniqueness (up to topological modes).

The results of the paper are new and interesting and have applications to several interesting physical problems. I recommend publication of this paper.

Requested changes

I have mentioned the potential changes in the previous sections, but I will summarize them here.

  1. Provide explicit formula for some of the configuration space ideas introduced in section 2 (for example, for $\varpi$).

  2. How does the discussion change if $\Sigma$ is taken to be an asymptotic Cauchy slice, $\Sigma \to {\mathscr I}^\pm\cup i^\pm$. Similar discussions on null infinity are of recent interest so relating the two would be very interesting.

  3. Superselection sectors vs memory effects.

  4. Is choosing the SdW connection the same as working in Coulomb gauge?

---

## Round 1 · Referee Report · Anonymous (Referee 2) · 2020-5-11

Strengths

  1. Innovative perspective on gluing and emergence of physical degrees of freedom, with interesting consequence and ramifications.
  2. Clear and solid mathematical construction (modulo details as reported below)
  3. Self contained presentation of the framework and clear structure of the paper, its objectives and results.

Weaknesses

I do however feel the paper falls short on a number of issues, mostly related to preciseness of statements and (sometimes) mathematical clarity. I have detailed my concerns below. Summarising and highlighting some:

  1. I would like to suggest the authors pay attention to the comments related to their Section 2, where I believe their mathematical setup is imprecise (but fixable). This is especially important when discussing generalisations of this formalism, a few comments on which should probably be given somewhere.

  2. I strongly suggest that the authors curb the paper from as much jargon as possible, and that keep mathematical statements clearly separate from physical interpretation. This is a recurring weakness that I feel should be addressed. There are a number of instances of undefined terminology used to draw conclusions. Speculative interpretations can be perhaps condensed at the end of sections. In particular the redundancy of specifiers like "horizontal", "physical" and "radiative" or "vertical", "gauge", "Coulombic" does not help the reader.

  3. The issue of smoothness of fields in Section 4 requires more attention. See my specific comments about it.

  4. Lastly, more words on comparisons with other approaches, especially when there are contradictory conclusions, should be given.

Report

This paper builds upon previous work of the authors on a field-space geometric approach to gauge theories, aimed at an analysis of regional decompositions of field perturbations and a splitting of degrees of freedom into "gauge" and "physical". In this context this manuscript extends previous work in the context of Lorentzian manifolds and focuses mainly on the behaviour of relevant data upon gluing of subregions in phase space, keeping the symplectic structure under consideration throughout.

The manuscript (as part of a program that the authors admittedly started elsewhere) succeeds in presenting an interesting an innovative point of view on a purely geometric approach to Yang--Mills theory on a (d+1) dimensional Lorentzian manifold, and present a consistent framework to deal with gluing of data associated to subregions in the phase space of the system.

Additionally, by means of a choice of a connection defining a notion of horizontality in field space, the authors propose to match horizontality with the notion of physicality of degrees of freedom and explore the consequences of this choice upon gluing subregions. This choice appears to be guided by the existence of a natural metric on the space of field configurations, and hence a unique connection that is metric compatible. This is called Singer-deWitt connection, and it is used throughout the paper.

The splitting into physical and gauge degrees of freedom induced by the connection brings up a natural question when it comes to gluing: whether this decomposition of fields is preserved upon gluing. The authors show that this is generally the case and argue what are the limitations of this statement, by a careful analysis of the gluing procedure, which is sensitive of global topological data and singularities in the group action (stabilisers). This comes with a nontrivial interplay of physical and gauge fields, that the authors argue can be taken as an explanation of several claims in the literature regarding the emergence of physical modes from gauge theoretic considerations.

The authors propose an analysis of reducible configurations that is interesting per se, and provides a good handle on this issue. All in all, the perspective on the problem that the authors propose is (to my knowledge) innovative and highly detailed, in that it provides a concrete method to explore the gluing of subregions.

I do think that the paper contains new material relevant to the understanding of regional properties of field theories (i.e. on bounded regions), and it provides an intriguing perspective that, to the best of my knowledge, has not been tested before (save on the author's previous works).

In its own, this paper is self contained and clear, making it a capstone summary of this program that, in principle, I could suggest for publication in Scipost, after addressing a few issues.

Attachment

---

## Round 2 · Referee Report · Anonymous · 2021-3-26

# Referee Report
## SCIPOST Submission $202001\_00038v2$ - revision
## The quasilocal degrees of freedom of Yang-Mills theory

## Overview

The revised manuscript has been significantly improved in clarity and mathematical rigor. I would like to thank the authors for having carefully read and implemented the requested/suggested changes in my previous (admittedly very long) report. I do have just a few minor comments that the authors may or may not decide to implement. At any rate, as I do not have other obstructive comments or remarks, I do suggest this manuscript for publication in Scipost.

## Minor comments

1. In the introduction "we in this article limit ourselves" may be rephrased.

2. A general comment on the differential geometry on field space used here, although I understand this is not a strictly mathematical paper: It is probably good to reiterate that most things are done within the setting of "local" calculus, and not as general differential geometry on Frechet manifolds, for which much more detail would unfortunately be needed: the dual of a Frechet space is not Frechet, so one needs to define what cotangent bundles mean. Except from the connection form, everything seems to be "local" to me, in the sense of pullback from the (infinite) jet bundle. The "cotangent bundle" considered here is the fiberwise dualisation of the vector bundle whose sections are the fields. This could be a decent way to clarify in what sense all of these quantities are understood.

3. After (1): the normal components of $A$ naturally do not show up on a "time slice" $\Sigma$. I do not think this is a "gauge". Rather, the "quasilocal YM configuration space" is a subspace of the space of initial data (over R), in the sense of a Cauchy problem (naturally, one would specify $A, E$ at $\Sigma$, on shell of Gauss' constraint). I guess the authors might be after a slightly different perspective here, but I am still not sure this should be considered as a gauge, since the restriction map $\iota^*$ for $\iota \colon \Sigma \to \Sigma \times \mathbb{R}$ "kills" the transversal part of a connection $A_t dt$.

4. Similarly, the authors consider $\mathcal{A}/\mathcal{G}$ as the "true" configuration space of the theory. I guess this makes sense, although it seems to me that the more relevant space to look at would be the symplectic reduction of C inside the phase space, where C denotes configurations that satisfy Gauss' law. I would call that the reduced phase space (i.e. what the authors use in 3.4).

5. After (9): typo "tcovariance"

6. After Def 2.3. The authors define forms $\lambda \in \Omega^k(\mathcal{A}) \otimes \Gamma(\Sigma, W)$. Do they really want to consider forms on the space of fields with values in sections of a vector bundle on which there is an action of $G$? I found this confusing for I do not see an obvious application for such a general object, so perhaps a couple words might be useful for the reader?

7. In section 3: what is a "canonical completion" of a symplectic structure?

8. The ideas relating to a modification of the symplectic reduction picture, presented in section 3.4 are interesting. It is a little confusing though that a general theorem is invoked according to which the reduced form on $\Phi//\mathcal{G}$ IS symplectic, but then it is said not to be. I would fix this by making it clear that the authors are taking inspiration from standard symplectic reduction techniques, and provide a similar construction which shows a behaviour sensitive of boundaries. This is implied by the title, but it remains slightly unclear to the reader whether a general theorem is invoked (and in my understanding, it isn't).

---

## Round 2 · Author Response

First, we would like to thank the referee for their report. They were examplary: incredibly dilligent and constructively critical.

We would also like to apologize for the inordinately long hiatus between the receipt of the referee reports and our response. Needless to say, this was a tough year, and we both had more than a few pandemic-related personal problems that delayed attention to the paper by many months. Hopefully, you will understand our situation.

We have now resubmitted an updated draft, which we believe satisfactorily amends the paper as per both referees’ recommendations.

The referee’s questions and suggestions are very relevant and we believe that their. observations allowed us to considerably improve the current manuscript. To respond to their most cogent concerns, we decided to do a major rewriting of the entire first part of the manuscript (what are now section 1, 2, 3, 4). One significant change we made was to adopt a Definition/Proposition/Proof format for our most important statements throughout the entire manuscript. This forced us to restructure our presentation in a way that we believe makes our statements clearer and easier to navigate.

Please note that Section 3.8 was significantly expanded, becoming our new section 4. That is because an extension of our geometric formalism that encompasses non-Abelian reducible configurations is complicated, and a full exposition would require substantially more work. For these reasons, even limiting ourselves in this article to laying down some general considerations on the non-Abelian case and leaving the detailed analysis of the symplectic geometry associated to these charges to future work, a substantial expansion was required.

Below we will answer the referee's queries one by one.

---

## Round 2 · List of Changes

ANSWER TO ANONYMOUS REPORT 1

First, we would like to thank the referee for their report.

We would also like to apologize for the inordinately long hiatus between the receipt of the referee reports and our response. Needless to say, this was a tough year, and we both had more than a few pandemic-related personal problems that delayed attention to the paper by many months. Hopefully, you will understand our situation.

We have now resubmitted an updated draft, which we believe satisfactorily amends the paper as per both referees’ recommendations.

The referee’s questions and suggestions are very relevant; in particular, those related to configuration v. phase-space and those related to the properties of asymptotic infinity. The latter touch a topical point that has fostered a lot of research in the last five years. Indeed, the very questions they are asking have always been part of our motivation to work on this very topic, i.e. the nature of the quasi-local degrees of freedom in gauge theories (and gravity). Moreover, some of the referees’ comments spurred research that has led to enough new results to require a separate publication (although we made sure that the present manuscript takes into account these new results, their proofs are not contained in here. See in particular section 3.4).

We would like to point out that, in order to satisfactorily answer the referees’ comments and thus improve the quality of the manuscript, this second version has seen major re-writings of the entire first part of the paper. This means that section 1, 2, 3 have been largely rewritten. Section 4 is what used to be section 3.8 (see below). Section 5 is just a promotion of what used to be appendix D (no changes there). And finally section 6 (the old section 4 on “gluing”) has seen only minor edits.

Section 3.8 was significantly expanded, becoming our new section 4. That is because an extension of our geometric formalism that encompasses non-Abelian reducible configurations is complicated, and a full exposition would require substantially more work. For these reasons, even limiting ourselves in this article to laying down some general considerations on the non-Abelian case and leaving the detailed analysis of the symplectic geometry associated to these charges to future work, a substantial expansion was required.

We proceed to answer the referee's points one by one here below:

1. \textit{Re: Configuration Space}.
In the new draft we switched from a treatment in the non-standard configuration space to one in phase space. We think this change positively impacted both the exposition and the mathematical rigor of our results. We have also re-written the introductory part of the paper to make it more self-contained and to include suggestions put forward by the other referee.

2 & 3. \textit{Re: asymptotic infinity and superselection.}
The main reason we have not addressed the questions on the asymptotic limit in which $\Sigma \to \mathscr{I}$ in this work is that this would have required the introduction of yet another set of advanced concepts and subtle limiting procedures, not to mention a whole new set of questions to be addressed. For these reasons we have consciously decided to limit the present work to the topic of ``finite boundaries,’’ while striving to be as exhaustive as possible in their treatment.

Nonetheless, a first set of questions concerning the asymptotic limit was approached by one of us in a separate publication (arXiv:1904:07410v2, published in JHEP). This publication, which heavily relies on the present one, features encouraging results: (1) proof is given of the fact that, in the asymptotic limit, our definition of radiative phase space recovers that of Ashtekar and Streubel, and (2) whereas the bulk notion of symmetry is heavily restricted to a finite number of “global’’ generators, in the asymptotic limit an infinite set of symmetries and their soft charges emerge thanks to \textit{both} the null nature of $\mathscr{I}$ and the presence of appropriate fall-off conditions (here, symmetries are understood as would-be gauge transformations in the kernel of $\varpi$).

In this context, as physically demanded, the electric flux does not need to be the same at the initial and final cuts on $\mathscr{I}$; thus what would be a physically dubious restriction is never introduced. No contradiction with the memory effect is present. (Another, more heuristic, way to see this is that at asymptotic infinity the boundary of $\Sigma$ is given by two disconnected components which are the $r \to \infty$ limit of an inner and outer sphere bounding a shell-like region; in this geometry, one is free to fix f independently on both boundaries.)

Still, more work is certainly needed to understand how to frame the sub-leading soft charges and the memory effects in the language developed in the manuscript reviewed here. To be dealt with in the appropriate detail, we think all these developments deserve (and require) a separate publication.

Concerning the superselection of the electric flux in the present manuscript, we would like to emphasize that this statement holds (1) at finite boundaries $\partial R$, and (2) in relation to all (gauge invariant) operators supported in the \textit{interior} of the region $R$. More covariantly, the physics pertaining to a (D+1)-dimensional causal diamond $D(R)$, identified by its (D-1)-dimensional $\partial R$, is determined by the flux $f$ through $\partial R$ (which is a well-defined spacetime scalar quantity). However, all physical quantities \textit{within} $D(R)$ commute with the flux $f$. This is discussed in in more detail the outlook section. Also, we would like to point out that more results on the topic of superselection can now be found in arXiv:2010.15894 by one of us. A brief review of these results is now also incorporated in the new version of this manuscript.

4. \textit{Re: the choice of connection/gauge.}
The referee is correct in pointing out that a choice of functional connection and gauge-fixing are related (although they are mathematically not the same thing). One of the most important updates of this second versions regards precisely this: new results have been included which show that all our constructions, at the end of the day, do \textit{not} depend on the choice of functional connection. This include the symplectic reduction and the gluing. Unfortunately it was not possible to include the proof of the results on the symplectic reduction in the present manuscript, which is already quite long. Those results whose proof would have been too lengthy to include have nonetheless been reviewed (see section 3.4, and the separate publication already cited above, arXiv:2010.15894).

===========================

ANSWER TO ANONYMOUS REPORT 2

First, we would like to thank the referee for their report. They were examplary: incredibly dilligent and constructively critical.

We would also like to apologize for the inordinately long hiatus between the receipt of the referee reports and our response. Needless to say, this was a tough year, and we both had more than a few pandemic-related personal problems that delayed attention to the paper by many months. Hopefully, you will understand our situation.

We have now resubmitted an updated draft, which we believe satisfactorily amends the paper as per both referees’ recommendations.

The referee’s questions and suggestions are very relevant and we believe that their. observations allowed us to considerably improve the current manuscript. To respond to their most cogent concerns, we decided to do a major rewriting of the entire first part of the manuscript (what are now section 1, 2, 3, 4). One significant change we made was to adopt a Definition/Proposition/Proof format for our most important statements throughout the entire manuscript. This forced us to restructure our presentation in a way that we believe makes our statements clearer and easier to navigate.

Please note that Section 3.8 was significantly expanded, becoming our new section 4. That is because an extension of our geometric formalism that encompasses non-Abelian reducible configurations is complicated, and a full exposition would require substantially more work. For these reasons, even limiting ourselves in this article to laying down some general considerations on the non-Abelian case and leaving the detailed analysis of the symplectic geometry associated to these charges to future work, a substantial expansion was required.

Moreover, as per the referee’s suggestion, we promoted the old Appendix D to what is now section 5. (Its content has been left untouched.)

The gluing section (old section 4) is now section 6. This section has seen only minor revisions, as per the referee’s suggestions.

We proceed to answer the referee’s points one by one here below, following the structure of their own report:

OVERVIEW

1. Following the referee’s suggestions we did improve on the mathematical exposition of the our framework (See below)

2. In the re-writing of the manuscript we have taken into account the referee’s suggestion to make the language more precise.

3. In the discussion of gluing, the issue of smoothness is indeed a subtle problem. We have been more careful in the new statement. However, framing gluing in an “existence and uniqueness” framework, we decided to focus on the uniqueness statement only. Indeed, our main focus was to disprove the common belief that gluing is ambiguous in gauge theories, not to give necessary and sufficient conditions for gluing to be possible. We thought that this question, albeit quite interesting, goes beyond the scope of the present paper. In the new manuscript we state this limitation clearly.

4. In the outlook section we included a thorough comparison with the most similar of the other proposed approaches to deal with gauge and boundaries, the “edge mode” approach. After consultation with practioners in the field of BV-BFV, we have concluded that those approaches are too far removed from ours to allow a detailed comparison at this stage.

SECTION 1

The referee’s comments on our introduction section are well taken. We restructured the whole introduction, which is now much shorter and does not wrestle with the attempt to give a detailed summary of the results before laying out the necessary formalism and notions. As a consequence of our edits most of the issues they point towards in this section do not apply anymore. Those which still apply in relation to other parts of the paper will be addressed below.

SECTION 2

This section has been largely revised. The most important changes — which apply to entire manuscript — are:

A) We included a more mathematical perspective, clarifying in particular the role of field-dependent gauge transformations in terms of a Lie algebroid structure on field space (we thank the referee’s for pointing out the relevant literature). To make our statements more clear, we have also turned to a Definition/Proposition/Proof format for our most important statements.

B) In this new version we at all times be work in phase space (i.e. in the *co*tangent bundle of $\mathcal{A}$) rather than in configuration-velocity space (the tangent bundle). This makes the introduction of the electric field more straightforward and clear. In particular, there is no need to deal with time derivatives any more; the (off-shell) symplectic potential arises as the tautological 1-form on the cotangent bundle of $\mathcal{A}$; and finally, there $\varpi$ lifts from $\mathcal{A}$ to $T^*\mathcal{A}$ simply by pullback relative to the canonical projection.

Here below, we address the referee’s concern in the order they are raised:

1. Thanks to B) above, the confusion between E and the time derivative of A does not arise anymore.

2. Also does not apply anymore; not only because that equation does not appear, but we have also amended the definition of $\xi$ so to allow that kind of equation to make sense (see equation 5 on pg 7).

3. Again, thank to B) above this is not an issue anymore. Moreover, we do not introduce redundant notation (Now $\Phi$ is used only in the presence of matter, in the place of a more intricate expression — see eq 37 on p.18).

4. We agree that a lot of the formalism goes through for more general situation where one has a foliation but not a PFB structure. However, in YM theory (and gravity) a local PFB structure is given by results quoted in section 2 below equation (3).

5. “Fiducial”: we now include a brief definition of fiducial boundaries (not regions), in the introduction (page 3), and in footnote 62 at page 51.

Further questions:

1. We have clarified the locality properties of $\varpi$ in the paragraph just below definition 2.1 and especially in remark 2.2 (page 7 and 8). This becomes particularly clear in the following sections on the SdW connection and Dirac dressing, to which we refer.

2. Considering field-dependent gauge-transformations allows for a more flexible differential-geometric framework in field space which will be crucial in 1) discussing the symplectic reduction (new section 3.4 which summarizes the results of one of us in arXiv:2010.15894) and 2) in the gluing section, where the gluing procedure relies precisely on the selection of appropriate field-dependent gauge transformations.
We have made reference to these applications when we introduce field-dependent gauge transformations at the very beginning of the paragraph below eq 4 (top of page 7).

SECTION 3

1. We have amended this language issue. We now referr to $\theta$ (defined now as the tautological 1-form on $T^*\mathcal{A}$) as the “off-shell symplectic potential” and explain that “off-shell” refers to the fact that the fields are there considered to be off-shell of the Gauss constraint. Indeed, the field configurations satisfying the Gauss constraint are a subspace of $T^*\mathcal{A}$ to which we will have to eventually restrict ourselves to.

2 and 3. In the new formulation, which is now based on phase space rather than on the configuration-velocity space, the supermetric is introduced more abstractly already in section 2, before the introduction of the symplectic potential. We simply state that: if a supermetric is given which satisfies appropriate gauge covariance properties, then a functional connection can be deduced from it. Whether such a connection satisfies nice compatibility properties with the symplectic structure of the theory is a different question that has to be analyzed on a theory-by-theory basis. In YM, which is a second order theory, the kinetic supermetric provides a natural choice of connection which has a nice interplay with the symplectic structure of the theory. In a Chern-Simons theory, we would not know how to make that choice (which does not mean one such choice cannot exist, of course). Finally, one last important point: compared to the previous version of the manuscript, we have now de-emphasized the role of the SdW connection. It is an extremely useful example which plays an important role in our formalism (see e.g. section 3.4, or the combination of Lemma 6.2 and Theorem 6.1 on gluing). And, although it is the route by which we obtain many of our results, it is eventually shown to be just a crutch: we show our physically relevant results are *independent* of the choice of connection.

4. We thank the referee for this careful remark. We implemented the change (pg. 8 below eq 9)

5. This question is about a providing a sketch of proof of what is currently equation 27. This is now available in the proof of Prop 2.16.

6. Thanks to B above, this question does not apply anymore.

7. On the use of the word “Physical”. In the new version of the manuscript, we avoid to conflate 'physical' with horizontal. We have only stressed, in page 7, a definition of a physical direction in the space of gauge configurations, $\mathcal{A}$, as one that has a transverse component to the gauge orbit, and therefore has a horizontal component for this space as well.

8. We have already answered to this above (see SECTION 2 Further questions 2).
9. The paragraph the referee refers to has now been replaced by a more succinct and rigorous discussion at the end of section 3.3, which starts from corollary 3.7.

SECTION 4 (which is currently Section 6 — on Gluing)

We agree with the referee that our previous treatment was confusing regarding the fact that we prove only uniqueness but not existence of a “smooth” gluing (and we were ambiguous on the meaning of “smooth” as well). In the new version we make clear that we concern ourselves only with the proof of uniqueness, not existence. In other words, we do not try to analyze which conditions the regional $h_+$ and $h_-$ must precisely satisfy for a “smooth” gluing $H$ to exist. We simply state that if a gluing $H$ exists which is $C^1$, then it is unique (Lemma 6.2). We also generalize this statement to a more general, albeit formal, uniqueness theorem hat holds for any choice of functional connection (Theorem 6.1).

1. Although the uniqueness of gluing in the form of theorem 6.1 or lemma 6.2 is available for any Lie-algebra valued gauge potential, what is special to YM is that we can deduce from this result a result for the gluing of the symplectic structures (Theorem 6.4). In other words, we can leverage this result to further our understanding of the behaviour of the quasilocal dof of YM upon gluing. For a similar result to hold in other gauge theories such as Chern-Simons theory, we would need to analyze the properties of their symplectic structures in relation to gauge symmetry on a case-by-case basis. We know emphasize this fact in the last paragraph of the introduction as well as in the outlook section. For further remarks on this issue we refer also to our answer to questions 2 & 3 of section 3, above.

2. We agree with the referee that it is desirable to generalize our results to more general types of symmetries, but we think this goes beyond the scope of the present work. However, we do hope to study these issues in different theories with more general symmetry structures, especially in the context of general relativity and diffeomorphism symmetry.

3. We have added a clarificatory paragraph below what is now equation 153. The point is that the $\chi^+$-transformation is not to be thought of as an “asymmetric gauge transformation” (indeed W fails to be “gauge invariant” under any discontinuous gauge transformation), but as a procedure that creates two distinct global configurations from the gluing of the same doublet of regional configurations. For this to be possible $\chi^+$ must arise as an ambiguity in the determination of the vertical adjustments $\xi^\pm(h^+, h^-)$—a demand that implies that $\chi^+$ is a regional stabilizer. Then, what we show is that $W$ is a gauge invariant functional of the global (glued) state that is sensitive to that ambiguity in the gluing.

4. The referee’s question now refers to the last paragraph of section 6.3, which we have now modified to clarify my argument.
5. As elsewhere we agree that the language employed was misleading, and we have accordingly modified our presentation, and removed the most confusing paragraphs.

6. We thank the referee for this careful remark. We implemented the change just below the current eq 159.

7. The referee’s question now refers to equation 168. We have clarified the point by commenting right below this equation.

8. We improved on the language of what is now the last paragraph of section 6.7.

9. We agree that the footnote was confusing. We have decided that removing it altogether was the best option.
10 and 11. We agree with the referee, and we did not mean to contrast our approach (in that respect) to any other in particular. We now have re-written the last three paragraphs of (what is now) section 6.8.2, to clarify what we meant.

APPENDIX D

Following the referee’s recommendation this has now been promoted to be the current section 5.

GENERAL QUESTIONS:

1. We do not know of any geometrical meaning to the balance of energy referred to by the referee, apart from the following, which is already stated at different parts of the text: the global horizontal mode does not restrict to regional horizontal modes; there is a difference, which gives rise to the extra term in the energy. Thus one could perhaps interpret this result as saying that the global orthogonality to the fibers does not commute with restriction; that is: what is globally orthogonal to the fibers does not automatically restrict to what is regionally orthogonal to the fibers.

2. We will cite these references, but their mathematical language and their context seem to be significantly different and aimed at a different audience. Therefore, we feel a comparison would take us very far afield in an already long paper.

3. The referee’s question about the reliance of our construction from the SdW connection is very good. We now address it at several places in the new version. In particular, we have split statements that don't need the SdW connection from those that do. We now also include a summary of (Riello 2010.15894, reference 20), which shows that the quasilocal reduction is independent of the choice of connection (see Section 3.4), and an extension of our gluing theorem to encompass any choice of connection.

---

## Round 3 · Referee Report · Anonymous (Referee 2) · 2021-5-18

Report

All of my previous comments have been answered.
I do recommend this article for publication in Scipost.

---

## Round 3 · Author Response

First, we would like to thank again the referee for the comments which were instrumental for the sharpening the paper.
We attach our answers to this second, brief, round of comments.

---

## Round 3 · List of Changes

We have corrected the more perfunctory comments and answered the more substantial ones as follows (the numbering follows the referee's report):

(2) We added a brief paragraph at the beginning of section 2 to clarify the scope of the functional space we work in.

(3) After eq (1) we removed reference to temporal gauge.

(4) We corrected the accidental mischaracterization of P = A/G as the reduced phase space.

(6) We kept reference to the (extremely) general space of forms solely as an example that does not reoccur.

(7-8) We added two short paragraphs ("We refer to..." and "Mathematically...")) at the end of section 3.4 clarifying the status of the “canonical completion” and symplectic reduction in the presence of boundaries. However, we kept reference [20] as the main source of details on the topic.

---

## Editorial Decision

published